# SHARP GENERALIZATION FOR NONPARAMETRIC RE­GRESSION IN INTERPOLATION SPACE BY SHALLOW NEURAL NETWORKS WITH CHANNEL ATTENTION

## ABSTRACT

We study nonparametric regression using an over-parameterized two-layer neu­ral networks with channel attention in this paper, where the training features are drawn from arbitrary continuous distribution on the unit sphere in $\mathbb{R}^d$, and the target function lies in an interpolation space commonly studied in statistical learn­ing theory. We demonstrate that training the neural network with early-stopped gradient descent achieves a sharp nonparametric regression risk bound of $\mathcal{O}(\varepsilon_n^2)$, where $\varepsilon_n$ is the critical population rate of the kernel induced by the network with channel attention, and such risk bound is sharper than the current state-of-the-art regression risk (Yang, 2025) on the distribution-free spherical covariate. When the distribution of the covariate admits a widely studied eigenvalue decay rate with parameter $2\alpha$ such that $\alpha > 1/2$, our risk bound becomes $\mathcal{O}(n^{-\frac{6\alpha}{6\alpha+1}})$ when the target function is in an interpolation space with widely studied spectral bias in deep learning. This rate is even sharper than the currently known nearly-optimal rate of $\mathcal{O}(n^{-\frac{6\alpha}{6\alpha+1}}) \log^2(1/\delta)$ (Li et al., 2024), where $n$ is the size of the training data and $\delta \in (0,1)$ is a small probability. Our analysis is based on two key tech­nical contributions. First, we establish a principled decomposition of the network output at each GD step into a component lying in the reproducing kernel Hilbert space (RKHS) of a newly induced attention kernel and a residual term with small $L^\infty$-norm. Second, building on this decomposition, we employ local Rademacher complexity to obtain sharp bound for the complexity of the function class formed by all the neural network functions along the GD steps. Our findings further in­dicate that channel attention enables neural networks to escape the linear NTK regime and achieve sharper generalization than the vanilla neural network without channel attention, with the kernel complexity of the channel attention kernel lower than that of the standard NTK induced by the vanilla network. Our work is among the first to reveal the provable benefit of channel attention for nonparametric re­gression, with simulation results on both synthetic and real datasets.

## 1 INTRODUCTION

The remarkable success of deep learning (LeCun et al., 2015) has motivated theoretical studies on the optimization and generalization of deep neural networks (DNNs). Many works show that gradi­ent descent (GD) and stochastic gradient descent (SGD) can drive training loss to zero under various settings (Du et al., 2019b; Allen-Zhu et al., 2019; Du et al., 2019a; Arora et al., 2019; Zou & Gu, 2019; Su & Yang, 2019). In parallel, generalization theory seeks algorithmic guarantees on the error of gradient-based methods. An early direction is the Neural Tangent Kernel (NTK) (Jacot et al., 2018), showing that highly over-parameterized networks behave similar to kernel methods. Since weights stay near initialization, first-order expansion enables tractable analysis (Cao & Gu, 2019; Arora et al., 2019; Ghorbani et al., 2021), and it is shown that infinite-width models can also capture feature learning (Yang & Hu, 2021). Beyond NTK's linear regime, higher-order approximations such as QuadNTK (Bai & Lee, 2020), hybrid methods (Nichani et al., 2022), and mean-field anal­yses (Damian et al., 2022; Takakura & Suzuki, 2024) address the feature learning effects of neural networks.

The theoretical deep learning literature also studies nonparametric regression by DNNs with noisy data. DNNs can achieve minimax rates for smooth (Yarotsky, 2017; Bauer & Kohler, 2019; Schmidt-Hieber, 2020; Jiao et al., 2023; Zhang & Wang, 2023) and non-smooth (Imaizumi & Fukumizu, 2019) targets, but many results lack algorithmic guarantees, relying on special architectures not realized by GD. More recent works (Hu et al., 2021; Suh et al., 2022; Li et al., 2024; Yang & Li, 2024; Yang, 2025) advance this direction by analyzing generalization of DNNs trained with GD or SGD. The goal of this paper is to reveal the theoretical advantage of channel attention in neural networks for nonparametric regression, and the related works in this direction are reviewed below.

**Existing Empirical and Theoretical Works about Channel Attention and General Attention Mechanism.** Popular channel attention methods (Fu et al., 2019; Wang et al., 2020; Ali et al., 2021) enhances DNN representations by adaptively reweighting channels. In particular, XCiT (Ali et al., 2021) views channel attention as cross-covariance across features, showing strong performance for classification. Covariance pooling (Chen et al., 2025; Song et al., 2021; Wang et al., 2023) has been applied to channels with various theoretical results such as stability of DNNs and preconditioner effect. Kernelizable attention is studied in (Choromanski et al., 2021; Peng et al., 2021; Zheng et al., 2023) with fast attention matrix approximation, and (Hron et al., 2020) studies the behavior of multi-head attention architectures as Gaussian Processes with infinite number of heads. While few works, such as (Kim et al., 2024), study the optimality of attention-based neural networks on in-context learning (ICL) tasks, the theoretical benefits of the attention mechanism, especially channel attention, on standard nonparametric regression tasks remain largely unknown.

In this paper, we investigate nonparametric regression with an over-parameterized two-layer neural network equipped with the XCA-style channel attention (Ali et al., 2021) and trained by GD. With the target function lying in an interpolation space characterized by a spectral bias, formally introduced in Section 2.2, we establish that early-stopped training with channel attention attains a sharp risk bound of $\mathcal{O}(\varepsilon_n^2)$ for arbitrary continuous distribution on the spherical covariate, where $\varepsilon_n$ is the critical population rate of the kernel induced by the network with channel attention. The sharpness of such risk bound is reflected by the fact that the risk bound $\mathcal{O}(\varepsilon_n^2)$ is minimax optimal in the special case where the covariate distribution admits a polynomial eigenvalue decay rate (EDR), and such risk bound is sharper than that in the current state-of-the-art (Yang, 2025) on the distribution-free spherical covariate, or the risk bound in (Li et al., 2024) with the same EDR where the target function also lies in the same interpolation space. Detailed comparison to the current state-of-the-art is summarized in Section 3 and also shown in Table 1. To the best of our knowledge, our work is among the first to reveal the theoretical benefit of channel attention in neural networks for canonical nonparametric regression in the interpolation space, and channel attention provably helps the network learn spectrally biased target functions widely studied in deep learning (Rahaman et al., 2019; Arora et al., 2019; Cao et al., 2021; Choraria et al., 2022; Wang et al., 2024; 2025). Our results show that the network with channel attention induces a new kernel, referred to as the attention kernel, as opposed to the standard NTK induced by the counterpart network without channel attention. The attention kernel enjoys reduced kernel complexity compared to NTK, and our result is beyond the linear region of conventional NTK.

We organize this paper as follows. The rest of this section introduces the necessary notations. Section 2 details the problem setup, including the definition of the interpolation space. Section 3 presents our main results in detail, and Section 4 introduces the training algorithm for the over-parameterized two-layer neural network with channel attention by GD. The proof roadmap with our key technical results and the novel proof strategies are introduced in Section 5. Simulation results on synthetic and real data are deferred to Section E of the appendix.

**Notations.** We use bold letters for matrices and vectors, and regular lowercase letter for scalars throughout this paper. The bold letter with a single superscript indicates the corresponding column of a matrix, e.g., $\mathbf{A}^{(i)}$ is the $i$-th column of matrix $\mathbf{A}$, and the bold letter with subscripts indicates the corresponding rows or elements of a matrix or a vector. We put an arrow on top of a letter with subscript if it denotes a vector, e.g., $\vec{\mathbf{x}}_i$ denotes the $i$-th training feature. $\|\cdot\|_{\mathrm{F}}$ and $\|\cdot\|_p$ denote the Frobenius norm and the vector $\ell^p$-norm or the matrix $p$-norm. $[m:n]$ denotes all the integers between $m$ and $n$ inclusively, and $[1:n]$ is also written as $[n]$. $\mathrm{Var}\left[\cdot\right]$ denotes the variance of a random variable. $\mathbf{I}_n$ is a $n \times n$ identity matrix. $\mathbb{I}_{\{E\}}$ is an indicator function which takes the value of 1 if event $E$ happens, or 0 otherwise. The complement of a set $A$ is denoted by $A^c$, and $|A|$ is the cardinality of the set $A$. $\mathrm{vec}\left(\cdot\right)$ denotes the vectorization of a matrix or a set of vectors,

and $\mathrm{tr}\,(\cdot)$ is the trace of a matrix. We denote the unit sphere in $d$-dimensional Euclidean space by $\mathbb{S}^{d-1} := \{\mathbf{x} : \mathbf{x} \in \mathbb{R}^d, \|\mathbf{x}\|_2 = 1\}$. Let $L^2(\mathbb{S}^{d-1}, \mu)$ denote the space of square-integrable functions on $\mathbb{S}^{d-1}$ with probability measure $\mu$, and the inner product $\langle \cdot, \cdot \rangle_\mu$ and $\|\cdot\|_\mu^2$ are defined as $\langle f, g \rangle_{L^2} := \int_{\mathbb{S}^{d-1}} f(x)g(x)\mathrm{d}\mu(x)$ and $\|f\|_{L^2}^2 := \int_{\mathbb{S}^{d-1}} f^2(x)\mathrm{d}\mu(x) < \infty$. $\mathbf{B}\,(\mathbf{x}; r)$ is the Euclidean closed ball centered at $\mathbf{x}$ with radius $r$. $\langle \cdot, \cdot \rangle_\mathcal{H}$ and $\|\cdot\|_\mathcal{H}$ denote the inner product and the norm in the Hilbert space $\mathcal{H}$. $a = \mathcal{O}(b)$ or $a \lesssim b$ indicates that there exists a constant $c > 0$ such that $a \leq cb$. $\tilde{\mathcal{O}}$ indicates there are specific requirements in the constants of the $\mathcal{O}$ notation. $a = o(b)$ and $a = w(b)$ indicate that $\lim |a/b| = 0$ and $\lim |a/b| = \infty$, respectively. $a \asymp b$ or $a = \Theta(b)$ denotes that there exist constants $c_1, c_2 > 0$ such that $c_1 b \leq a \leq c_2 b$. Throughout this paper we let the input space $\mathcal{X} = \mathbb{S}^{d-1}$, and $\mathrm{Unif}\,(\mathcal{X})$ denotes the uniform distribution on $\mathcal{X}$. Given a function $g : \mathbb{S}^{d-1} \to \mathbb{R}$, its $L^\infty$-norm is denoted by $\|g\|_\infty := \sup_{\mathbf{x} \in \mathbb{S}^{d-1}} |g(\mathbf{x})|$. $L^\infty$ is the function class whose elements have almost surely bounded $L^\infty$-norm. The constants defined throughout this paper may change from line to line. For a Reproducing Kernel Hilbert Space (RKHS) $\mathcal{H}$, $\mathcal{H}(\mu_0)$ denotes the ball centered at the origin with radius $\mu_0$ in $\mathcal{H}$. We use $\mathbb{E}_P\,[\cdot]$ to denote the expectation with respect to the distribution $P$.

## 2 PROBLEM SETUP

We introduce the problem setup for nonparametric regression using a neural network with channel attention in this section.

### 2.1 TWO-LAYER NEURAL NETWORK

We are given the training data $\left\{(\vec{\mathbf{x}}_i, y_i)\right\}_{i=1}^n$ where each data point is a tuple of feature vector $\vec{\mathbf{x}}_i \in \mathcal{X}$ and its response $y_i \in \mathbb{R}$. Throughout this paper we assume that no two training features coincide, that is, $\vec{\mathbf{x}}_i \neq \vec{\mathbf{x}}_j$ for all $i, j \in [n]$ and $i \neq j$. We denote the training feature vectors by $\mathbf{S} = \left\{\vec{\mathbf{x}}_i\right\}_{i=1}^n$, and denote by $P_n$ the empirical distribution over $\mathbf{S}$. All the responses are stacked as a vector $\mathbf{y} = [y_1, \ldots, y_n]^\top \in \mathbb{R}^n$. The response $y_i$ is given by $y_i = f^*(\vec{\mathbf{x}}_i) + w_i$ for $i \in [n]$, where $\{w_i\}_{i=1}^n$ are i.i.d. sub-Gaussian random noise with mean 0 and variance proxy $\sigma_0^2$, that is, $\mathbb{E}\,[\exp(\lambda w_i)] \leq \exp(\lambda^2 \sigma_0^2 / 2)$ for any $\lambda \in \mathbb{R}$. $f^*$ is the target function to be detailed later. We define $\mathbf{y} := [y_1, \ldots, y_n]$, $\mathbf{w} := [w_1, \ldots, w_n]^\top$, and use $f^*(\mathbf{S}) := \left[f^*(\vec{\mathbf{x}}_1), \ldots, f^*(\vec{\mathbf{x}}_n)\right]^\top$ to denote the clean target labels. The feature vectors in $\mathbf{S}$ are drawn i.i.d. according to the data distribution $P$ with $\mu$ being the probability measure for $P$. We note that $P$ can be an arbitrary continuous distribution supported on $\mathcal{X}$. We consider a two-layer neural network (NN) with channel attention in this paper whose mapping function is

$$f(\mathbf{W}, \boldsymbol{a}, \mathbf{x}) = \frac{1}{\sqrt{m}} \sum_{r'=1}^m \sum_{r=1}^m a_r \sigma\left(\vec{\mathbf{w}}_{r'}^\top \mathbf{x}\right) \mathbf{A}_{r'r}, \ \ \mathbf{A} = \boldsymbol{\sigma}(\mathbf{W}(0), \mathbf{Q})\boldsymbol{\sigma}(\mathbf{W}(0), \mathbf{Q})^\top / (Nm), \ \ (1)$$

where $\mathbf{x} \in \mathcal{X}$ is the input, $\sigma(\cdot) = \max\{\cdot, 0\}$ is the ReLU activation function, $\mathbf{W} = \left\{\vec{\mathbf{w}}_r\right\}_{r=1}^m$ with $\vec{\mathbf{w}}_r \in \mathbb{R}^d$ for $r \in [m]$ denotes the weighting vectors in the first layer and $m$ is the number of neurons. We define $\boldsymbol{\sigma}(\mathbf{W}(0), \mathbf{x}) \in \mathbb{R}^m$ with $[\boldsymbol{\sigma}(\mathbf{W}(0), \mathbf{x})]_r = \sigma\left(\vec{\mathbf{w}}_r(0)^\top \mathbf{x}\right)$ for $r \in [m]$ as the $m$ channels for the input $\mathbf{x}$. $\mathbf{A} \in \mathbb{R}^{m \times m}$ is the channel attention matrix, and $\boldsymbol{a} = [a_1, \ldots, a_m] \in \mathbb{R}^m$ denotes the weights of the second layer. We use $\mathbf{W}(0) = \left\{\vec{\mathbf{w}}_r(0)\right\}_{r=1}^m$ to denote the set of all the random weighting vectors at initialization, and $\vec{\mathbf{w}}_r(0) \sim \mathcal{N}(\mathbf{0}, \kappa^2 \mathbf{I}_d)$ for all $r \in [m]$, where $\mathcal{N}(\boldsymbol{\mu}, \boldsymbol{\Sigma})$ denotes a Gaussian distribution with mean $\boldsymbol{\mu}$ and covariance $\boldsymbol{\Sigma}$, and $\kappa = \Theta(1) \in (0, 1)$ controls the magnitude of initialization. $\mathbf{Q} = \left\{\vec{\mathbf{q}}_i\right\}_{i=1}^N$ is a sample of $N$ i.i.d. random variables distributed according to $P$, and $\mathbf{Q}$ is independent of $\mathbf{W}(0)$, and we also define $\boldsymbol{\sigma}(\mathbf{W}(0), \mathbf{Q}) \in \mathbb{R}^{m \times n}$ with its $i$-th column being $\boldsymbol{\sigma}(\mathbf{W}(0), \mathbf{Q})^{(i)} = \boldsymbol{\sigma}(\mathbf{W}(0), \vec{\mathbf{x}}_i)$ for all $i \in [N]$.

**XCA-style Channel Attention (Ali et al., 2021) in the Two-Layer NN (1) and Generation of the Sample Q.** We remark that the channel attention used in the two-layer NN (1) is a Cross-Covariance Attention (XCA)-style channel attention (Ali et al., 2021), where self-attention is applied to the channels instead of tokens. Similar to XCA, the channel attention mechanism in our two-layer NN (1) is a variant of self-attention where the attention matrix $\mathbf{A}$ contains the attention weights across $m$ channels when understanding channels as tokens. Furthermore, depending on whether $P$ is known or unknown, $\mathbf{Q}$ can be sampled according to $P$ accurately or approximately, and more details about the generation of $\mathbf{Q}$ are deferred to Section B.1 of the appendix.

## 2.2 Kernel and Target Function for Nonparametric Regression

We define the following kernel,

$$K(\mathbf{u}, \mathbf{v}) \coloneqq \frac{\mathbf{u}^\top \mathbf{v} \left(\pi - \arccos(\mathbf{u}^\top \mathbf{v})\right) + \sqrt{1 - (\mathbf{u}^\top \mathbf{v})^2}}{2\pi}, \quad \forall \mathbf{u}, \mathbf{v} \in \mathcal{X}, \tag{2}$$

which is in fact the NTK associated with the vanilla two-layer NN without channel attention,

$$f^{(\text{vanilla})}(\mathbf{W}, \boldsymbol{a}, \mathbf{x}) = \frac{1}{\sqrt{m}} \sum_{r=1}^{m} \sum_{r=1}^{m} a_r \sigma \left( \overrightarrow{\mathbf{w}}_r^\top \mathbf{x} \right), \tag{3}$$

where the second-layer weights $\boldsymbol{a}$ are initialized to $\mathbf{0}$. $K$ is a positive-definite (PD) kernel. Let the gram matrix of $K$ over the training features $\mathbf{S}$ be $\mathbf{K} \in \mathbb{R}^{n \times n}$, $\mathbf{K}_{ij} = K(\overrightarrow{\mathbf{x}}_i, \overrightarrow{\mathbf{x}}_j)$ for $i, j \in [n]$, and $\mathbf{K}_n \coloneqq \mathbf{K}/n$ is the empirical NTK matrix. Let the eigendecomposition of $\mathbf{K}_n$ be $\mathbf{K}_n = \mathbf{U}\boldsymbol{\Sigma}\mathbf{U}^\top$ where $\mathbf{U}$ is a $n \times n$ orthogonal matrix, and $\boldsymbol{\Sigma}$ is a diagonal matrix with its diagonal elements $\left\{ \widehat{\lambda}_i \right\}_{i=1}^{n}$ being the eigenvalues of $\mathbf{K}_n$ and sorted in a non-increasing order. It is proved by existing works, such as (Du et al., 2019b), that $\mathbf{K}_n$ is non-singular. Let $\mathcal{H}_K$ be the Reproducing Kernel Hilbert Space (RKHS) associated with $K$. Because $K$ is continuous on the compact set $\mathcal{X} \times \mathcal{X}$, the integral operator $T_K \colon L^2(\mathcal{X}, \mu) \to L^2(\mathcal{X}, \mu), (T_K f)(\mathbf{x}) \coloneqq \int_{\mathcal{X}} K(\mathbf{x}, \mathbf{x}')f(\mathbf{x}')\mathrm{d}\mu(\mathbf{x}')$ is a positive, self-adjoint, and compact operator on $L^2(\mathcal{X}, \mu)$. By the spectral theorem, there is a countable orthonormal basis $\{e_j\}_{j \geq 0} \subseteq L^2(\mathcal{X}, \mu)$ and $\{\lambda_j\}_{j \geq 0}$ with $1 > \lambda_0 \geq \lambda_1 \geq \ldots > 0$ such that $e_j$ is the eigenfunction of $T_K$ with $\lambda_j$ being the corresponding eigenvalue. That is, $T_K e_j = \lambda_j e_j, j \geq 0$. Let $\{\mu_\ell\}_{\ell \geq 0}$ be the distinct eigenvalues associated with $T_K$, and let $m_\ell$ be the sum of multiplicity of the eigenvalues $\{\mu_{\ell'}\}_{\ell'=0}^{\ell}$. That is, $m_{\ell'} - m_{\ell'-1}$ is the multiplicity of $\mu_{\ell'}$ with $m_{-1} = 0$. It is well known that $\left\{ v_j = \sqrt{\lambda_j} e_j \right\}_{j \geq 1}$ is an orthonormal basis of $\mathcal{H}_K$. For a positive constant $\mu_0$, we define $\mathcal{H}_K(\mu_0) \coloneqq \left\{ f \in \mathcal{H}_K \colon \|f\|_{\mathcal{H}} \leq \mu_0 \right\}$ as the closed ball in $\mathcal{H}_K$ centered at 0 with radius $\mu_0$. We note that $\mathcal{H}_K(\mu_0)$ is also specified by $\mathcal{H}_K(\mu_0) = \left\{ f \in L^2(\mathcal{X}, \mu) \colon f = \sum_{j=1}^{\infty} \beta_j e_j, \sum_{j=1}^{\infty} \beta_j^2/\lambda_j \leq \mu_0^2 \right\}$.

**Target Function in an Interpolation Space with Spectral Bias.** Extensive theoretical and empirical studies find that it is easy for neural networks to learn spectrally biased target functions or low-frequency information in the training data (Rahaman et al., 2019; Arora et al., 2019; Cao et al., 2021; Choraria et al., 2022; Wang et al., 2024; 2025). For example, the studies in (Arora et al., 2019; Cao et al., 2021) reveal that it is easier for over-parameterized neural networks to learn target functions with spectral bias, such as polynomials of low-degree with spherical uniform data distribution on $\mathcal{X}$, or the low-rank part of the ground truth training class labels, or simple patterns of low-frequency. This observation motivates us to restrict the target function $f^*$ to a smaller class than $\mathcal{H}_K(\mu_0)$, which is $\mathcal{H}_{K^{(\text{attn})}}(\mu_0)$ to be studied in this paper.

We then define the attention kernel $K^{(\text{attn})}$, and explain why functions in $\mathcal{H}_{K^{(\text{attn})}}(\mu_0)$ has stronger spectral bias than that in $\mathcal{H}_K(\mu_0)$. $K^{(\text{attn})}$ and its empirical version, $\widehat{K}^{(\text{attn})}$, are defined as

$$K^{(\text{attn})}(\mathbf{x}, \mathbf{x}') \coloneqq \int_{\mathcal{X} \times \mathcal{X}} K(\mathbf{x}, \mathbf{v})K(\mathbf{v}, \mathbf{v}')K(\mathbf{v}', \mathbf{x}')\mathrm{d}\mu(\mathbf{v}) \otimes \mu(\mathbf{v}'), \tag{4}$$

$$\widehat{K}^{(\text{attn})}(\mathbf{x}, \mathbf{x}') \coloneqq \frac{1}{N^2} \sum_{i,j=1}^{N} K(\mathbf{x}, \overrightarrow{\mathbf{q}}_i)K(\overrightarrow{\mathbf{q}}_i, \overrightarrow{\mathbf{q}}_j)K(\overrightarrow{\mathbf{q}}_j, \mathbf{x}'). \tag{5}$$

The same sample $\mathbf{Q}$ as that in the channel attention matrix $\mathbf{A}$ is used in (5). Theorem D.1 in Appendix D.1 shows that the integral operator associated with $K^{(\text{attn})}$, $T_{K^{(\text{attn})}}$, has the same eigenfunctions $\{e_j\}_{j \geq 1}$ as $T_K$, and the eigenvalue corresponding to $e_j$ is $\lambda_j^{(\text{attn})} = \lambda_j^3 \in (0,1)$ for all $j \geq 1$. Because $K^{(\text{attn})}$ is still a PD kernel, the RKHS associated with $K^{(\text{attn})}$ is well-defined, and $\mathcal{H}_{K^{(\text{attn})}}(\mu_0)$ indicates a subset of $\mathcal{H}_{K^{(\text{attn})}}$ with the RKHS-norm $\|\cdot\|_{K^{(\text{attn})}}$ bounded by $\mu_0$. It can be verified that $\mathcal{H}_{K^{(\text{attn})}}(\mu_0) = \{f \in L^2(\mathcal{X}, \mu) \colon f = \sum_{j=1}^{\infty} \beta_j e_j, \sum_{j=1}^{\infty} \beta_j^2 / \lambda_j^3 \leq \mu_0^2\}$. As $\lambda_j \to 0$ with $j \to \infty$ and $\lambda_j^3 < \lambda_j < 1$, which follow from the spectral theorem, we have $\mathcal{H}_{K^{(\text{attn})}}(\mu_0) \subseteq \mathcal{H}_K(\mu_0)$. Compared to a function in $\mathcal{H}_K(\mu_0)$, a function in $\mathcal{H}_{K^{(\text{attn})}}(\mu_0)$ admits expansion coefficients $\{\beta_j\}_{j \geq 1}$ that concentrate more on the leading eigenfunctions with smaller index $j$. In this sense, we say that the target function $f^* \in \mathcal{H}_{K^{(\text{attn})}}(\mu_0)$ has stronger spectral bias than one in $\mathcal{H}_K(\mu_0)$. Such spectrally biased targets have been widely studied in the deep learning literature. For example, (Ghorbani et al., 2021; Bai & Lee, 2020; Cao et al., 2021) investigate learning low-degree polynomials of degree $\ell \geq 0$ via linearization or higher-order approximations to neural networks. These polynomials can be expressed as finite linear combinations of the leading eigenfunctions $\{e_j\}_{j=1}^{m_{\ell+1}}$. With properly chosen $\mu_0$ and coefficients $\{\beta_j\}_{j=1}^{m_{\ell+1}}$, we have $\sum_{j=1}^{m_{\ell+1}} \beta_j e_j \in \mathcal{H}_{K^{(\text{attn})}}(\mu_0)$. Moreover, $\mathcal{H}_{K^{(\text{attn})}}(\mu_0)$ coincides with the interpolation space $[\mathcal{H}_K]^s$ (with $s = 3$) of the RKHS, which characterizes the regularity of regression functions studied in (Steinwart & Scovel, 2012; Fischer & Steinwart, 2020). The general interpolation space for $s' > 0$ is defined as $[\mathcal{H}_K]^{s'}(\mu_0) := \left\{ \sum_{j \geq 1} a_j \lambda_j^{s'/2} e_j \colon \sum_{j \geq 1} a_j^2 \leq \mu_0 \right\}$. In this work we focus on $s' = 3$, and we consider $f^* \in [\mathcal{H}_K]^3(\mu_0)$. It can be verified that $\mathcal{H}_{K^{(\text{attn})}}(\mu_0) = [\mathcal{H}_K]^3(\mu_0)$. In summary, *with $f^* \in \mathcal{H}_{K^{(\text{attn})}}(\mu_0) = [\mathcal{H}_K]^3(\mu_0) \subseteq \mathcal{H}_K(\mu_0)$, we say that $f^*$ lies in an interpolation space with spectral bias.*

**The Task of Nonparametric Regression.** The task of nonparametric regression studied in this paper is to find an estimator $\widehat{f}$ from the training data $\left\{ (\vec{\mathbf{x}}_i, y_i) \right\}_{i=1}^{n}$ so that the risk $\mathbb{E}_P\left[ \left( \widehat{f} - f^* \right)^2 \right]$ can converge to 0 with a fast rate, with $f^* \in [\mathcal{H}_K]^3(\mu_0) = \mathcal{H}_{K^{(\text{attn})}}(\mu_0)$. The over-parameterized NN (1) trained from the training data serves as the estimator $\widehat{f}$. The statistical learning literature has established rich results in the sharp convergence rates for the risk of nonparametric kernel regression (Stone, 1985; Yang & Barron, 1999; Raskutti et al., 2014; Yuan & Zhou, 2016). The representative result in (Raskutti et al., 2014) about kernel regression shows that $\mathbb{E}_P\left[ \left( \widehat{f} - f^* \right)^2 \right] \lesssim \varepsilon_{K,n}^2$, where $\varepsilon_{K,n}$ is the critical population rate of the PD kernel $K$ used for kernel regression, also referred to as the critical radius (Wainwright, 2019) of $K$, and $\widehat{f}$ is obtained through kernel regression trained by regular GD with early stopping. The risk bound $\varepsilon_{K,n}^2$ is sharp, since it is minimax optimal in several popular learning setups, such as the setup where the eigenvalues $\{\lambda_i\}_{i \geq 1}$ of $T_K$ exhibit a certain polynomial decay rate when $f^* \in \mathcal{H}_K(\mu_0)$. We will show in the next section that the two-layer NN with channel attention (1) trained by GD renders sharper and minimax optimal risk bounds compared to the current state-of-the-art, when $f^* \in \mathcal{H}_{K^{(\text{attn})}}(\mu_0) = [\mathcal{H}_K]^3(\mu_0)$.

## 3 MAIN RESULTS

All results and discussions in this paper are presented under the fixed-dimension setting with $d \geq 5$, a widely adopted setting in prior works (Hu et al., 2021; Suh et al., 2022; Li et al., 2024; Yang & Li, 2024; Yang, 2025). We present the first main result, Theorem 3.1, as follows, for arbitrary continuous distribution $P$ supported on $\mathcal{X}$.

**Theorem 3.1.** Suppose $P$ is an arbitrary continuous distribution on $\mathcal{X}$, $\delta \in (0,1)$,

$$m \gtrsim \max\left\{ (d^2 + \log^2 m)/\varepsilon_n^{16}, (d \log m)^3/\varepsilon_n^8, n \right\}, \quad N \gtrsim \log(n/\delta)/\varepsilon_n^8, \tag{6}$$

and the neural network $f_t = f(\mathbf{W}(t), \boldsymbol{a}(t), \boldsymbol{a}(t), \cdot)$ is trained by GD in Algorithm 1 with the learning rate $\eta = \Theta(1) \in (0,1)$ with $T \leq \widehat{T}$. Then for every $t \in [c_t T : T]$, with probability at least $1 - 2/n - \delta - \exp(-\Theta(n)) - 7\exp(-\Theta(n\varepsilon_n^2))$ over $\mathbf{w}, \mathbf{S}, \mathbf{Q}, \mathbf{W}(0)$, the stopping time satisfies

$\widehat{T} \asymp \varepsilon_n^{-2}$, and

$$\mathbb{E}_P \left[ (f_t - f^*)^2 \right] \lesssim \varepsilon_n^2. \tag{7}$$

**Sharper Risk Bound by Theorem 3.1 .** We emphasize that Theorem 3.1 renders sharper regression risk bound than the current state-of-the-art, (Yang, 2025, Theorem 5.1), and both works consider arbitrary continuous distribution $P$ of the covariate. In particular, (Yang, 2025, Theorem 5.1) shows a sharp regression risk bound of $\mathcal{O}(\varepsilon_{K,n}^2)$ when training the vanilla network $f^{(\text{vanilla})}$ (3) without channel attention by GD, when $f \in \mathcal{H}_K(\mu_0)$. In the same distribution-free manner in the covariate, Theorem 3.1 shows the clear theoretical benefit of the XCA-style channel attention (Ali et al., 2021) on the vanilla network . That is, when the target function $f^*$ lies in the interpolation space $[\mathcal{H}_K]^3(\mu_0) \subseteq \mathcal{H}_K(\mu_0)$ with spectral bias, a sharper regression risk bound of $\mathcal{O}(\varepsilon_n^2)$ is achieved when training the network (1) by GD. This is because $\varepsilon_n^2 \leq \varepsilon_{K,n}^2$ according to Propotition C.2 in the appendix. The sharper risk bound by Theorem 3.1 is instantiated for certain distribution $P$ in Theorem 3.2 below.

Applying Theorem 3.1 to the case where the distribution $P$ admits a polynomial EDR for the kernel $K$ (2) such that $\lambda_j \asymp j^{-2\alpha}$ for $\alpha > 1/2$ and all $j \geq 1$, we have the following theorem as a direct consequence of Theorem 3.1. Such polynomial EDR holds when $P$ is the uniform distribution on $\mathcal{X}$, with $2\alpha = d/(d-1)$, which is shown by existing works such as (Hu et al., 2021, Lemma 3.1). The proofs of Theorem 3.1 and Theorem 3.2 are deferred to Section C.3 of the appendix.

**Theorem 3.2.** Suppose the distribution $P$ admits a polynomial EDR of $\lambda_j \asymp j^{-2\alpha}$ for $\alpha > 1/2$ and all $j \geq 1$, $\delta \in (0,1)$,

$$m \gtrsim n^{48\alpha/(6\alpha+1)} d^3 \log^3 m, \quad N \gtrsim n^{\frac{24\alpha}{6\alpha+1}} \log(n/\delta), \tag{8}$$

and the neural network $f_t = f(\mathbf{W}(t), \boldsymbol{a}(t), \cdot)$ is trained by GD in Algorithm 1 with the learning rate $\eta = \Theta(1) \in (0,1)$ with $T \leq \widehat{T}$. Then for every $t \in [c_t T \colon T]$, with probability at least $1 - 2/n - \delta - \exp\left(-\Theta(n)\right) - 7\exp\left(-\Theta(n^{\frac{1}{6\alpha+1}})\right)$ over $\mathbf{w}, \mathbf{S}, \mathbf{Q}, \mathbf{W}(0)$, the stopping time satisfies $\widehat{T} \asymp n^{\frac{6\alpha}{6\alpha+1}}$, and

$$\mathbb{E}_P \left[ (f_t - f^*)^2 \right] \lesssim n^{-\frac{6\alpha}{6\alpha+1}}. \tag{9}$$

Table 1: Comparisons with existing works on the regression risk bounds and assumptions for non-parametric regression using over-parameterized neural networks with algorithmic guarantees. The results listed are under a widely studied setup where $f^* \in \mathcal{H}_{\tilde{K}}$ and the responses $\{y_i\}_{i=1}^n$ are corrupted by i.i.d. Gaussian or sub-Gaussian noise. Here $P$ denotes the distribution of the training features, and $\tilde{K}$ represents the kernel induced by the neural architecture and optimization method of each particular work. For all prior works, $\tilde{K}$ corresponds to the regular NTK, while in this work we instead have $\tilde{K} = K^{(\text{attn})}$. Both our work and (Li et al., 2024) consider target functions satisfying $f^* \in [\mathcal{H}_K]^3$ under the polynomial EDR $\lambda_j \asymp j^{-2\alpha}$, and $2\alpha = d/(d-1)$ in (Li et al., 2024; Hu et al., 2021; Suh et al., 2022). Moreover, (Li et al., 2024, Proposition 13) can be adapted to our setting with no bias/intercept learned in the first layer, leading to the polynomial EDR of $\lambda_j \asymp j^{-\frac{d}{d-1}}$ rather than $\lambda_j \asymp j^{-\frac{d+1}{d}}$.

| Existing Works and Our Result | Distributional Assumptions | Eigenvalue Decay Rate (EDR) | Rate of Nonparametric Regression Risk |
|---|---|---|---|
| (Kuzborskij & Szepesvári, 2021, Theorem 2) | No | – | $\sigma^2 + \mathcal{O}(n^{\frac{-2}{2+d}})$ |
| (Hu et al., 2021, Theorem 5.2), (Suh et al., 2022, Theorem 3.11) | $P$ is Unif $(\mathcal{X})$ | $\lambda_j \asymp j^{-\frac{d}{d-1}}$ | $\mathcal{O}(n^{-\frac{2d}{2d-1}}) = \mathcal{O}(n^{-\frac{d}{2d-1}})$ |
| (Li et al., 2024, Proposition 13) | $P$ is sub-Gaussian | $\lambda_j \asymp j^{-\frac{d}{d-1}}$ | $\mathcal{O}(n^{-\frac{6\alpha}{6\alpha+1}}) \log^2(1/\delta)$ |
| (Yang, 2025, Theorem 5.1) | Arbitrary continuous distribution on $\mathcal{X}$ | No requirement for EDR | $\mathcal{O}(\varepsilon_{K,n}^2)$ |
| (Yang, 2025, Corollary 5.2) | $P$ admits the polynomial EDR $\lambda_j \asymp j^{-2\alpha}$ | $\lambda_j \asymp j^{-\frac{d}{d-1}}$ | $\mathcal{O}(n^{-\frac{2\alpha}{2\alpha+1}})$ |
| Our Result (Theorem 3.1) | Arbitrary continuous distribution on $\mathcal{X}$ | No requirement for EDR | $\mathcal{O}(\varepsilon_n^2)$. |
| Our Result (Theorem 3.2) | $P$ admits the polynomial EDR $\lambda_j \asymp j^{-2\alpha}$ | $\lambda_j^{(\text{attn})} \asymp j^{-\frac{3d}{d-1}}$ | $\mathcal{O}(n^{-\frac{6\alpha}{6\alpha+1}}) = \mathcal{O}(n^{-\frac{3d}{4d-1}})$. |

**Minimax Optimality of the Risk Bound by Theorem 3.2.** While the rate of $\mathcal{O}(n^{-\frac{d}{2d-1}})$ in Hu et al. (2021); Suh et al. (2022); Yang & Li (2024) remains minimax optimal in the context of kernel regression with the regular NTK $\tilde{K}$ when $f^* \in \mathcal{H}_{\tilde{K}}(\mu_0)$, a faster rate is achievable if the target function lies in the interpolation space $[\mathcal{H}_K]^3(\mu_0)$, a subset of $\mathcal{H}_K(\mu_0)$. In particular, if $f^* \in$

$\mathcal{H}_{K^{(\mathrm{attn})}}(\mu_0) = [\mathcal{H}_K]^{s'}(\mu_0) \subseteq \mathcal{H}_K(\mu_0)$, then kernel regression using the attention kernel $K^{(\mathrm{attn})}$ attains the sharper rate $\mathcal{O}(n^{-\frac{6\alpha}{6\alpha+1}})$, which, as in Theorem 3.2, is minimax optimal in the sense of regression by the attention kernel $K^{(\mathrm{attn})}$ over the space $\mathcal{H}_{K^{(\mathrm{attn})}}(\mu_0)$ (Stone, 1985; Yang & Barron, 1999; Yuan & Zhou, 2016). It is remarked that our risk bound (9) is sharper than the nearly-optimal one in (Li et al., 2024, Proposition 13), $\mathcal{O}(n^{-\frac{6\alpha}{6\alpha+1}})\log^2(1/\delta)$. Theorem D.1 shows that $K^{(\mathrm{attn})}$ has the EDR of $\lambda_j^{(\mathrm{attn})} = \lambda_j^3 \asymp j^{-6\alpha}$ for all $j \geq 1$. Accordingly, the associated fixed point of the kernel complexity function associated with the attention kernel $K^{(\mathrm{attn})}$ is $\mathcal{O}(\varepsilon_n^2) = \mathcal{O}(n^{-\frac{6\alpha}{6\alpha+1}})$, yielding the claimed sharper risk bound in (9).

We also provide more detailed comparison to existing works in Table 1, where broader relevant works on nonparametric regression with target functions lying in an interpolation space or a general RKHS are included.

## 4 TRAINING BY GRADIENT DESCENT

In the training process of our two-layer NN (1), only $\mathbf{W}$ is optimized with $\boldsymbol{a}$ randomly initialized to $\pm 1$ with equal probabilities and then fixed. The following quadratic loss function is minimized during the training process:

$$L(\mathbf{W}) \coloneqq \frac{1}{2n} \sum_{i=1}^{n} \left( f(\mathbf{W}, \boldsymbol{a}, \vec{\mathbf{x}}_i) - y_i \right)^2. \tag{10}$$

---

**Algorithm 1** Training the Two-Layer NN with Channel Attention (1) by GD

1: $\mathbf{W}(T) \leftarrow$ Training-by-GD$(T, \mathbf{W}(0))$
2: **input:** $T, \mathbf{W}(0), \eta$
3: **for** $t = 1, \ldots, T$ **do**
4:     Perform the $t$-th step of GD by (11)-(12)
5: **end for**
6: **return** $\mathbf{W}(T)$

---

In the $(t + 1)$-th step of GD with $t \geq 0$, the weights of the neural network, $\mathbf{W}$ and $\boldsymbol{a}$, are updated by one-step of GD through

$$\mathrm{vec}\left(\mathbf{W}_{\mathbf{S}}(t+1)\right) = \mathrm{vec}\left(\mathbf{W}_{\mathbf{S}}(t)\right) - \frac{\eta}{n}\mathbf{Z}_{\mathbf{S}}(t)(\widehat{\mathbf{y}}(t) - \mathbf{y}), \tag{11}$$

$$\boldsymbol{a}(t+1) = \boldsymbol{a}(t) - \frac{\eta}{n\sqrt{m}}\mathbf{A}\boldsymbol{\sigma}(\mathbf{W}(t), \mathbf{S})(\widehat{\mathbf{y}}(t) - \mathbf{y}), \tag{12}$$

where $\mathbf{y}_i = y_i$, $\widehat{\mathbf{y}}(t) \in \mathbb{R}^n$ with $[\widehat{\mathbf{y}}(t)]_i = f(\mathbf{W}(t), \boldsymbol{a}(t), \vec{\mathbf{x}}_i)$. The notations with the subscript $\mathbf{S}$ indicate the dependence on the training features $\mathbf{S}$. We also denote $f(\mathbf{W}(t), \boldsymbol{a}(t), \cdot)$ as $f_t(\cdot)$ which is the neural network function with weights $\mathbf{W}(t)$ and $\boldsymbol{a}(t)$ obtained right after the $t$-th step of GD. We define $\mathbf{Z}_{\mathbf{S}}(t) \in \mathbb{R}^{md \times n}$ which is computed by $[\mathbf{Z}_{\mathbf{S}}(t)]_{[(r-1)d+1:rd]i} = \frac{1}{\sqrt{m}}\mathbb{I}_{\left\{\vec{\mathbf{w}}_r(t)^\top \vec{\mathbf{x}}_i \geq 0\right\}}\vec{\mathbf{x}}_i [\mathbf{A}\boldsymbol{a}(t)]_r$ for all $i \in [n], r \in [m]$, where $[\mathbf{Z}_{\mathbf{S}}(t)]_{[(r-1)d+1:rd]i} \in \mathbb{R}^d$ is a vector with elements in the $i$-th column of $\mathbf{Z}_{\mathbf{S}}(t)$ with indices in $[(r-1)d+1:rd]$. We have $\boldsymbol{a} = \mathbf{0}$ at the initialization, so that $\widehat{\mathbf{y}}(0) = \mathbf{0}$. We run Algorithm 1 to train the two-layer NN by GD, where $T$ is the total number of steps for GD. Early stopping is enforced in Algorithm 1 through a bounded $T$ via $T \leq \widehat{T}$.

## 5 ROADMAP OF PROOFS

We present the roadmap of our theoretical results which lead to the main results, Theorem 3.1 and Theorem 3.2. We first introduce kernel complexity in Section 5.1, a key concept in our results and their proofs. Section 5.2 details the roadmap, key technical results in the proofs, our novel proof strategies and insights from our theoretical results.

### 5.1 KERNEL COMPLEXITY

The local kernel complexity has been studied by (Bartlett et al., 2005; Koltchinskii, 2006; Mendelson, 2002). For the PD kernel $K$, we define the empirical kernel complexity $\widehat{R}_K$ and the population kernel complexity $R_K$ as

$$\widehat{R}_K(\varepsilon) \coloneqq \sqrt{\frac{1}{n}\sum_{i=1}^{n}\min\left\{\widehat{\lambda}_i, \varepsilon^2\right\}}, \quad R_K(\varepsilon) \coloneqq \sqrt{\frac{1}{n}\sum_{i=1}^{\infty}\min\left\{\lambda_i, \varepsilon^2\right\}}. \tag{13}$$

It can be verified that both $\sigma_0 R_K(\varepsilon)$ and $\sigma_0 \widehat{R}_K(\varepsilon)$ are sub-root functions (Bartlett et al., 2005) in terms of $\varepsilon^2$. Sub-root functions are defined in Definition A.2. For a given noise ratio $\sigma_0$, the critical empirical radius $\widehat{\varepsilon}_{K,n} > 0$ is the smallest positive solution to the inequality $\widehat{R}_K(\varepsilon) \le \varepsilon^2/\sigma_0$, where $\widehat{\varepsilon}_{K,n}^2$ is the also the fixed point of $\sigma_0 \widehat{R}_K(\varepsilon)$ as a function of $\varepsilon^2$: $\sigma_0 \widehat{R}_K(\widehat{\varepsilon}_{K,n}) = \widehat{\varepsilon}_{K,n}^2$. Similarly, the critical population rate $\varepsilon_{K,n}$ is defined to be the smallest positive solution to the inequality $R_K(\varepsilon) \le \varepsilon^2/\sigma_0$, where $\varepsilon_{K,n}^2$ is the fixed point of $\sigma_0 \widehat{R}_K(\varepsilon)$ as a function of $\varepsilon^2$: $\sigma_0 R_K(\varepsilon_{K,n}) = \varepsilon_{K,n}^2$. Kernel complexity can also be defined for the attention kernel $K^{(\mathrm{attn})}$, leading to the empirical kernel complexity $\widehat{R}_{K^{(\mathrm{attn})}}$ and the population kernel complexity $R_{K^{(\mathrm{attn})}}$ for $K^{(\mathrm{attn})}$, with the critical empirical radius $\widehat{\varepsilon}_{K^{(\mathrm{attn})},n}$ and the critical population rate $\varepsilon_{K^{(\mathrm{attn})},n}$, respectively. For simplicity of the notations, we use $\varepsilon_n$ and $\widehat{\varepsilon}_n$ to denote $\varepsilon_{K^{(\mathrm{attn})},n}$ and $\widehat{\varepsilon}_{K^{(\mathrm{attn})},n}$, respectively. In this paper we consider the kernel $K$ such that $\min\{\varepsilon_{K,n}, \varepsilon_n\} \cdot n \to \infty$ as $n \to \infty$, which covers most popular positive semi-definite kernels including the kernel (2) and a broad range of data distributions (Yang et al., 2017). Let $\eta_t := \eta t$ for all $t \ge 0$, we then define the stopping time $\widehat{T}$ as

$$\widehat{T} := \min\left\{T \colon \widehat{R}_{K^{(\mathrm{attn})}}(\sqrt{1/\eta_t}) > (\sigma_0 \eta_t)^{-1}\right\} - 1. \tag{14}$$

The stopping time in fact limits the number of steps $T$ for Algorithm 1, which enforces the early stopping mechanism. In fact, as will be shown later in this section, we need to have $T \le \widehat{T}$ when training the two-layer NN (1) by GD with Algorithm 1.

## 5.2 Detailed Roadmap, Key Results, and Novel Proof Strategies and Insights

The detailed roadmap, summary of the key technical results, and our novel proof strategies which lead to Theorem 3.1 are presented as follows. Theorem 3.2 follows from Theorem 3.1 by applying Theorem 3.1 to the case of the polynomial EDR of $\lambda_j \asymp j^{-2\alpha}$ for all $j \ge 1$.

**Roadmap and Key Technical Results.** First, uniform convergence of the empirical attention kernel, $\widehat{K}^{(\mathrm{attn})}$, to the attention kernel, $K^{(\mathrm{attn})}$, is established during training the two-layer NN with channel attention (1) by GD.

**Theorem 5.1.** For every fixed $\mathbf{x}' \in \mathcal{X}$ and every $\delta \in (0,1)$, with probability at least $1 - \delta$ over the random sample $\mathbf{Q}$, we have $\left\|\widehat{K}^{(\mathrm{attn})}(\mathbf{x}, \mathbf{x}') - K^{(\mathrm{attn})}(\mathbf{x}, \mathbf{x}')\right\|_\infty \lesssim \sqrt{\frac{\log 1/\delta}{N}}$.

Based on Theorem 5.1, we establish a novel decomposition of the neural network function at any GD step into a function within the RKHS associated with the attention kernel $K^{(\mathrm{attn})}$ and an error function small $L^\infty$-norm with high probability, as stated in Theorem 5.2.

**Theorem 5.2.** Suppose $\delta \in (0,1)$, $w \in (0,1)$, $m, N$ are sufficiently large and finite, and the neural network $f_t = f(\mathbf{W}(t), \boldsymbol{a}(t), \cdot)$ is trained by GD using Algorithm 1 with the learning rate $\eta = \Theta(1) \in (0,1)$. Then for every $t \in [T]$ with $T \le \widehat{T}$, with high probability over the random initialization $\mathbf{W}(0)$, the random noise $\mathbf{w}$, and the random sample $\mathbf{Q}$, $f_t$ has the following decomposition on $\mathcal{X}$:

$$f_t = h_t + e_t, \tag{15}$$

where $h_t \in \mathcal{H}_K(B_h)$ with $B_h$ defined in (29), $e_t \in L^\infty$ with $\|e_t\|_\infty \le w$. The lower bounds for $m, N$ depends on $w$, and smaller $w$ leads to larger lower bounds for $m, N$.

Second, leveraging the decomposition in Theorem 5.2, we introduce a new technique based on local Rademacher complexity to obtain a tight bound on the Rademacher complexity of the function class formed by all neural network functions generated through GD iterations. This development leads directly to the sharp regression risk bound presented in Theorem 5.3 below.

**Theorem 5.3.** Suppose $\delta \in (0,1)$, $w \in (0,1)$, $m, N$ are sufficiently large and finite, and the neural network $f_t = f(\mathbf{W}(t), \boldsymbol{a}(t), \cdot)$ is trained by GD using Algorithm 1 with the learning rate $\eta = \Theta(1) \in (0,1)$, and $T \le \widehat{T}$. Then for every $t \in [T]$ and every $\delta \in (0,1)$, with high probability over the random initialization $\mathbf{W}(0)$, the random noise $\mathbf{w}$, the random training features $\mathbf{S}$, and the random sample $\mathbf{Q}$, $\mathbb{E}_P\left[(f_t - f^*)^2\right] - 2\mathbb{E}_{P_n}\left[(f_t - f^*)^2\right] \lesssim \varepsilon_n^2 + w$.

We then obtain Theorem 3.1 using Theorem 5.3 where $w$ is set to $\varepsilon_n^2$, with the empirical loss $\mathbb{E}_{P_n}\left[(f_t - f^*)^2\right]$ bounded by $\Theta(1/(\eta t)) \asymp \varepsilon_n^2$ with high probability by Theorem C.7 deferred to Section C.4 of the appendix.

**Novel Proof Strategies and Insights for the Benefit of Channel Attention.** Our results rely on two substantially novel proof strategies. First, the uniform convergence of $\widehat{K}^{(\mathrm{attn})}$ to $K^{(\mathrm{attn})}$ during the training process of the network with channel attention (1), established in Theorem 5.1, enables a new decomposition of the neural network function at any step of GD into a function in $\mathcal{H}_{K^{(\mathrm{attn})}} = [\mathcal{H}_K]^3$ and an error function with small $L^\infty$-norm with high probability in Theorem 5.2. The uniform convergence of $\widehat{K}^{(\mathrm{attn})}$ to $K^{(\mathrm{attn})}$ in Theorem 5.1 is proved by employing the martingale-based concentration inequality for Banach space-valued processes (Pinelis, 1992, Theorem 2). Second, leveraging the decomposition in Theorem 5.2, we introduce a new technique based on local Rademacher complexity, which tightly bounds the Rademacher complexity of the function class consisting of all neural network functions generated by GD iterations. This leads to the sharp regression risk bound in Theorem 5.3.

To the best of our knowledge, our results reveal the theoretical benefit of the XCA-style channel attention (Ali et al., 2021) mechanism used in our two-layer NN (1) for nonparametric regression in an interpolation space. In particular, with sufficiently large network width $m$ in the over-parameterized regime, the network (1) trained by GD approximately performs kernel regression with the new attention kernel $K^{(\mathrm{attn})}$. In a strong contrast, over-parameterized neural networks trained by GD, widely studied in existing works on such as (Suh et al., 2022; Li et al., 2024; Yang & Li, 2024; Yang, 2025), induces the standard NTK of the form such as (2). As elaborated in "Sharper Regression Risk Bound by Theorem 3.1" in Section 3, channel attention yields a sharper regression risk bound $\mathcal{O}(\varepsilon_n^2)$ for learning the target function in the interpolation space $[\mathcal{H}_K]^3$, compared to the risk bound $\mathcal{O}(\varepsilon_{K,n}^2)$ rendered by the vanilla network $f^{(\mathrm{vanilla})}$ (3) without such channel attention. The fundamental reason for the sharper bound is that, the network with channel attention (1) induces the attention kernel $K^{(\mathrm{attn})}$, whose kernel complexity $R_{K^{(\mathrm{attn})}}$ is lower than the kernel complexity of the standard NTK (2), which is induced by the vanilla network $f^{(\mathrm{vanilla})}$.

**Beyond the Regular NTK Limit.** We remark that our result is beyond the NTK limit or the linear region of the regular NTK (2), since the function represented by the two-layer NN trained with our novel GD is arbitrarily close to some $h_t \in \mathcal{H}_{K^{(\mathrm{attn})}}(B_h)$, where $\mathcal{H}_{K^{(\mathrm{attn})}}$ is an RKHS distinct from $\mathcal{H}_K$ associated with the regular NTK (2). In particular, training the counterpart network without channel attention, $f^{(\mathrm{vanilla})}$ (3), cannot achieve our sharp risk bound. Furthermore, it is technically nontrivial to induce the new kernel $K^{(\mathrm{attn})}$ when training with the proposed GD algorithm, as detailed through our proof strategies described above. Our results are significantly from the existing kernel learning literature and they lead to a better lower bound on the network width $m$ compared to the existing literature, which are detailed in Section B.3 and Section B.2 of the appendix.

## 6 SIMULATION STUDY

We present simulation results in Section E of the appendix on both synthetic and real data, including mini-ImageNet (Vinyals et al., 2016), which demonstrate the advantage of our two-layer NN with channel attention (1) over the vanilla network $f^{(\mathrm{vanilla})}$ (3) without such attention mechanism.

## 7 CONCLUSION

We study nonparametric regression by training an over-parameterized two-layer neural network with channel attention where the target function lies in an interpolation space with spectral bias. We show that, if the neural network is trained by GD with early stopping, a sharp rate of the order $\mathcal{O}(\varepsilon_n^2)$ can be obtained for arbitrary continuous covariate distribution on the unit sphere in $\mathbb{R}^d$, and such rate is minimax optimal for the case that the covariate distribution admits a polynomial eigenvalue decay rate. Novel proof strategies are employed to achieve our results, complemented by comparisons with the current state-of-the-art and supporting simulation studies.

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

# A  MATHEMATICAL TOOLS

The appendix of this paper is organized as follows. We present the basic mathematical results employed in our proofs in Section A, and then present the detailed proofs in Section C. More results about the attention kernel are presented in Section D.1, and simulation results are presented in Section E.

## A.1  CONCENTRATION INEQUALITIES FOR SUPREMUM OF EMPIRICAL PROCESSES

The Rademacher complexity of a function class and its empirical version are defined below.

*Definition* A.1. Let $\boldsymbol{\sigma} = \{\sigma_i\}_{i=1}^n$ be $n$ i.i.d. random variables such that $\Pr[\sigma_i = 1] = \Pr[\sigma_i = -1] = \frac{1}{2}$. The Rademacher complexity of a function class $\mathcal{F}$ is defined as

$$\mathfrak{R}(\mathcal{F}) = \mathbb{E}_{\left\{\vec{\mathbf{x}}_i\right\}_{i=1}^n, \{\sigma_i\}_{i=1}^n} \left[ \sup_{f \in \mathcal{F}} \frac{1}{n} \sum_{i=1}^n \sigma_i f(\vec{\mathbf{x}}_i) \right]. \tag{16}$$

The empirical Rademacher complexity is defined as

$$\widehat{\mathfrak{R}}(\mathcal{F}) = \mathbb{E}_{\{\sigma_i\}_{i=1}^n} \left[ \sup_{f \in \mathcal{F}} \frac{1}{n} \sum_{i=1}^n \sigma_i f(\vec{\mathbf{x}}_i) \right], \tag{17}$$

For simplicity of notations, Rademacher complexity and empirical Rademacher complexity are also denoted by $\mathbb{E}\left[\sup_{f \in \mathcal{F}} \frac{1}{n} \sum_{i=1}^{n} \sigma_i f(\vec{\mathbf{x}}_i)\right]$ and $\mathbb{E}_{\boldsymbol{\sigma}}\left[\sup_{f \in \mathcal{F}} \frac{1}{n} \sum_{i=1}^{n} \sigma_i f(\vec{\mathbf{x}}_i)\right]$ respectively.

For data $\left\{\vec{\mathbf{x}}\right\}_{i=1}^{n}$ and a function class $\mathcal{F}$, we define the notation $R_n \mathcal{F}$ by $R_n \mathcal{F} := \sup_{f \in \mathcal{F}} \frac{1}{n} \sum_{i=1}^{n} \sigma_i f(\vec{\mathbf{x}}_i)$.

**Theorem A.1** ((Bartlett et al., 2005, Theorem 2.1)). Let $\mathcal{X}, P$ be a probability space, $\left\{\vec{\mathbf{x}}_i\right\}_{i=1}^{n}$ be independent random variables distributed according to $P$. Let $\mathcal{F}$ be a class of functions that map $\mathcal{X}$ into $[a, b]$. Assume that there is some $r > 0$ such that for every $f \in \mathcal{F}$, $\mathrm{Var}\left[f(\vec{\mathbf{x}}_i)\right] \leq r$. Then, for every $x > 0$, with probability at least $1 - e^{-x}$,

$$\sup_{f \in \mathcal{F}} \left(\mathbb{E}_P[f(\mathbf{x})] - \mathbb{E}_{P_n}[f(\mathbf{x})]\right) \leq \inf_{\alpha > 0} \left(2(1+\alpha)\mathbb{E}_{\left\{\vec{\mathbf{x}}_i\right\}_{i=1}^{n}, \{\sigma_i\}_{i=1}^{n}}[R_n \mathcal{F}] + \sqrt{\frac{2rx}{n}} + (b-a)\left(\frac{1}{3} + \frac{1}{\alpha}\right)\frac{x}{n}\right), \tag{18}$$

and with probability at least $1 - 2e^{-x}$,

$$\sup_{f \in \mathcal{F}} \left(\mathbb{E}_P[f(\mathbf{x})] - \mathbb{E}_{P_n}[f(\mathbf{x})]\right) \leq \inf_{\alpha \in (0,1)} \left(\frac{2(1+\alpha)}{1-\alpha}\mathbb{E}_{\{\sigma_i\}_{i=1}^{n}}[R_n \mathcal{F}] + \sqrt{\frac{2rx}{n}} + (b-a)\left(\frac{1}{3} + \frac{1}{\alpha} + \frac{1+\alpha}{2\alpha(1-\alpha)}\right)\frac{x}{n}\right). \tag{19}$$

$P_n$ is the empirical distribution over $\left\{\vec{\mathbf{x}}_i\right\}_{i=1}^{n}$ with $\mathbb{E}_{P_n}[f(\mathbf{x})] = \frac{1}{n} \sum_{i=1}^{n} f(\vec{\mathbf{x}}_i)$. Moreover, the same results hold for $\sup_{f \in \mathcal{F}} \left(\mathbb{E}_{P_n}[f(\mathbf{x})] - \mathbb{E}_P[f(\mathbf{x})]\right)$.

In addition, we have the contraction property for Rademacher complexity, which is due to Ledoux and Talagrand (Ledoux, 1991).

**Theorem A.2.** Let $\phi$ be a contraction, that is, $|\phi(x) - \phi(y)| \leq \mu |x - y|$ for $\mu > 0$. Then, for every function class $\mathcal{F}$,

$$\mathbb{E}_{\{\sigma_i\}_{i=1}^{n}}\left[R_n \phi \circ \mathcal{F}\right] \leq \mu \mathbb{E}_{\{\sigma_i\}_{i=1}^{n}}\left[R_n \mathcal{F}\right], \tag{20}$$

where $\phi \circ \mathcal{F}$ is the function class defined by $\phi \circ \mathcal{F} = \{\phi \circ f \colon f \in \mathcal{F}\}$.

*Definition* A.2 (Sub-root function,(Bartlett et al., 2005, Definition 3.1)). A function $\psi \colon [0, \infty) \to [0, \infty)$ is sub-root if it is nonnegative, nondecreasing and if $\frac{\psi(r)}{\sqrt{r}}$ is nonincreasing for $r > 0$.

**Theorem A.3** ((Bartlett et al., 2005, Theorem 3.3)). Let $\mathcal{F}$ be a class of functions with ranges in $[a, b]$ and assume that there are some functional $T \colon \mathcal{F} \to \mathbb{R}+$ and some constant $\bar{B}$ such that for every $f \in \mathcal{F}$, $\mathrm{Var}[f] \leq T(f) \leq \bar{B}P(f)$. Let $\psi$ be a sub-root function and let $r^*$ be the fixed point of $\psi$. Assume that $\psi$ satisfies that, for any $r \geq r^*$, $\psi(r) \geq \bar{B}\mathfrak{R}(\{f \in \mathcal{F} \colon T(f) \leq r\})$. Fix $x > 0$, then for any $K_0 > 1$, with probability at least $1 - e^{-x}$,

$$\forall f \in \mathcal{F}, \quad \mathbb{E}_P[f] \leq \frac{K_0}{K_0 - 1}\mathbb{E}_{P_n}[f] + \frac{704 K_0}{\bar{B}}r^* + \frac{x\left(11(b-a) + 26\bar{B}K_0\right)}{n}.$$

Also, with probability at least $1 - e^{-x}$,

$$\forall f \in \mathcal{F}, \quad \mathbb{E}_{P_n}[f] \leq \frac{K_0 + 1}{K_0}\mathbb{E}_P[f] + \frac{704 K_0}{\bar{B}}r^* + \frac{x\left(11(b-a) + 26\bar{B}K_0\right)}{n}.$$

# B MORE DETAILS ABOUT THE THEORETICAL RESULTS AND EXPERIMENTS

## B.1 GENERATION OF THE RANDOM SAMPLE $\mathbf{Q}$

We note that the sample $\mathbf{Q}$ contains i.i.d. random variables $\left\{\vec{\mathbf{q}}_i\right\}_{i=1}^{N}$ distributed according to $P$, the same distribution as the training features $\mathbf{S}$. When $N \leq n$, we can directly use a subset of size $N$ of

**S** as **Q**. Otherwise, **S** is used as a subset of **Q**. A remaining set $\mathbf{Q}'$ of $N - n$ i.i.d. random variables distributed according to $P$ is sampled, and $\overline{\mathbf{Q}} = \mathbf{Q}' \cup \mathbf{S}$. To be shown in the next paragraph, if $P$ is known, $\mathbf{Q}'$ can be sampled exactly according to $P$ so that $\overline{\mathbf{Q}}$ serves as **Q**. If $P$ is unknown, $\mathbf{Q}'$ can be sampled approximately according to $P$, and $\overline{\mathbf{Q}}$ can be used in practice as an approximation to **Q**.

In practice, $\mathbf{Q}'$ can be sampled depending on if $P$ is known or not. $\mathbf{Q}'$ can be sampled exactly according to $P$ if $P$ is known, or sampled approximately according to $P$ if $P$ is unknown. In particular, if $P$ is a known distribution, $\mathbf{Q}'$ can be sampled by inverse transform sampling with invertible cumulative distribution function (CDF) of the distribution $P$ or rejection sampling using the probability density function of $P$ without invertible CDF. If $P$ is unknown, we can train generative models on **S** and then generate synthetic data points distributed approximately according to $P$ as the sample $\mathbf{Q}'$. These generative models learn an approximation $\widehat{P}$ to the underlying data distribution $P$ and enable efficient sampling from $\widehat{P}$. Popular classes include (1) diffusion models, which generate samples by iteratively denoising noise using a learned reverse-time stochastic process (Ho et al., 2020; Song et al.), (2) flow matching methods, which directly learn continuous-time vector fields that transport noise to data distributions (Lipman et al., 2023), and (3) normalizing flows, which construct an invertible transformation mapping a simple base distribution (e.g., Gaussian) into the data distribution $P$, trained via maximum likelihood (Rezende & Mohamed, 2015; Papamakarios et al., 2021). These approaches provide a principled mechanism for generating synthetic data points that approximate the unknown $P$.

## B.2 DIFFERENCE FROM EXISTING KERNEL LEARNING THEORY

In this subsection, we demonstrate that our results in Section 3 are fundamentally different from the existing kernel learning theory, such as the minimax lower rate in (Caponnetto & De Vito, 2007) for kernel regression using the regular NTK defined in (2). In particular, our regression risk bound obtained by the network with channel attention (1) is sharper and fundamentally different from that in (Caponnetto & De Vito, 2007). Under the same source condition on the target function that $f^* \in \mathcal{H}_{K^{(\text{attn})}}(\mu_0) = [\mathcal{H}_K]^3(\mu_0)$, the existing minimax lower rate in (Caponnetto & De Vito, 2007, Theorem 2) for kernel regression using the regular NTK (2) is $\mathcal{O}\left(\tau(\delta)\, n^{-\frac{6\alpha}{6\alpha+1}}\right)$ with probability $1 - \delta$, where $\tau(\delta) \to \infty$ as $\delta \to 0$, under the polynomial EDR of $\lambda_j \asymp j^{-2\alpha}$ for $\alpha > 1/2$. That is, to ensure the rate holds with probability approaching 1 (or $1 - \delta$ with $\delta \to 0$), there is an additional cost $\tau(\delta) \to \infty$ as $\delta \to 0$ (Caponnetto & De Vito, 2007). This is the fundamental reason that the rate obtained by (Li et al., 2024, Proposition 13) is $\mathcal{O}\left(n^{-\frac{6\alpha}{6\alpha+1}}\right)\log^2(1/\delta)$, which contains the additional logarithmic factor $\log^2(1/\delta)$ compared to our rate in Theorem 3.2.

In strong contrast, the two-layer NN with channel attention (1) trained by GD achieves the sharper and minimax optimal rate of $\mathcal{O}\left(n^{-\frac{6\alpha}{6\alpha+1}}\right)$ in Theorem 3.2. The fundamental reason for such a sharper rate is that canonical kernel regression methods (Caponnetto & De Vito, 2007; Yao et al., 2007) only apply to kernels with the original capacity condition, such as the regular NTK (2) with the polynomial EDR $\lambda_j \asymp j^{-2\alpha}$. On the other hand, two-layer NN with channel attention (1) trained by GD approximately performs kernel regression with a completely different new kernel, namely the attention kernel $K^{(\text{attn})}$, which satisfies the smoother capacity condition $\lambda_j^{(\text{attn})} = \lambda_j^3 \asymp j^{-6\alpha}$. Our key insight is that the interpolation space $[\mathcal{H}_K]^3$ is in fact the RKHS associated with the integral kernel, that is, $[\mathcal{H}_K]^3 = \mathcal{H}_{K^{(\text{attn})}}$. Kernel regression with the integral kernel and the target function $f^* \in \mathcal{H}_{K^{(\text{attn})}}(\mu_0)$ renders the minimax optimal rate of $\mathcal{O}\left(n^{-\frac{6\alpha}{6\alpha+1}}\right)$ according to the analytical results in (Stone, 1985; Yang & Barron, 1999; Yuan & Zhou, 2016), which coincides with the rate in Theorem 3.2.

## B.3 BETTER LOWER BOUND FOR NETWORK WIDTH $m$

The lower bound on the network width $m$ required for our result in Theorem 3.2, $m \gtrsim n^{48\alpha/(6\alpha+1)}d^3\log^3 m$ with $\alpha = d/(2(d-1))$, is smaller than that required by the current state-of-the-art. In particular, (Suh et al., 2022, Theorem 3.11) show that $m/\log^3 m \gtrsim L^{20}n^{24}$, where $L$ is the number of layers of the DNN in their work, which further implies $m/\log^3 m \gtrsim 2^{20}n^{24}$

even for the two-layer NN considered here with $L = 2$. Similarly, (Li et al., 2024) require $m/(\log m)^{12} \gtrsim n^{24}$ for regression with target function $f^* \in [\mathcal{H}_K]^3$, which is the same source condition studied in this paper. Both lower bounds for $m$ in (Suh et al., 2022; Li et al., 2024) are therefore much larger than ours in the regime $n \to \infty$ with fixed $d$, which is precisely the setting considered in prior works on training over-parameterized neural networks for nonparametric regression with sharp rates and algorithmic guarantees (Hu et al., 2021; Suh et al., 2022; Li et al., 2024; Yang & Li, 2024; Yang, 2025).

## C   DETAILED PROOFS

We present the detailed proofs for the theoretical results of this paper, and the basic notations are introduced are first introduced in Section C.1.

### C.1   BASIC DEFINITIONS

We introduce the following definitions for our analysis. We introduce the following definitions for the proof of Theorem 3.1 and Theorem 3.2. Let the gram matrix of $K^{(\mathrm{attn})}$ over the training features $\mathbf{S}$ be $\mathbf{K}^{(\mathrm{attn})} \in \mathbb{R}^{n \times n}$, $\mathbf{K}_{ij}^{(\mathrm{attn})} = K^{(\mathrm{attn})}(\overrightarrow{\mathbf{x}}_i, \overrightarrow{\mathbf{x}}_j)$ for $i, j \in [n]$, and $\mathbf{K}_n^{(\mathrm{attn})} \coloneqq \mathbf{K}^{(\mathrm{attn})}/n$. Similarly, $\widehat{K}^{(\mathrm{attn})} \in \mathbb{R}^{n \times n}$ is the gram matrix of $\widehat{K}^{(\mathrm{attn})}$ over $\mathbf{S}$, and $\widehat{\mathbf{K}}_n^{(\mathrm{attn})} = \widehat{\mathbf{K}}^{(\mathrm{attn})}/n$. Let the singular value decomposition of $\mathbf{K}_n^{(\mathrm{attn})}$ be $\mathbf{K}_n^{(\mathrm{attn})} = \mathbf{U}^{(\mathrm{attn})} \mathbf{\Sigma}^{(\mathrm{attn})} \mathbf{U}^{(\mathrm{attn})\top}$, where $\mathbf{\Sigma}^{(\mathrm{attn})}$ is a diagonal matrix with its diagonal elements $\left\{ \widehat{\lambda}_i^{(\mathrm{attn})} \right\}_{i=1}^n$ being the eigenvalues of $\mathbf{K}_n^{(\mathrm{attn})}$ and sorted in a non-increasing order. We have $\widehat{\lambda}_1^{(\mathrm{attn})} \in (0, 1)$, and we show in Proposition D.2 deferred to Section D.1 that $\mathbf{K}_n^{(\mathrm{attn})}$ is always non-singular. We define

$$\mathbf{u}(t) \coloneqq \widehat{\mathbf{y}}(t) - \mathbf{y} \tag{21}$$

as the difference between the network output $\widehat{\mathbf{y}}(t)$ and the training response vector $\mathbf{y}$ right after the $t$-th step of GD. Let $0 < \tau \leq 1$, for $\tau$, $t \geq 0$, and $T \geq 1$ we define the following quantities: $c_{\mathbf{u}} \coloneqq \mu_0 / \min\left\{ \sqrt{2e\eta}, 1 \right\} + \sigma_0 + \tau + 1$,

$$R \coloneqq \frac{\eta c_{\mathbf{u}} \sqrt{2d + 3\log(2mn)} T}{\sqrt{m}}, \tag{22}$$

$$\mathcal{V}_t \coloneqq \left\{ \mathbf{v} \in \mathbb{R}^n \colon \mathbf{v} = -\left( \mathbf{I}_n - \eta \mathbf{K}_n^{(\mathrm{attn})} \right)^t f^*(\mathbf{S}) \right\}, \tag{23}$$

$$\mathcal{E}_{t,\tau} \coloneqq \left\{ \mathbf{e} \colon \mathbf{e} = \overrightarrow{\mathbf{e}}_1 + \overrightarrow{\mathbf{e}}_2 \in \mathbb{R}^n, \overrightarrow{\mathbf{e}}_1 = -\left( \mathbf{I}_n - \eta \mathbf{K}_n^{(\mathrm{attn})} \right)^t \mathbf{w}, \left\| \overrightarrow{\mathbf{e}}_2 \right\|_2 \leq \sqrt{n}\tau \right\}. \tag{24}$$

We define

$$\mathcal{W}_0 \coloneqq \{ \mathbf{W}(0) \colon (31) \text{ holds} \} \tag{25}$$

as the set of all the good random initializations which satisfy (31) in Theorem C.1.

Lemma C.3 in Section C.4 shows that with high probability over the random initialization $\mathbf{W}(0)$ and the random noise $\mathbf{w}$, the distance of every weighting vector $\mathbf{w}_r(t)$ to its initialization $\mathbf{w}_r(0)$ is bounded by $R$, and the distance of every weighting vector $a_r(t)$ to its initialization $0$ is bounded by $2R$. In addition, $\mathbf{u}(t)$ can be composed into two vectors, $\mathbf{u}(t) = \mathbf{v}(t) + \mathbf{e}(t)$ such that $\mathbf{v}(t) \in \mathcal{V}_t$ and $\mathbf{e}(t) \in \mathcal{E}_{t,\tau}$. We then define the set of the neural network weights during the training by GD using Algorithm 1 as follows:

$$\mathcal{W}(\mathbf{S}, \mathbf{W}(0), T) \coloneqq \left\{ (\mathbf{W}, \boldsymbol{a}) \colon \exists t \in [T] \text{ s.t. } \mathrm{vec}\,(\mathbf{W}) = \mathrm{vec}\,(\mathbf{W}(0)) - \sum_{t'=0}^{t-1} \frac{\eta}{n} \mathbf{Z}_{\mathbf{S}}(t') \mathbf{u}(t'), \right.$$

$$\boldsymbol{a}(t) = \sum_{t'=0}^{t-1} -\frac{\eta}{n\sqrt{m}} \mathbf{A} \boldsymbol{\sigma}(\mathbf{W}(t'), \mathbf{S}) \mathbf{u}(t'),$$

$$\mathbf{u}(t') \in \mathbb{R}^n, \mathbf{u}(t') = \mathbf{v}(t') + \mathbf{e}(t'), \mathbf{v}(t') \in \mathcal{V}_{t'}, \mathbf{e}(t') \in \mathcal{E}_{t',\tau}, \text{ for all } t' \in [0, t-1] \Big\}. \tag{26}$$

We will also show by Lemma C.3 that with high probability over $\mathbf{w}$, $\mathcal{W}(\mathbf{S}, \mathbf{W}(0), T)$ is the set of the weights of the two-layer NN (1) trained by GD on the training features $\mathbf{S}$ with the random initialization $\mathbf{W}(0)$ and the number of steps of GD not greater than $T$.

The set of the functions represented by the neural network with weights in $\mathcal{W}(\mathbf{S}, \mathbf{W}(0), T)$ is then defined as

$$\mathcal{F}_{\mathrm{NN}}(\mathbf{S}, \mathbf{W}(0), T) \coloneqq \{f_t = f(\mathbf{W}(t), \boldsymbol{a}(t), \cdot) \colon \exists t \in [T], (\mathbf{W}(t), \boldsymbol{a}(t)) \in \mathcal{W}(\mathbf{S}, \mathbf{W}(0), T)\}. \tag{27}$$

We also define the function class $\mathcal{F}(B, w)$ for any $B, w > 0$ as

$$\mathcal{F}(B, w) \coloneqq \{f \colon f = h + e, h \in \mathcal{H}_K(B), \|e\|_\infty \le w\}. \tag{28}$$

We will show by Theorem C.4 in Section C.4 that with high probability over $\mathbf{w}$, $\mathcal{F}_{\mathrm{NN}}(\mathbf{S}, \mathbf{W}(0), T)$ is a subset of $\mathcal{F}(B_h, w)$, where a smaller $w$ requires a larger network width $m$, and $B_h > \mu_0$ is an absolute positive constant defined by

$$B_h \coloneqq \mu_0 + 1 + \sqrt{2}. \tag{29}$$

## C.2 Uniform Convergence to the NTK (2)

We define the following functions with $\mathbf{W} = \{\mathbf{w}_r\}_{r=1}^m$:

$$h(\mathbf{w}, \mathbf{u}, \mathbf{v}) \coloneqq \sigma\left(\mathbf{w}^\top \mathbf{u}\right) \sigma\left(\mathbf{w}^\top \mathbf{v}\right), \qquad \widehat{h}(\mathbf{W}, \mathbf{u}, \mathbf{v}) \coloneqq \frac{1}{m} \sum_{r=1}^m h(\vec{\mathbf{w}}_r, \mathbf{u}, \mathbf{v}), \tag{30}$$

where $\mathbf{u}, \mathbf{v} \in \mathcal{X}$. Then we have the following theorem stating the uniform convergence of $\widehat{h}(\mathbf{W}(0), \cdot, \cdot)$ to $K(\cdot, \cdot)$.

**Theorem C.1.** Suppose $m \gtrsim n$ and $m/\log m \ge d$. Then with probability at least $1 - 1/n$ over the random initialization $\mathbf{W}(0) = \left\{\vec{\mathbf{w}}_r(0)\right\}_{r=1}^m$,

$$\sup_{\mathbf{u} \in \mathcal{X}, \mathbf{v} \in \mathcal{X}} \left| K(\mathbf{u}, \mathbf{v}) - \widehat{h}(\mathbf{W}(0), \mathbf{u}, \mathbf{v}) \right| \le C_1(m, d, 1/n) \lesssim d \log m \sqrt{\frac{d \log m}{m}}, \tag{31}$$

where $C_1(m, d, 1/n)$ is a positive number depending on $(m, d, n)$, and its formal definition is deferred to (48) in Section C.5.

*Proof.* This theorem follows from Theorem C.8 in Section C.5.

$\square$

We define

$$\mathcal{W}_0 \coloneqq \{\mathbf{W}(0) \colon (31) \text{ holds}\} \tag{32}$$

as the set of all the good random initializations which satisfy (31) in Theorem C.1. Theorem C.1 shows that we have good random initialization with high probability, that is, $\Pr\left[\mathbf{W}(0) \in \mathcal{W}_0\right] \ge 1 - 1/n$. When $\mathbf{W}(0) \in \mathcal{W}_0$, the uniform convergence (31) holds with high probability, which is important for the analysis of the training dynamics of the two-layer NN with channel attention (1) by GD.

## C.3 Proofs for the Main Result, Theorem 3.1 and Theorem 3.2

We note that Theorem C.4, Theorem C.6, and Theorem C.22 are the formal versions of Theorem 5.2, Theorem 5.3, and Theorem 5.1 in Section 5.2 of this paper.

**Proof of Theorem 3.1.** We apply Theorem C.6 and Theorem C.7 to prove this theorem.

First, with the condition on $m$ in this theorem, Theorem C.1 hold, and $\Pr\left[\mathbf{W}(0) \in \mathcal{W}_0\right] \geq 1 - 1/n$. With $\eta = \Theta(1)$, it follows by Theorem C.7 that with probability at least $1 - \exp\left(-\Theta(n\widehat{\varepsilon}_n^2)\right)$ over the random noise $\mathbf{w}$,

$$\mathbb{E}_{P_n}\left[(f_t - f^*)^2\right] \lesssim \frac{1}{\eta t}.$$

Plugging such bound for $\mathbb{E}_{P_n}\left[(f_t - f^*)^2\right]$ in (44) of Theorem C.6 leads to

$$\mathbb{E}_P\left[(f_t - f^*)^2\right] \lesssim \frac{1}{\eta t} + \varepsilon_n^2 + w. \tag{33}$$

Due to the definition of $\widehat{T}$ and $\widehat{\varepsilon}_n^2$, we have

$$\widehat{\varepsilon}_n^2 \leq \frac{1}{\eta\widehat{T}} \leq \frac{2}{\eta(\widehat{T}+1)} \leq 2\widehat{\varepsilon}_n^2. \tag{34}$$

It follows from Lemma C.19 that that $\widehat{\varepsilon}_n^2 \asymp \varepsilon_n^2$ with probability at least $1 - 4\exp(-\Theta(n\varepsilon_n^2))$ over $\mathbf{S}$. In addition, combined with the fact that $T \asymp \widehat{T}$, for any $t \in [c_t T, T]$, we have

$$\frac{1}{\eta t} \asymp \frac{1}{\eta\widehat{T}} \asymp \frac{1}{\eta T} \asymp \widehat{\varepsilon}_n^2 \asymp \varepsilon_n^2.$$

We set $w = \varepsilon_n^2$ in (33), then with $\eta = \Theta(1)$,

$$\mathbb{E}_P\left[(f_t - f^*)^2\right] \lesssim \varepsilon_n^2. \tag{35}$$

With $N \gtrsim \log(n/\delta)/\varepsilon_n^8$, the requirement on $N$, (40) in Theorem C.4 that $N \gtrsim \max\left\{T^2\log(n/\delta)/w^2, T^4\log(n/\delta)\right\}$ is satisfied. In addition, with

$$m \gtrsim \max\left\{(d^2 + \log^2 m)/\varepsilon_n^{16}, (d\log m)^3/\varepsilon_n^8, n\right\},$$

and $w = \varepsilon_n^2$, the condition (39) on $m$ in Theorem C.4 is satisfied. $\qquad\square$

**Proof of Theorem 3.2.** We apply Theorem 3.1 to prove this theorem. First, it then follows from Theorem D.1 in Section D.1 that $\lambda_j^{(\text{attn})} = \lambda_j^3 \asymp j^{-6\alpha}$ for $j \geq 1$. For such EDR of $\left\{\lambda_j^{(\text{attn})}\right\}_{j \geq 1}$, it is well known, such as (Raskutti et al., 2014, Corollary 3), that $\varepsilon_n^2 \asymp n^{-\frac{6\alpha}{6\alpha+1}}$. It also follows from Lemma C.19 that that $\widehat{\varepsilon}_n^2 \asymp \varepsilon_n^2$ with probability at least $1 - 4\exp(-\Theta(n\varepsilon_n^2))$ over $\mathbf{S}$. This theorem is then proved by plugging in $\varepsilon_n^2 \asymp \varepsilon_n^2 \asymp n^{-\frac{6\alpha}{6\alpha+1}}$ and $w = \varepsilon_n^2 \asymp n^{-\frac{6\alpha}{6\alpha+1}}$ in Theorem 3.1. $\qquad\square$

**Proposition C.2.** We have $\varepsilon_n^2 \leq \varepsilon_{K,n}^2$.

*Proof.* First, it follows from Theorem D.1 that $0 < \lambda_j < (1+\pi)/(2\pi) < 1$ for all $j \geq 1$, and $\lambda_j^{(\text{attn})} = \lambda_j^3 < \lambda_j$ for all $j \geq 1$. As a result, we have

$$R_{K^{(\text{attn})}}(\varepsilon) \leq R_K(\varepsilon), \quad \forall \varepsilon \geq 0. \tag{36}$$

Setting $\varepsilon = \varepsilon_n$ in (36), we have $\varepsilon_n^2 = \sigma_0 R_{K^{(\text{attn})}}(\varepsilon_n) \leq \sigma_0 R_K(\varepsilon_n)$. Since $\sigma_0 R_K(\varepsilon)$ is a sub-root function of $\varepsilon^2$ with the unique fixed point of $\varepsilon_{K,n}^2$, it then follows from (Bartlett et al., 2005, Lemma 3.2) that $\varepsilon_n^2 \leq \varepsilon_{K,n}^2$.

$\qquad\square$

## C.4 KEY TECHNICAL RESULTS

We present our key technical results regarding optimization and generalization of the two-layer NN (1) trained by GD in this section. Lemma C.3 is our main result about the optimization of the network (1), which states that with high probability over $\mathbf{W}(0)$ and $\mathbf{w}$, the weights of the network $(\mathbf{W}(t), \boldsymbol{a}(t))$ obtained right after the $t$-th step of GD using Algorithm 1 belongs to $\mathcal{W}(\mathbf{S}, \mathbf{W}(0), T)$. Furthermore, every weighing vector $\mathbf{w}_r$ and $a_r$ have bounded distances to their corresponding initialized values, $\overrightarrow{\mathbf{w}}_r(0)$ and $0$. The proof of Lemma C.3 is based on Lemma C.9, Lemma C.10, Lemma C.11, and Lemma C.12 deferred to Section C.6 of this appendix.

**Lemma C.3.** Suppose $\delta \in (0,1)$, $N \gtrsim (d \log N)^3$, $m \gtrsim (d \log m)^3$,

$$m \gtrsim \max \left\{ T^2 (d \log m)^3 / \tau^2, T^6 (d^2 + \log^2 m) / \tau^2, n \right\}, \tag{37}$$

$$N \gtrsim T^2 \log(n/\delta) / \tau^2, \tag{38}$$

the neural network $f(\mathbf{W}(t), \boldsymbol{a}(t), \cdot)$ trained by GD using Algorithm 1 with the learning rate $\eta = \Theta(1) \in (0,1)$, the random initialization $\mathbf{W}(0) \in \mathcal{W}_0$. Then with probability at least $1 - 1/n - \delta - \exp(-\Theta(n))$ over the random initialization $\mathbf{W}(0)$, the random noise $\mathbf{w}$, and the random sample $\mathbf{Q}$, $(\mathbf{W}(t), \boldsymbol{a}(t)) \in \mathcal{W}(\mathbf{S}, \mathbf{W}(0), T)$ for every $t \in [T]$. Moreover, for every $t \in [0, T]$, $\mathbf{u}(t) = \mathbf{v}(t) + \mathbf{e}(t)$ where $\mathbf{u}(t) = \widehat{\mathbf{y}}(t) - \mathbf{y}$, $\mathbf{v}(t) \in \mathcal{V}_t$, $\mathbf{e}(t) \in \mathcal{E}_{t,\tau}$, $\|\mathbf{u}(t)\|_2 \leq c_\mathbf{u} \sqrt{n}$, and $\left\| \overrightarrow{\mathbf{w}}_r(t) - \overrightarrow{\mathbf{w}}_r(0) \right\|_2 \leq R$, $|a_r(t) - a_r(0)| \leq 2R$.

The following theorem, Theorem C.4, states that with high probability over $\mathbf{w}$, $\mathcal{F}_{\mathrm{NN}}(\mathbf{S}, \mathbf{W}(0), T) \subseteq \mathcal{F}(B_h, w)$, with the early stopping mechanism such that $T \leq \widehat{T}$.

**Theorem C.4.** Suppose $\delta \in (0,1)$, $w \in (0,1)$,

$$m \gtrsim \max \left\{ T^4 (d \log m)^3, T^8 (d^2 + \log^2 m), T^2 (d \log m)^3 / w^2, T^4 (d^2 + \log^2 m) / w^2, n \right\}, \tag{39}$$

$$N \gtrsim \max \left\{ T^4 \log(n/\delta), T^2 \log(n/\delta) / w^2 \right\}, \tag{40}$$

and the neural network $f_t = f(\mathbf{W}(t), \boldsymbol{a}(t), \cdot)$ is trained by GD using Algorithm 1 with the learning rate $\eta = \Theta(1) \in (0,1)$, the random initialization $\mathbf{W}(0) \in \mathcal{W}_0$. Then for every $t \in [T]$ with $T \leq \widehat{T}$, with probability at least $1 - 1/n - \delta - \exp(-\Theta(n)) - \exp(-\Theta(n \widehat{\varepsilon}_n^2))$ over the random initialization $\mathbf{W}(0)$, the random noise $\mathbf{w}$, and the random sample $\mathbf{Q}$, $f_t \in \mathcal{F}_{\mathrm{NN}}(\mathbf{S}, \mathbf{W}(0), T) \subseteq \mathcal{F}(B_h, w)$, and $f_t$ has the following decomposition on $\mathcal{X}$:

$$f_t = h_t + e_t, \tag{41}$$

where $h_t \in \mathcal{H}_K(B_h)$ with $B_h$ defined in (29), $e_t \in L^\infty$ with $\|e_t\|_\infty \leq w$.

Lemma C.5 below gives a sharp upper bound for the Rademacher complexity of a localized subset of the function class $\mathcal{F}(B, w)$. Based on Lemma C.5, Theorem C.4, and using the local Rademacher complexity based analysis (Bartlett et al., 2005), Theorem C.6 presents a sharp upper bound for the nonparametric regression risk, $\mathbb{E}_P \left[ (f_t - f^*)^2 \right]$, where $f_t$ is the function represented by the two-layer NN with channel attention (1) right after the $t$-th step of GD using Algorithm 1.

**Lemma C.5.** For every $B, w > 0$ every $r > 0$,

$$\mathfrak{R} \left( \left\{ f \in \mathcal{F}(B, w) \colon \mathbb{E}_P \left[ f^2 \right] \leq r \right\} \right) \leq \varphi_{B,w}(r), \tag{42}$$

where

$$\varphi_{B,w}(r) := \min_{Q \colon Q \geq 0} \left( (\sqrt{r} + w) \sqrt{\frac{Q}{n}} + B \left( \frac{\sum_{q=Q+1}^\infty \lambda_q^{(\mathrm{attn})}}{n} \right)^{1/2} \right) + w. \tag{43}$$

We then have the following theorem giving the sharp bound for the regression risk of $f_t$ right after every step $t$ of GD.

**Theorem C.6.** Suppose $w \in (0,1)$ and $m, N$ satisfy (39) and (40), respectively. Suppose the neural network $f_t = f(\mathbf{W}(t), \boldsymbol{a}(t), \cdot)$ is trained by GD in Algorithm 1 with the learning rate $\eta = \Theta(1) \in (0,1)$ on the random initialization $\mathbf{W}(0) \in \mathcal{W}_0$, and $T \leq \widehat{T}$. Then for every $t \in [T]$ and every $\delta \in (0,1)$, with probability at least $1 - 1/n - \delta - \exp(-\Theta(n)) - \exp(-\Theta(n \widehat{\varepsilon}_n^2)) - \exp(-\Theta(n \varepsilon_n^2))$ over the random initialization $\mathbf{W}(0)$, the random noise $\mathbf{w}$, the random training features $\mathbf{S}$, and the random sample $\mathbf{Q}$,

$$\mathbb{E}_P \left[ (f_t - f^*)^2 \right] - 2 \mathbb{E}_{P_n} \left[ (f_t - f^*)^2 \right] \lesssim \varepsilon_n^2 + w. \tag{44}$$

Theorem C.7 below shows that the empirical loss $\mathbb{E}_{P_n} \left[ (f_t - f^*)^2 \right]$ is bounded by $\Theta(1/(\eta t))$ with high probability over $\mathbf{w}$. Such upper bound for the empirical loss by Theorem C.7 will be plugged in the risk bound in Theorem C.6 to prove Theorem 3.1 and Theorem 3.2.

**Theorem C.7.** Suppose the neural network trained after the $t$-th step of GD, $f_t = f(\mathbf{W}(t), \boldsymbol{a}(t), \cdot)$, satisfies $\mathbf{u}(t) = f_t(\mathbf{S}) - \mathbf{y} = \mathbf{v}(t) + \mathbf{e}(t)$ with $\mathbf{v}(t) \in \mathcal{V}_t$ and $\mathbf{e}(t) \in \mathcal{E}_{t,\tau}$, and $t \in [T]$ with $T \le \widehat{T}$. If

$$\tau \lesssim \frac{1}{\eta T}, \tag{45}$$

Then for every $t \in [T]$, with probability at least $1 - \exp\left(-\Theta(n\widehat{\varepsilon}_n^2)\right)$ over the random noise $\mathbf{w}$, we have

$$\mathbb{E}_{P_n}\left[(f_t - f^*)^2\right] \le \frac{3}{\eta t}\left(\frac{\mu_0^2}{2e} + \frac{1}{\eta T} + 2\right). \tag{46}$$

## C.5 PROOFS FOR RESULTS IN SECTION C.2 AND SECTION C.4

We have the following theorem, Theorem C.8, regarding the uniform convergence to the PD kernel $K$ defined in (2) on the unit sphere $\mathcal{X}$. The proof of Theorem C.8 is deferred to Section D.2 of this appendix.

**Theorem C.8.** Let $\mathbf{W}(0) = \left\{\overrightarrow{\mathbf{w}}_r(0)\right\}_{r=1}^m$, where each $\overrightarrow{\mathbf{w}}_r(0) \sim \mathcal{N}(\mathbf{0}, \kappa^2\mathbf{I}_d)$ for $r \in [m]$. Then for any $\delta \in (0,1)$, with probability at least $1 - \delta$ over $\mathbf{W}(0)$,

$$\sup_{\mathbf{u} \in \mathcal{X}, \mathbf{v} \in \mathcal{X}} \left|K(\mathbf{u}, \mathbf{v}) - \widehat{h}(\mathbf{W}(0), \mathbf{u}, \mathbf{v})\right| \le C_1(m, d, \delta), \tag{47}$$

where

$$C_1(m, d, \delta) :=$$

$$\frac{12M_{\frac{\delta}{2(1+2m)^{2d}}}\left(2d + 3\log\frac{6m(1+2m)^{2d}}{\delta}\right)}{m} + M^2_{\frac{\delta}{2(1+2m)^{2d}}}\left(\sqrt{\frac{2\log\frac{4(1+2m)^{2d}}{\delta}}{m}} + \frac{16\log\frac{4(1+2m)^{2d}}{\delta}}{3m}\right), \tag{48}$$

$$M_\delta := \kappa\sqrt{2\log(2m)} + \kappa\sqrt{2\log(3/\delta)} + \frac{\sqrt{2d + 3\log(3m/\delta)}}{m}.$$

In addition, when $m \gtrsim n$, $m/\log m \ge d$, and $\delta \asymp 1/n$, we have $M_{\frac{\delta}{2(1+2m)^{2d}}} \lesssim \sqrt{d\log m}$ and

$$C_1(m, d, \delta) \lesssim d\log m\sqrt{\frac{d\log m}{m}} + \frac{(d\log m)^{3/2}}{m} \lesssim d\log m\sqrt{\frac{d\log m}{m}}.$$

**Proof of Lemma C.3.** First, $\mathbf{E}_{m,\eta,\delta}$ is defined by (80) of Lemma C.10, and we have

$$\mathbf{E}_{m,\eta,\delta} \lesssim \frac{T^2\sqrt{n}(d + \log m) + \sqrt{n}(d\log m)^{3/2}}{\sqrt{m}} + \sqrt{\frac{n\log(n/\delta)}{N}}.$$

When $m \gtrsim \max\left\{T^2(d\log m)^3/\tau^2, T^6(d^2 + \log^2 m)/\tau^2, n\right\}$, $N \gtrsim T^2\log(n/\delta)/\tau^2$ with proper constants, it can be verified that $\mathbf{E}_{m,\eta,\delta} \le \tau\sqrt{n}/T$. We then use mathematical induction to prove this theorem. We will first prove that $\mathbf{u}(t) = \mathbf{v}(t) + \mathbf{e}(t)$ where $\mathbf{v}(t) \in \mathcal{V}_t$, $\mathbf{e}(t) \in \mathcal{E}_{t,\tau}$, and $\|\mathbf{u}(t)\|_2 \le c_{\mathbf{u}}\sqrt{n}$ for for all $t \in [0, T]$.

When $t = 0$, we have

$$\mathbf{u}(0) = -\mathbf{y} = \mathbf{v}(0) + \mathbf{e}(0), \tag{49}$$

where $\mathbf{v}(0) := -f^*(\mathbf{S}) = -\left(\mathbf{I} - \eta\mathbf{K}_n^{(\text{attn})}\right)^0 f^*(\mathbf{S})$, $\mathbf{e}(0) = -\mathbf{w} = \overrightarrow{\mathbf{e}}_1(0) + \overrightarrow{\mathbf{e}}_2(0)$ with $\overrightarrow{\mathbf{e}}_1(0) = -\left(\mathbf{I} - \eta\mathbf{K}_n^{(\text{attn})}\right)^0 \mathbf{w}$ and $\overrightarrow{\mathbf{e}}_2(0) = \mathbf{0}$. Therefore, $\mathbf{v}(0) \in \mathcal{V}_0$ and $\mathbf{e}(0) \in \mathcal{E}_{0,\tau}$. Also, it follows from the proof of Lemma C.9 that $\|\mathbf{u}(0)\|_2 \le c_{\mathbf{u}}\sqrt{n}$ with probability at least $1 - \exp(-\Theta(n))$ over the random noise $\mathbf{w}$.

Suppose that for all $t_1 \in [0, t]$ with $t \in [0, T-1]$, $\mathbf{u}(t_1) = \mathbf{v}(t_1) + \mathbf{e}(t_1)$ where $\mathbf{v}(t_1) \in \mathcal{V}_{t_1}$, and $\mathbf{e}(t_1) = \vec{\mathbf{e}}_1(t_1) + \vec{\mathbf{e}}_2(t_1)$ with $\mathbf{v}(t_1) \in \mathcal{V}_{t_1}$ and $\mathbf{e}(t_1) \in \mathcal{E}_{t_1, \tau}$, and $\|\mathbf{u}(t_1)\|_2 \leq c_{\mathbf{u}} \sqrt{n}$ for all $t_1 \in [0, t]$. Then it follows from Lemma C.10 that the recursion $\mathbf{u}(t'+1) = \left(\mathbf{I} - \eta \mathbf{K}_n^{(\mathrm{attn})}\right) \mathbf{u}(t') + \mathbf{E}(t'+1)$ holds for all $t' \in [0, t]$. As a result, we have

$$\mathbf{u}(t+1) = \left(\mathbf{I} - \eta \mathbf{K}_n^{(\mathrm{attn})}\right) \mathbf{u}(t) + \mathbf{E}(t+1)$$

$$= -\left(\mathbf{I} - \eta \mathbf{K}_n^{(\mathrm{attn})}\right)^{t+1} f^*(\mathbf{S}) - \left(\mathbf{I} - \eta \mathbf{K}_n^{(\mathrm{attn})}\right)^{t+1} \mathbf{w} + \sum_{t'=1}^{t+1} \left(\mathbf{I} - \eta \mathbf{K}_n^{(\mathrm{attn})}\right)^{t+1-t'} \mathbf{E}(t')$$

$$= \mathbf{v}(t+1) + \mathbf{e}(t+1), \tag{50}$$

where $\mathbf{v}(t+1)$ and $\mathbf{e}(t+1)$ are defined as

$$\mathbf{v}(t+1) := -\left(\mathbf{I} - \eta \mathbf{K}_n^{(\mathrm{attn})}\right)^{t+1} f^*(\mathbf{S}) \in \mathcal{V}_{t+1}, \tag{51}$$

$$\mathbf{e}(t+1) := \underbrace{-\left(\mathbf{I} - \eta \mathbf{K}_n^{(\mathrm{attn})}\right)^{t+1} \mathbf{w}}_{\vec{\mathbf{e}}_1(t+1)} + \underbrace{\sum_{t'=1}^{t+1} \left(\mathbf{I} - \eta \mathbf{K}_n^{(\mathrm{attn})}\right)^{t+1-t'} \mathbf{E}(t')}_{\vec{\mathbf{e}}_2(t+1)}. \tag{52}$$

We now prove the upper bound for $\vec{\mathbf{e}}_2(t+1)$. With $\eta \in (0,1)$, we have $\left\|\mathbf{I} - \eta \mathbf{K}_n^{(\mathrm{attn})}\right\|_2 \in (0,1)$. It follows that

$$\left\|\vec{\mathbf{e}}_2(t+1)\right\|_2 \leq \sum_{t'=1}^{t+1} \left\|\mathbf{I} - \eta \mathbf{K}_n^{(\mathrm{attn})}\right\|_2^{t+1-t'} \|\mathbf{E}(t')\|_2 \leq \tau \sqrt{n}, \tag{53}$$

where the last inequality follows from the fact that $\|\mathbf{E}(t)\|_2 \leq \mathbf{E}_{m,\eta,\delta} \leq \tau \sqrt{n}/T$ for all $t \in [T]$. It follows that $\mathbf{e}(t+1) \in \mathcal{E}_{t+1,\tau}$. Also, it follows from Lemma C.9 that

$$\|\mathbf{u}(t+1)\|_2 \leq \|\mathbf{v}(t+1)\|_2 + \left\|\vec{\mathbf{e}}_1(t+1)\right\|_2 + \left\|\vec{\mathbf{e}}_2(t+1)\right\|_2 \leq \left(\frac{\mu_0}{\sqrt{2e\eta}} + \sigma_0 + \tau + 1\right) \sqrt{n}$$

$$\leq c_{\mathbf{u}} \sqrt{n}.$$

The above inequality completes the induction step, which also completes the proof. It is noted that $\left\|\vec{\mathbf{w}}_r(t) - \vec{\mathbf{w}}_r(0)\right\|_2 \leq R$ and $|a_r(t) - a_r(0)| \leq 2R$ hold for all $t \in [T]$ by Lemma C.12.

$\qquad\qquad\qquad\qquad\qquad\qquad\qquad\qquad\qquad\qquad\qquad\qquad\qquad\qquad\qquad\qquad\qquad\qquad\quad \square$

**Proof of Theorem C.4.** In this proof, we abbreviate $f_t$ as $f$ and $\mathbf{W}(t)$ as $\mathbf{W}$. It follows from Lemma C.3 and its proof that conditioned on an event $\Omega$ with probability at least $1 - 1/n - \delta - \exp(-\Theta(n))$, $f \in \mathcal{F}_{\mathrm{NN}}(\mathbf{S}, \mathbf{W}(0), T)$ with $\mathbf{W}(0) \in \mathcal{W}_0$. Moreover, $f = f(\mathbf{W}, \boldsymbol{a}, \cdot)$ with $(\mathbf{W}, \boldsymbol{a}) = \left(\left\{\vec{\mathbf{w}}_r\right\}_{r=1}^m, \boldsymbol{a}\right) \in \mathcal{W}(\mathbf{S}, \mathbf{W}(0), T)$, and $\mathrm{vec}(\mathbf{W}) = \mathrm{vec}(\mathbf{W_S}) = \mathrm{vec}(\mathbf{W}(0)) - \sum_{t'=0}^{t-1} \eta/n \cdot \mathbf{Z_S}(t') \mathbf{u}(t')$ for some $t \in [T]$, where $\mathbf{u}(t') \in \mathbb{R}^n$, $\mathbf{u}(t') = \mathbf{v}(t') + \mathbf{e}(t')$ with $\mathbf{v}(t') \in \mathcal{V}_{t'}$ and $\mathbf{e}(t') \in \mathcal{E}_{t',\tau}$ for all $t' \in [0, t-1]$. It also follows from Lemma C.3 that conditioned on $\Omega$, $\left\|\vec{\mathbf{w}}_r(t) - \vec{\mathbf{w}}_r(0)\right\|_2 \leq R$, $|a_r(t) - a_r(0)| \leq 2R$ hold for all $t \in [T]$.

$\vec{\mathbf{w}}_r, \boldsymbol{a}$ are expressed as

$$\vec{\mathbf{w}}_r = \vec{\mathbf{w}}_{\mathbf{S},r}(t) = \vec{\mathbf{w}}_r(0) - \sum_{t'=0}^{t-1} \frac{\eta}{n} \left[\mathbf{Z_S}(t')\right]_{[(r-1)d+1:rd]} \mathbf{u}(t'), \tag{54}$$

$$\boldsymbol{a} = \boldsymbol{a}(t) = \sum_{t'=0}^{t-1} -\frac{\eta}{n\sqrt{m}} \mathbf{A}\boldsymbol{\sigma}(\mathbf{W}(t'), \mathbf{S})\mathbf{u}(t'), \tag{55}$$

where the notation $\vec{\mathbf{w}}_{\mathbf{S},r}$ emphasizes that $\vec{\mathbf{w}}_r$ depends on the training features $\mathbf{S}$. Let $\boldsymbol{a} = \boldsymbol{a}(t) = [a_1(t), \ldots, a_m(t)]$, we approximate $f(\mathbf{W}, \boldsymbol{a}, \mathbf{x})$ by

$$g(\mathbf{x}) := \frac{1}{\sqrt{m}} \sum_{r'=1}^{m} \sum_{r=1}^{m} a_r(t) \sigma\left(\vec{\mathbf{w}}_{r'}(0)^\top \mathbf{x}\right) \mathbf{A}_{r'r}.$$

We have

$$|f(\mathbf{W}, \boldsymbol{a}, \mathbf{x}) - g(\mathbf{x})|$$

$$= \frac{1}{\sqrt{m}} \left| \sum_{r'=1}^{m} \sum_{r=1}^{m} a_r(t) \sigma\left(\vec{\mathbf{w}}_{r'}(t)^\top \mathbf{x}\right) \mathbf{A}_{r'r} - \sum_{r'=1}^{m} \sum_{r=1}^{m} a_r(t) \mathbb{1}_{\left\{\vec{\mathbf{w}}_{r'}(0)^\top \mathbf{x} \geq 0\right\}} \vec{\mathbf{w}}_{r'}(0)^\top \mathbf{x} \mathbf{A}_{r'r} \right|$$

$$\leq \frac{1}{\sqrt{m}} \sum_{r'=1}^{m} \sum_{r=1}^{m} |a_r| \left\| \vec{\mathbf{w}}_{r'}(t) - \vec{\mathbf{w}}_{r'}(0) \right\|_2 \mathbf{A}_{r'r}$$

$$\leq \frac{1}{\sqrt{m}} \cdot 2R\sqrt{m} \cdot \|\mathbf{A}\|_2 \cdot R\sqrt{m} \leq 2R^2\sqrt{m}, \forall \mathbf{x} \in \mathcal{X}, \tag{56}$$

where last inequality follows from $\|\mathbf{A}\|_2 \leq 1$ due to (83) in the proof of Lemma C.10 with $\mathbf{W}(0) \in \mathcal{W}_0$ and $m \gtrsim (d\log m)^3$. Using (55), $g(\mathbf{x})$ is expressed as

$$g(\mathbf{x}) = -\sum_{t'=0}^{t-1} \frac{\eta}{nm} \underbrace{\sum_{r'=1}^{m} \sum_{r=1}^{m} \sigma\left(\vec{\mathbf{w}}_{r'}(0)^\top \mathbf{x}\right) \mathbf{A}_{r'r} \mathbf{A}_r \boldsymbol{\sigma}(\mathbf{W}(t'), \mathbf{S}) \mathbf{u}(t')}_{:=G_{t'}(\mathbf{x})}. \tag{57}$$

For each $G_{t'}$ on the RHS of (57), we have

$$G_{t'}(\mathbf{x}) = \frac{\eta}{nm} \underbrace{\sum_{r'=1}^{m} \sum_{r=1}^{m} \sigma\left(\vec{\mathbf{w}}_{r'}(0)^\top \mathbf{x}\right) \mathbf{A}_{r'r} \mathbf{A}_r \boldsymbol{\sigma}(\mathbf{W}(0), \mathbf{S}) \mathbf{u}(t')}_{:=D(\mathbf{x},t')}$$

$$+ \frac{\eta}{nm} \underbrace{\sum_{r'=1}^{m} \sum_{r=1}^{m} \sigma\left(\vec{\mathbf{w}}_{r'}(0)^\top \mathbf{x}\right) \mathbf{A}_{r'r} \mathbf{A}_r \left(\boldsymbol{\sigma}(\mathbf{W}(t'), \mathbf{S}) - \boldsymbol{\sigma}(\mathbf{W}(0), \mathbf{S})\right) \mathbf{u}(t')}_{:=E_1(\mathbf{x},t')}$$

$$= \frac{\eta}{n} \sum_{j=1}^{n} K^{(\text{attn})}(\mathbf{x}, \vec{\mathbf{x}}_j) \mathbf{u}_j(t') + E_1(\mathbf{x}, t') + E_2(\mathbf{x}, t'), \tag{58}$$

where $E_2(\mathbf{x}, t') := D(\mathbf{x}, t') - \eta/n \cdot \sum_{j=1}^{n} K^{(\text{attn})}(\mathbf{x}, \vec{\mathbf{x}}_j) \mathbf{u}_j(t')$. We now analyze each term on the RHS of (58). It follows from Lemma C.16 that

$$\|E_2(\mathbf{x}, t')\|_\infty \lesssim C_1(m, d, 1/n) + \sqrt{\frac{\log(n/\delta)}{N}}. \tag{59}$$

Let $h(\cdot, t') \colon \mathcal{X} \to \mathbb{R}$ be defined by $h(\mathbf{x}, t') := \frac{\eta}{n} \sum_{j=1}^{n} K^{(\text{attn})}(\mathbf{x}, \vec{\mathbf{x}}_j) \mathbf{u}_j(t')$, then $h(\cdot, t') \in \mathcal{H}_{K^{(\text{attn})}}$ for each $t' \in [0, t-1]$. We further define

$$h_t(\cdot) := -\sum_{t'=0}^{t-1} h(\cdot, t') \in \mathcal{H}_{K^{(\text{attn})}}. \tag{60}$$

We note that $E_1(\mathbf{x}, t') = \eta/(nm) \cdot \boldsymbol{\sigma}^\top(\mathbf{W}(0), \mathbf{x}) \mathbf{A}^2 \left(\boldsymbol{\sigma}(\mathbf{W}(t'), \mathbf{S}) - \boldsymbol{\sigma}(\mathbf{W}(0), \mathbf{S})\right) \mathbf{u}(t')$, and it follows that

$$\|E_1(\mathbf{x}, t')\|_\infty \leq \frac{\eta}{nm} \left\| \boldsymbol{\sigma}^\top(\mathbf{W}(0), \mathbf{x}) \right\|_2 \|\mathbf{A}\|_2^2 \|\boldsymbol{\sigma}(\mathbf{W}(t'), \mathbf{S}) - \boldsymbol{\sigma}(\mathbf{W}(0), \mathbf{S})\|_2 \|\mathbf{u}(t')\|_2$$

$$\overset{\textcircled{1}}{\leq} \frac{\eta}{nm} \cdot \sqrt{m} \cdot \sqrt{nm}R \cdot c_{\mathbf{u}}\sqrt{n} = \eta c_{\mathbf{u}} R. \tag{61}$$

Since $\mathbf{W}(0) \in \mathcal{W}_0$ and $m \gtrsim (d \log m)^3$, $\left\| \boldsymbol{\sigma}^\top (\mathbf{W}(0), \mathbf{x}) \right\|_2 \leq \sqrt{m}$. It follows from (83) in the proof of Lemma C.10 that $\|\mathbf{A}\|_2 \leq 1$. Because $\left\| \overrightarrow{\mathbf{w}}_r(t) - \overrightarrow{\mathbf{w}}_r(0) \right\|_2 \leq R$ for all $t \in [T]$, we have $\|\boldsymbol{\sigma}(\mathbf{W}(t'), \mathbf{S}) - \boldsymbol{\sigma}(\mathbf{W}(0), \mathbf{S})\|_2 \leq \sqrt{nm}R$. As a result, ① holds.

It follows from (58), (59), and (61), for any $t' \in [0, t-1]$,

$$\|G_{t'}(\mathbf{x}) - h(\mathbf{x}, t')\|_\infty \leq \|E_1\|_\infty + \|E_2\|_\infty \lesssim \eta c_{\mathbf{u}} R + C_1(m, d, 1/n) + \sqrt{\frac{\log(n/\delta)}{N}}. \quad (62)$$

Define $e_t(\cdot) = f(\mathbf{W}, \boldsymbol{a}, \cdot) - h_t(\cdot)$. It then follows from (56), (57), and (62) that

$$\|e_t\|_\infty \leq \|f(\mathbf{W}, \boldsymbol{a}, \mathbf{x}) - g(\mathbf{x})\|_\infty + \|g(\mathbf{x}) - h_t(\mathbf{x})\|_\infty$$

$$\leq \|f(\mathbf{W}, \boldsymbol{a}, \mathbf{x}) - g(\mathbf{x})\|_\infty + \sum_{t'=0}^{t-1} \|G_{t'}(\mathbf{x}) - h(\mathbf{x}, t')\|_\infty$$

$$\overset{④}{\lesssim} 2R^2\sqrt{m} + T\left( \eta c_{\mathbf{u}} R + C_1(m, d, 1/n) + \sqrt{\frac{\log(n/\delta)}{N}} \right) := \Delta_{m,n,\delta,T}. \quad (63)$$

We now give an estimate for $\Delta_{m,n,\delta,T}$. With $\mathbf{W}(0) \in \mathcal{W}_0$, $C_1(m, d, 1/n) \lesssim d \log m \sqrt{\frac{d \log m}{m}}$. As a result, plugging $R = \eta c_{\mathbf{u}} \sqrt{2d + 3\log(2mn)} T / \sqrt{m}$ on the RHS of (63), we have

$$\Delta_{m,n,\delta,T} \lesssim \frac{T^2(d + \log m) + (d \log m)^{3/2}T}{\sqrt{m}} + T\sqrt{\frac{\log(n/\delta)}{N}}.$$

By direct calculations, for any $w > 0$, when

$$m \gtrsim \max\left\{ T^2(d \log m)^3/w^2, T^4(d^2 + \log^2 m)/w^2, n \right\}, \quad (64)$$

$$N \gtrsim T^2 \log(n/\delta)/w^2, \quad (65)$$

we have $\Delta_{m,n,\delta,T} \lesssim w$.

It follows from Lemma C.15 that with probability at least $1 - \exp\left( -\Theta(n\widehat{\varepsilon}_n^2) \right)$ over the random noise $\mathbf{w}$, $\|h_t\|_{\mathcal{H}_{K^{(\text{attn})}}} \leq B_h$, where $B_h$ is defined in (29), and $\tau$ is required to satisfy $\tau \leq 1/(\eta T)$. Lemma C.3 requires that

$$m \gtrsim \max\left\{ T^2(d \log m)^3/\tau^2, T^6(d^2 + \log^2 m)/\tau^2, n \right\},$$

$$N \gtrsim T^2 \log(n/\delta)/\tau^2.$$

As a result, we have $m \gtrsim \max\left\{ T^4(d \log m)^3, T^8(d^2 + \log^2 m), n \right\}$ and $N \gtrsim T^4 \log(n/\delta)$, which lead to the conditions on $m, N$, (39) and (40), when combined with (64)-(65).

$\square$

**Proof of Theorem C.6.** We first remark that the conditions on $m, N$ are required by Theorem C.4. It also follows from Theorem C.4 that conditioned on an event $\Omega$ with probability at least $1 - 1/n - \delta - \exp\left( -\Theta(n) \right) - \exp\left( -\Theta(n\widehat{\varepsilon}_n^2) \right)$ over $\mathbf{w}$ and $\mathbf{Q}$, we have $(\mathbf{W}(t), \boldsymbol{a}(t)) \in \mathcal{W}(\mathbf{S}, \mathbf{Q}, \mathbf{W}(0), T)$, and

$$f(\mathbf{W}(t), \boldsymbol{a}(t), \cdot) = f_t = h + e \in \mathcal{F}(B_h, w)$$

with $h \in \mathcal{H}_{K^{(\text{attn})}}(B_h)$ and $\|e\|_\infty \leq w$. The proof then follows a similar strategy to that of (Yang & Li, 2024, Theorem VI.5) and (Yang, 2025, Theorem C.10). We then derive the sharp upper bound for $\mathbb{E}_P\left[ (f_t - f^*)^2 \right]$ by applying Theorem A.3 to the function class

$$\mathcal{F} = \left\{ F = (f - f^*)^2 : f \in \mathcal{F}(B_h, w) \right\}.$$

With $B_0 = B_h + 1 + \mu_0 \geq B_h + w + \mu_0$, we have $\|F\|_\infty \leq B_0^2$ with $F \in \mathcal{F}$, so that $\mathbb{E}_P\left[ F^2 \right] \leq B_0^2 \mathbb{E}_P[F]$. Let $T(F) = B_0^2 \mathbb{E}_P[F]$ for $F \in \mathcal{F}$. Then $\text{Var}[F] \leq \mathbb{E}_P\left[ F^2 \right] \leq T(F) = B_0^2 \mathbb{E}_P[F]$. We have

$$\mathfrak{R}\left( \{F \in \mathcal{F} : T(F) \leq r\} \right) = \mathfrak{R}\left( \left\{ (f - f^*)^2 : f \in \mathcal{F}(B_h, w), \mathbb{E}_P\left[ (f - f^*)^2 \right] \leq \frac{r}{B_0^2} \right\} \right)$$

$$\overset{①}{\leq} 2B_0 \mathfrak{R}\left(\left\{f - f^* \colon f \in \mathcal{F}(B_h, w), \mathbb{E}_P\left[(f - f^*)^2\right] \leq \frac{r}{B_0^2}\right\}\right)$$

$$\overset{②}{\leq} 4B_0 \mathfrak{R}\left(\left\{f \in \mathcal{F}(B_h, w) \colon \mathbb{E}_P\left[f^2\right] \leq \frac{r}{4B_0^2}\right\}\right), \tag{66}$$

where ① is due to the contraction property of Rademacher complexity in Theorem A.2. Since $f^* \in \mathcal{F}(B_h, w)$, $f \in \mathcal{F}(B_h, w)$, we have $\frac{f - f^*}{2} \in \mathcal{F}(B_h, w)$ due to the fact that $\mathcal{F}(B_h, w)$ is symmetric and convex, and it follows that ② holds.

It follows from (66) and Lemma C.5 that

$$B_0^2 \mathfrak{R}\left(\{F \in \mathcal{F} \colon T(F) \leq r\}\right) \leq 4B_0^3 \mathfrak{R}\left(\left\{f \colon f \in \mathcal{F}(B_h, w), \mathbb{E}_P\left[f^2\right] \leq \frac{r}{4B_0^2}\right\}\right)$$

$$\leq 4B_0^3 \varphi_{B_h, w}\left(\frac{r}{4B_0^2}\right) := \psi(r). \tag{67}$$

It follows from the definition of $\varphi_{B_h, w}$ in (43) and the Cauchy-Schwarz inequality that

$$\psi(r) = 4B_0^3 \min_{Q \colon Q \geq 0}\left(\left(\frac{\sqrt{r}}{2B_0} + w\right)\sqrt{\frac{Q}{n}} + B_h\left(\frac{\sum\limits_{q=Q+1}^{\infty} \lambda_q}{n}\right)^{1/2}\right) + 4B_0^3 w$$

$$\leq 4B_0^3 B_h \min_{Q \colon Q \geq 0}\left(\sqrt{\frac{Qr}{n}} + \left(\frac{\sum\limits_{q=Q+1}^{\infty} \lambda_q}{n}\right)^{1/2}\right) + 8B_0^3 w$$

$$\leq \frac{4\sqrt{2}B_0^3 B_h}{\sigma_0} \cdot \sigma_0 R_{K^{(\text{attn})}}(\sqrt{r}) + 8B_0^3 w := \psi_1(r),$$

where the last inequality follows from the definition of the kernel complexity. It can be verified that $\psi_1(r)$ is a sub-root function. Let the fixed point of $\psi_1(r)$ be $r_1^*$. Because the fixed point of $\sigma_0 R_K(\sqrt{r})$ as a function of $r$ is $\varepsilon_n^2$, it follows from Lemma C.20 that

$$r_1^* \leq \max\left\{\frac{32B_0^6 B_h^2}{\sigma_0^2}, 1\right\}\varepsilon_n^2 + 16B_0^3 w. \tag{68}$$

It then follows from Theorem A.3 with $K_0 = 2$ that with probability at least $1 - \exp(-x)$,

$$\mathbb{E}_P\left[(f_t - f^*)^2\right] - 2\mathbb{E}_{P_n}\left[(f_t - f^*)^2\right] \lesssim r_1^* + \frac{x}{n}.$$

Letting $x = n\varepsilon_n^2$, then plugging the upper bound (68) for $r_1^*$ in the above inequality leads to

$$\mathbb{E}_P\left[(f_t - f^*)^2\right] - 2\mathbb{E}_{P_n}\left[(f_t - f^*)^2\right] \lesssim \varepsilon_n^2 + w, \tag{69}$$

which proves (44).

$$\square$$

**Proof of Theorem C.7.** We have

$$f_t(\mathbf{S}) = f^*(\mathbf{S}) + \mathbf{w} + \mathbf{v}(t) + \mathbf{e}(t), \tag{70}$$

where $\mathbf{v}(t) \in \mathcal{V}_t$, $\mathbf{e}(t) \in \mathcal{E}_{t,\tau}$, $\vec{\mathbf{e}}(t) = \vec{\mathbf{e}}_1(t) + \vec{\mathbf{e}}_2(t)$ with $\vec{\mathbf{e}}_1(t) = -\left(\mathbf{I}_n - \eta\mathbf{K}_n^{(\text{attn})}\right)^t \mathbf{w}$ and $\left\|\vec{\mathbf{e}}_2(t)\right\|_2 \lesssim \sqrt{n}\tau$. It follows from (70) that

$$\mathbb{E}_{P_n}\left[(f_t - f^*)^2\right] = \frac{1}{n}\|f_t(\mathbf{S}) - f^*(\mathbf{S})\|_2^2 = \frac{1}{n}\|\mathbf{v}(t) + \mathbf{w} + \mathbf{e}(t)\|_2^2$$

$$= \frac{1}{n}\left\|-\left(\mathbf{I} - \eta\mathbf{K}_n^{(\text{attn})}\right)^t f^*(\mathbf{S}) + \left(\mathbf{I}_n - \left(\mathbf{I}_n - \eta\mathbf{K}_n^{(\text{attn})}\right)^t\right)\mathbf{w} + \vec{\mathbf{e}}_2(t)\right\|_2^2$$

$$\overset{\text{\textcircled{1}}}{\leq} \frac{3}{n} \sum_{i=1}^{n} \left(1 - \eta\widehat{\lambda}_i^{(\text{attn})}\right)^{2t} \left[\mathbf{U}^\top f^*(\mathbf{S})\right]_i^2 + \frac{3}{n} \sum_{i=1}^{n} \left(1 - \left(1 - \eta\widehat{\lambda}_i^{(\text{attn})}\right)^t\right)^2 \left[\mathbf{U}^\top \mathbf{w}\right]_i^2 + \frac{3}{n}\left\|\overrightarrow{\mathbf{e}}_2(t)\right\|_2^2$$

$$\overset{\text{\textcircled{2}}}{\leq} \frac{3\mu_0^2}{2e\eta t} + \frac{3}{n} \sum_{i=1}^{n} \left(1 - \left(1 - \eta\widehat{\lambda}_i^{(\text{attn})}\right)^t\right)^2 \left[\mathbf{U}^\top \mathbf{w}\right]_i^2 + 3\tau^2$$

$$\leq \frac{3}{\eta t}\left(\frac{\mu_0^2}{2e} + \frac{1}{\eta T}\right) + 3 \cdot \underbrace{\frac{1}{n} \sum_{i=1}^{n} \left(1 - \left(1 - \eta\widehat{\lambda}_i^{(\text{attn})}\right)^t\right)^2 \left[\mathbf{U}^\top \mathbf{w}\right]_i^2}_{:=E_\varepsilon} \tag{71}$$

Here \textcircled{1} follows from the Cauchy-Schwarz inequality, \textcircled{2} follows from (78) in the proof of Lemma C.9, and \textcircled{3} follows from the conditions on $N, \tau$ in (45).

We then derive the upper bound for $E_\varepsilon$ on the RHS of (71). We define the diagonal matrix $\mathbf{R} \in \mathbb{R}^{n \times n}$ with $\mathbf{R}_{ii} = \left(1 - (1 - \eta\lambda_i)^t\right)^2$. Then we have $E_\varepsilon = 1/n \cdot \text{tr}\left(\mathbf{U}\mathbf{R}\mathbf{U}^\top \mathbf{w}\mathbf{w}^\top\right)$. It follows from (Wright, 1973) that

$$\Pr\left[1/n \cdot \text{tr}\left(\mathbf{U}\mathbf{R}\mathbf{U}^\top \mathbf{w}\mathbf{w}^\top\right) - \mathbb{E}\left[1/n \cdot \text{tr}\left(\mathbf{U}\mathbf{R}\mathbf{U}^\top \mathbf{w}\mathbf{w}^\top\right)\right] \geq u\right]$$
$$\leq \exp\left(-c\min\left\{nu/\|\mathbf{R}\|_2, n^2 u^2/\|\mathbf{R}\|_\text{F}^2\right\}\right) \tag{72}$$

holds for all $u > 0$, and $c$ is a positive constant. With $\eta_t = \eta t$ for all $t \geq 0$, we have

$$\mathbb{E}\left[1/n \cdot \text{tr}\left(\mathbf{U}\mathbf{R}\mathbf{U}^\top \mathbf{w}\mathbf{w}^\top\right)\right] \leq \frac{\sigma_0^2}{n} \sum_{i=1}^{n} \left(1 - \left(1 - \eta\widehat{\lambda}_i^{(\text{attn})}\right)^t\right)^2 \overset{\text{\textcircled{1}}}{\leq} \frac{\sigma_0^2}{n} \sum_{i=1}^{n} \min\left\{1, \eta_t^2(\widehat{\lambda}_i^{(\text{attn})})^2\right\}$$

$$\leq \frac{\sigma_0^2 \eta_t}{n} \sum_{i=1}^{n} \min\left\{\frac{1}{\eta_t}, \eta_t(\widehat{\lambda}_i^{(\text{attn})})^2\right\} \overset{\text{\textcircled{2}}}{\leq} \frac{\sigma_0^2 \eta_t}{n} \sum_{i=1}^{n} \min\left\{\frac{1}{\eta_t}, \widehat{\lambda}_i^{(\text{attn})}\right\} = \sigma_0^2 \eta_t \widehat{R}_{K^{(\text{attn})}}^2(\sqrt{1/\eta_t}) \leq \frac{1}{\eta_t}. \tag{73}$$

Here \textcircled{1} follows from the fact that $(1 - \eta\widehat{\lambda}_i^{(\text{attn})})^t \geq \max\left\{0, 1 - t\eta\widehat{\lambda}_i^{(\text{attn})}\right\}$, and \textcircled{2} follows from $\min\{a, b\} \leq \sqrt{ab}$ for any nonnegative numbers $a, b$. Because $t \leq T \leq \widehat{T}$, we have $R_{K^{(\text{attn})}}(\sqrt{1/\eta_t}) \leq 1/(\sigma\eta_t)$, so the last inequality holds.

Moreover, we have the upper bounds for $\|\mathbf{R}\|_2$ and $\|\mathbf{R}\|_\text{F}$ as follows. First, we have

$$\|\mathbf{R}\|_2 \leq \max_{i \in [n]}\left(1 - \left(1 - \eta\widehat{\lambda}_i^{(\text{attn})}\right)^t\right)^2 \leq \min\left\{1, \eta_t^2(\widehat{\lambda}_i^{(\text{attn})})^2\right\} \leq 1. \tag{74}$$

We also have

$$\frac{1}{n}\|\mathbf{R}\|_\text{F}^2 = \frac{1}{n} \sum_{i=1}^{n}\left(1 - \left(1 - \eta\widehat{\lambda}_i^{(\text{attn})}\right)^t\right)^4 \leq \frac{\eta_t}{n} \sum_{i=1}^{n} \min\left\{\frac{1}{\eta_t}, \eta_t^3(\widehat{\lambda}_i^{(\text{attn})})^4\right\}$$

$$\leq \frac{\eta_t}{n} \sum_{i=1}^{n} \min\left\{\widehat{\lambda}_i^{(\text{attn})}, \frac{1}{\eta_t}\right\} = \eta_t \widehat{R}_{K^{(\text{attn})}}^2(\sqrt{1/\eta_t}) \leq \frac{1}{\sigma_0^2 \eta_t}. \tag{75}$$

Combining (72)-(75), we have

$$\Pr\left[1/n \cdot \text{tr}\left(\mathbf{U}\mathbf{R}\mathbf{U}^\top \mathbf{w}\mathbf{w}^\top\right) - \mathbb{E}\left[1/n \cdot \text{tr}\left(\mathbf{U}\mathbf{R}\mathbf{U}^\top \mathbf{w}\mathbf{w}^\top\right)\right] \geq u\right] \leq \exp\left(-cn\min\left\{u, u^2\sigma_0^2\eta_t\right\}\right).$$

Let $u = 1/(\eta t)$ in the above inequality, we have

$$\exp\left(-cn\min\left\{u, u^2\sigma_0^2\eta_t\right\}\right) = \exp\left(-c'n/\eta_t\right) \leq \exp\left(-c'n\widehat{\varepsilon}_n^2\right)$$

where $c' = c\min\left\{1, \sigma_0^2\right\}$, and the last inequality is due to the fact that $1/\eta_t \geq \widehat{\varepsilon}_n^2$ since $t \leq T \leq \widehat{T}$. It follows that with probability at least $1 - \exp\left(-\Theta(n\widehat{\varepsilon}_n^2)\right)$,

$$E_\varepsilon \leq u + \frac{1}{\eta_t} = \frac{2}{\eta_t}. \tag{76}$$

It then follows from (71), (72)-(76) that

$$\mathbb{E}_{P_n}\left[(f_t - f^*)^2\right] \leq \frac{3}{\eta t}\left(\frac{\mu_0^2}{2e} + \frac{1}{\eta T} + 2\right)$$

with probability at least $1 - \exp\left(-c'n\widehat{\varepsilon}_n^2\right)$.

$\square$

## C.6 Proof of the Lemmas Required for the Proofs in Section C.5

**Lemma C.9.** Let $t \in [0, T]$, $\mathbf{v} = -\left(\mathbf{I} - \eta\mathbf{K}_n^{(\text{attn})}\right)^t f^*(\mathbf{S})$, $\mathbf{e} = -\left(\mathbf{I} - \eta\mathbf{K}_n^{(\text{attn})}\right)^t \mathbf{w}$, and $\eta = \Theta(1) \in (0,1)$. Suppose $\delta \in (0, 1/2)$, then with probability at least $1 - \exp\left(-\Theta(n)\right)$ over the random noise $\mathbf{w}$,

$$\|\mathbf{v}\|_2 + \|\mathbf{e}\|_2 \leq \left(\frac{\mu_0}{\min\left\{\sqrt{2e\eta}, 1\right\}} + \sigma_0 + 1\right) \cdot \sqrt{n}. \tag{77}$$

*Proof.* When $t \geq 1$, we have

$$\|\mathbf{v}\|_2^2 = \sum_{i=1}^{n}\left(1 - \eta\widehat{\lambda}_i\right)^{2t}\left[\mathbf{U}^\top f^*(\mathbf{S})\right]_i^2 \overset{\text{①}}{\leq} \sum_{i=1}^{n}\frac{1}{2e\eta\widehat{\lambda}_i t}\left[\mathbf{U}^\top f^*(\mathbf{S})\right]_i^2 \overset{\text{②}}{\leq} \frac{n\mu_0^2}{2e\eta t} \leq \frac{\mu_0^2}{2e\eta} \cdot n. \tag{78}$$

Here ① follows from Lemma C.18, ② follows by Lemma C.17. Moreover, it follows from the concentration inequality about quadratic forms of sub-Gaussian random variables in (Wright, 1973) that $\Pr\left[\|\mathbf{w}\|_2^2 - \mathbb{E}\left[\|\mathbf{w}\|_2^2\right] > n\right] \leq \exp\left(-\Theta(n)\right)$, so that $\|\mathbf{e}\|_2 \leq \|\mathbf{w}\|_2 \leq \sqrt{\mathbb{E}\left[\|\mathbf{w}\|_2^2\right]} + \sqrt{n} = \sqrt{n}(\sigma_0 + 1)$ with probability at least $1 - \exp\left(-\Theta(n)\right)$. As a result, (77) follows from this inequality and (78) for $t \geq 1$. When $t = 0$, $\|\mathbf{v}\|_2 \leq \mu_0\sqrt{n}$, so that (77) still holds.

$\square$

**Lemma C.10.** Let $\eta \in (0, 1)$, $0 \leq t \leq T - 1$ for $T \geq 1$, and suppose that $\|\widehat{\mathbf{y}}(t') - \mathbf{y}\|_2 \leq c_{\mathbf{u}}\sqrt{n}$ holds for all $0 \leq t' \leq t$, $N \gtrsim (d\log N)^3$, $m \gtrsim (d\log m)^3$, and the random initialization $\mathbf{W}(0) \in \mathcal{W}_0$. Let $\delta \in (0, 1)$, then with probability at least $1 - 1/n - \delta$ over $\mathbf{Q}$ and $\mathbf{W}(0)$,

$$\widehat{\mathbf{y}}(t+1) - \mathbf{y} = \left(\mathbf{I} - \eta\mathbf{K}_n^{(\text{attn})}\right)(\widehat{\mathbf{y}}(t) - \mathbf{y}) + \mathbf{E}(t+1), \tag{79}$$

where $\|\mathbf{E}(t+1)\|_2 \lesssim \mathbf{E}_{m,\eta,\delta}$, and $\mathbf{E}_{m,\eta,\delta}$ is defined by

$$\mathbf{E}_{m,\eta,\delta} := R^2\sqrt{mn} + \eta c_{\mathbf{u}}\sqrt{n}\left(R + C_1(m, d, 1/n) + \sqrt{\frac{\log(n/\delta)}{N}}\right). \tag{80}$$

*Proof.* Because $\|\widehat{\mathbf{y}}(t') - \mathbf{y}\|_2 \leq \sqrt{n}c_{\mathbf{u}}$ holds for all $t' \in [0, t]$, it follows from Lemma C.12 that for every $t' \in [0, t+1]$ and $r \in [m]$,

$$\left\|\vec{\mathbf{w}}_{\mathbf{S},r}(t') - \vec{\mathbf{w}}_r(0)\right\|_2 \leq R, \quad |a_r(t')| \leq 2R. \tag{81}$$

We have

$$\widehat{\mathbf{y}}_i(t+1) - \widehat{\mathbf{y}}_i(t) = \frac{1}{\sqrt{m}}\sum_{r'=1}^{m}\sum_{r=1}^{m}(a_r(t+1) - a_r(t))\,\sigma\left(\vec{\mathbf{w}}_{r'}(t)^\top\vec{\mathbf{x}}_i\right)\mathbf{A}_{r'r}$$

$$+ \underbrace{\frac{1}{\sqrt{m}}\sum_{r'=1}^{m}\sum_{r=1}^{m}a_r(t+1)\left(\sigma\left(\vec{\mathbf{w}}_{r'}(t+1)^\top\vec{\mathbf{x}}_i\right) - \sigma\left(\vec{\mathbf{w}}_{r'}(t)^\top\vec{\mathbf{x}}_i\right)\right)\mathbf{A}_{r'r}}_{:=\mathbf{E}_i^{(1)}}$$

$$= -\frac{\eta}{nm} \sum_{r'=1}^{m} \sum_{r=1}^{m} \sigma\left(\vec{\mathbf{w}}_{r'}(t)^\top \vec{\mathbf{x}}_i\right) \mathbf{A}_{r'r} [\mathbf{A}]_r \, \boldsymbol{\sigma}(\mathbf{W}(t), \mathbf{S})(\widehat{\mathbf{y}}(t) - \mathbf{y}) + \mathbf{E}_i^{(1)}$$

$$= \underbrace{-\frac{\eta}{nm} \boldsymbol{\sigma}^\top(\mathbf{W}(0), \vec{\mathbf{x}}_i) \mathbf{A}^2 \boldsymbol{\sigma}(\mathbf{W}(t), \mathbf{S})(\widehat{\mathbf{y}}(t) - \mathbf{y})}_{:=\mathbf{D}_i^{(1)}} + \mathbf{E}_i^{(1)}, \tag{82}$$

and $\mathbf{D}^{(1)}, \mathbf{E}^{(1)} \in \mathbb{R}^n$ are vectors with their $i$-th element being $\mathbf{D}_i^{(1)}$ and $\mathbf{E}_i^{(1)}$ defined on the RHS of (82). Now we derive the upper bound for $\mathbf{E}_i^{(1)}$. Define $\mathbf{H_Q}(0) \in \mathbb{R}^{N \times N} = \boldsymbol{\sigma}(\mathbf{W}(0), \mathbf{Q})^\top \boldsymbol{\sigma}(\mathbf{W}(0), \mathbf{Q})/(Nm)$, then it follows from Theorem C.1 that $\|\mathbf{H_Q}(0) - \mathbf{K}_N\|_2 \leq C_1(m, d, 1/n)$ with $\mathbf{W}(0) \in \mathcal{W}_0$, where $\mathbf{K}_{\mathbf{Q},N} \in \mathbb{R}^{N \times N}$ and $[\mathbf{K}_{\mathbf{Q},N}]_{ij} = K(\vec{\mathbf{q}}_i, \vec{\mathbf{q}}_j)/N$. It follows that

$$\|\mathbf{A}\|_2 = \|\mathbf{H_Q}(0)\|_2 \leq \|\mathbf{K}_{\mathbf{Q},N}\|_2 + C_1(m, d, 1/n) \leq \frac{1+\pi}{2\pi} + C_1(m, d, 1/n) \leq 1, \tag{83}$$

since $m \gtrsim (d \log m)^3$. For all $i \in [n]$ we have

$$\left|\mathbf{E}_i^{(1)}\right| = \left| \frac{1}{\sqrt{m}} \sum_{r'=1}^{m} \sum_{r=1}^{m} a_r(t+1) \left( \sigma\left(\vec{\mathbf{w}}_{r'}(t+1)^\top \vec{\mathbf{x}}_i\right) - \sigma\left(\vec{\mathbf{w}}_{r'}(t)^\top \vec{\mathbf{x}}_i\right) \right) \mathbf{A}_{r'r} \right|$$

$$\leq \frac{R}{\sqrt{m}} \sum_{r'=1}^{m} \sum_{r=1}^{m} |a_r(t+1)| \, \mathbf{A}_{r'r} \leq 2R^2 \sqrt{m} \cdot \|\mathbf{A}\|_2 \leq 2R^2 \sqrt{m}, \tag{84}$$

where the last inequality follow from (83). $\mathbf{D}_i^{(1)}$ on the RHS of (82) is expressed by

$$\mathbf{D}_i^{(1)} = -\frac{\eta}{nm} \boldsymbol{\sigma}^\top(\mathbf{W}(0), \vec{\mathbf{x}}_i) \mathbf{A}^2 \boldsymbol{\sigma}(\mathbf{W}(t), \mathbf{S})(\widehat{\mathbf{y}}(t) - \mathbf{y})$$

$$= \underbrace{-\frac{\eta}{nm} \boldsymbol{\sigma}^\top(\mathbf{W}(0), \vec{\mathbf{x}}_i) \mathbf{A}^2 \boldsymbol{\sigma}(\mathbf{W}(0), \mathbf{S})(\widehat{\mathbf{y}}(t) - \mathbf{y})}_{:=\mathbf{D}_i^{(2)}}$$

$$+ \underbrace{\frac{\eta}{nm} \boldsymbol{\sigma}^\top(\mathbf{W}(0), \vec{\mathbf{x}}_i) \mathbf{A}^2 \left(\boldsymbol{\sigma}(\mathbf{W}(0), \mathbf{S}) - \boldsymbol{\sigma}(\mathbf{W}(t), \mathbf{S})\right)(\widehat{\mathbf{y}}(t) - \mathbf{y})}_{:=\mathbf{E}_i^{(2)}} = \mathbf{D}_i^{(2)} + \mathbf{E}_i^{(2)}. \tag{85}$$

It follows from (81) that $\left\|\mathbf{E}^{(2)}\right\|_2$ is bounded by

$$\left\|\mathbf{E}^{(2)}\right\|_2 \leq \frac{\eta}{nm} \|\boldsymbol{\sigma}^\top(\mathbf{W}(0), \mathbf{S})\|_2 \|\mathbf{A}\|_2^2 \|\boldsymbol{\sigma}(\mathbf{W}(0), \mathbf{S}) - \boldsymbol{\sigma}(\mathbf{W}(t), \mathbf{S})\|_2 \|\widehat{\mathbf{y}}(t) - \mathbf{y}\|_2$$

$$\overset{\text{①}}{\leq} \frac{\eta}{nm} \cdot \sqrt{nm} \cdot \sqrt{nm} R \cdot c_{\mathbf{u}} \sqrt{n} = \eta c_{\mathbf{u}} R \sqrt{n}. \tag{86}$$

With $\mathbf{W}(0) \in \mathcal{W}_0$ and $m \gtrsim (d \log m)^3$, it follows from (101) in the proof of Lemma C.13 that $\left\|\boldsymbol{\sigma}^\top(\mathbf{W}(0), \mathbf{S})\right\|_2 \leq \sqrt{nm}$. It also follows from (83) that $\|\mathbf{A}\|_2 \leq 1$, so that ① holds.

$\mathbf{D}_i^{(2)}$ on the RHS of (89) is expressed by

$$\mathbf{D}^{(2)} = -\frac{\eta}{nm} \boldsymbol{\sigma}^\top(\mathbf{W}(0), \mathbf{S}) \mathbf{A}^2 \boldsymbol{\sigma}(\mathbf{W}(0), \mathbf{S})(\widehat{\mathbf{y}}(t) - \mathbf{y})$$

$$= \underbrace{-\eta \mathbf{K}_n^{(\text{attn})}(\widehat{\mathbf{y}}(t) - \mathbf{y})}_{:=\mathbf{D}^{(3)}} + \underbrace{\eta \left( \frac{1}{nm} \boldsymbol{\sigma}^\top(\mathbf{W}(0), \mathbf{S}) \mathbf{A}^2 \boldsymbol{\sigma}(\mathbf{W}(0), \mathbf{S}) - \mathbf{K}_n^{(\text{attn})} \right)(\widehat{\mathbf{y}}(t) - \mathbf{y})}_{:=\mathbf{E}^{(3)}}. \tag{87}$$

It follows from Lemma C.11 that $\left\|\mathbf{E}^{(3)}\right\|_2$ is bounded by

$$\left\|\mathbf{E}^{(3)}\right\|_2 \lesssim \eta c_{\mathbf{u}} \sqrt{n} \left( C_1(m, d, 1/n) + \sqrt{\frac{\log(n/\delta)}{N}} \right). \tag{88}$$

It follows from (85) and (87) that

$$\mathbf{D}_i^{(1)} = \mathbf{D}_i^{(3)} + \mathbf{E}_i^{(2)} + \mathbf{E}_i^{(3)}. \tag{89}$$

It then follows from (82) and (89) that

$$\widehat{\mathbf{y}}(t+1) - \widehat{\mathbf{y}}(t) = \mathbf{D}^{(3)} + \underbrace{\mathbf{E}^{(1)} + \mathbf{E}^{(2)} + \mathbf{E}^{(3)}}_{:=\mathbf{E}_i} = -\eta \mathbf{K}_n^{(\mathrm{attn})}(\widehat{\mathbf{y}}(t) - \mathbf{y}) + \mathbf{E}, \tag{90}$$

where $\mathbf{E} \in \mathbb{R}^n$ with its $i$-th element being $\mathbf{E}_i$, and $\mathbf{E} = \mathbf{E}^{(1)} + \mathbf{E}^{(2)} + \mathbf{E}^{(3)}$. It then follows from (84), (86), and (88) that

$$\|\mathbf{E}\|_2 \lesssim R^2 \sqrt{mn} + \eta c_{\mathbf{u}} \sqrt{n} \left( R + C_1(m, d, 1/n) + \sqrt{\frac{\log(n/\delta)}{N}} \right), \tag{91}$$

which together with (90) completes the proof.

$\square$

**Lemma C.11.** Suppose the random initialization $\mathbf{W}(0) \in \mathcal{W}_0$, $m \gtrsim (d \log m)^3$. Then with probability at least $1 - \delta$ over $\mathbf{Q}$,

$$\left\| \frac{1}{nm} \boldsymbol{\sigma}^\top(\mathbf{W}(0), \mathbf{S}) \mathbf{A}^2 \boldsymbol{\sigma}(\mathbf{W}(0), \mathbf{S}) - \mathbf{K}_n^{(\mathrm{attn})} \right\|_2 \lesssim C_1(m, d, 1/n) + \sqrt{\frac{\log(n/\delta)}{N}}. \tag{92}$$

*Proof.* We have

$$\frac{1}{nm} \boldsymbol{\sigma}^\top(\mathbf{W}(0), \mathbf{S}) \mathbf{A}^2 \boldsymbol{\sigma}(\mathbf{W}(0), \mathbf{S}) = \frac{1}{nN^2} \widehat{K}_{\mathbf{S}, \mathbf{Q}} \widehat{K}_{\mathbf{Q}, \mathbf{Q}} \widehat{K}_{\mathbf{S}, \mathbf{Q}}^\top, \tag{93}$$

where $\widehat{K}_{\mathbf{S}, \mathbf{Q}} := \boldsymbol{\sigma}^\top(\mathbf{W}(0), \mathbf{S}) \boldsymbol{\sigma}(\mathbf{W}(0), \mathbf{Q})/m$, $\widehat{K}_{\mathbf{Q}, \mathbf{Q}} := \boldsymbol{\sigma}(\mathbf{W}(0), \mathbf{Q})^\top \boldsymbol{\sigma}(\mathbf{W}(0), \mathbf{Q})/m$. We define $\mathbf{K}_{\mathbf{S}, \mathbf{Q}} \in \mathbb{R}^{n \times N}$ with $[\mathbf{K}_{\mathbf{S}, \mathbf{Q}}]_{ij} = K(\vec{\mathbf{x}}_i, \vec{\mathbf{q}}_j)$ for all $i \in [n]$, $j \in [N]$, and $\mathbf{K}_{\mathbf{Q}, \mathbf{Q}} \in \mathbb{R}^{N \times N}$ with $[\mathbf{K}_{\mathbf{Q}, \mathbf{Q}}]_{ij} = K(\vec{\mathbf{q}}_i, \vec{\mathbf{q}}_j)$ for all $i, j \in [N]$.

Since $\mathbf{W}(0) \in \mathcal{W}_0$, we have $\left\| \widehat{K}_{\mathbf{S}, \mathbf{Q}} - \mathbf{K}_{\mathbf{S}, \mathbf{Q}} \right\|_2 \leq \sqrt{nN} C_1(m, d, 1/n)$ and $\left\| \widehat{K}_{\mathbf{Q}, \mathbf{Q}} - \mathbf{K}_{\mathbf{Q}, \mathbf{Q}} \right\|_2 \leq NC_1(m, d, 1/n)$. As a result,

$$\widehat{K}_{\mathbf{S}, \mathbf{Q}} \widehat{K}_{\mathbf{Q}, \mathbf{Q}} \widehat{K}_{\mathbf{S}, \mathbf{Q}}^\top = \underbrace{(\widehat{K}_{\mathbf{S}, \mathbf{Q}} - \mathbf{K}_{\mathbf{S}, \mathbf{Q}}) \widehat{K}_{\mathbf{Q}, \mathbf{Q}} \widehat{K}_{\mathbf{S}, \mathbf{Q}}^\top}_{E_1} + \underbrace{\mathbf{K}_{\mathbf{S}, \mathbf{Q}}(\widehat{K}_{\mathbf{Q}, \mathbf{Q}} - \mathbf{K}_{\mathbf{Q}, \mathbf{Q}}) \widehat{K}_{\mathbf{S}, \mathbf{Q}}^\top}_{E_2}$$
$$+ \underbrace{\mathbf{K}_{\mathbf{S}, \mathbf{Q}} \mathbf{K}_{\mathbf{Q}, \mathbf{Q}} (\widehat{K}_{\mathbf{S}, \mathbf{Q}} - \mathbf{K}_{\mathbf{S}, \mathbf{Q}})^\top}_{E_3} + \mathbf{K}_{\mathbf{S}, \mathbf{Q}} \mathbf{K}_{\mathbf{Q}, \mathbf{Q}} \mathbf{K}_{\mathbf{S}, \mathbf{Q}}^\top, \tag{94}$$

where

$$\max \left\{ \|E_1\|_2, \|E_2\|_2, \|E_3\|_2 \right\} \leq nN^2 C_1(m, d, 1/n), \tag{95}$$

since $\max \left\{ \left\| \widehat{K}_{\mathbf{Q}, \mathbf{Q}} \right\|_\infty, \left\| \widehat{K}_{\mathbf{S}, \mathbf{Q}} \right\|_\infty, \|\mathbf{K}_{\mathbf{S}, \mathbf{Q}}\|_\infty, \|\mathbf{K}_{\mathbf{Q}, \mathbf{Q}}\|_\infty \right\} \leq 1$ with $m \gtrsim (d \log m)^3$. We also have $\mathbf{K}_{\mathbf{S}, \mathbf{Q}} \mathbf{K}_{\mathbf{Q}, \mathbf{Q}} \mathbf{K}_{\mathbf{S}, \mathbf{Q}}^\top / (nN^2) = \widehat{\mathbf{K}}_n^{(\mathrm{attn})}$. It follows from Theorem C.22 that with probability at least $1 - \delta$ over $\mathbf{Q}$, $\left\| \widehat{\mathbf{K}}^{(\mathrm{attn})} - \mathbf{K}^{(\mathrm{attn})} \right\|_2 \lesssim n \sqrt{\frac{\log(n/\delta)}{N}}$. Combining such result with (93)- (95), we have

$$\left\| \frac{1}{nm} \boldsymbol{\sigma}^\top(\mathbf{W}(0), \mathbf{S}) \mathbf{A}^2 \boldsymbol{\sigma}(\mathbf{W}(0), \mathbf{S}) - \mathbf{K}_n^{(\mathrm{attn})} \right\|_2 \leq \left\| \frac{1}{nN^2} \widehat{K}_{\mathbf{S}, \mathbf{Q}} \widehat{K}_{\mathbf{Q}, \mathbf{Q}} \widehat{K}_{\mathbf{S}, \mathbf{Q}}^\top - \mathbf{K}_n^{(\mathrm{attn})} \right\|_2$$
$$\lesssim C_1(m, d, 1/n) + \sqrt{\frac{\log(n/\delta)}{N}},$$

which completes the proof.

$\square$

**Lemma C.12.** Suppose that $t \in [0, T-1]$ for $T \geq 1$, $\|\widehat{\mathbf{y}}(t') - \mathbf{y}\|_2 \leq \sqrt{n} c_{\mathbf{u}}$ for all $0 \leq t' \leq t$, $N \gtrsim (d \log N)^3$, $m \gtrsim \max\left\{4(2d + 3\log(2mn))\left(\eta c_{\mathbf{u}} T\right)^2, (d \log m)^3\right\}$, and the random initialization $\mathbf{W}(0) \in \mathcal{W}_0$ Then with probability at least $1 - 1/n$ over $\mathbf{Q}$ and $\mathbf{W}(0)$, for all $0 \leq t' \leq t + 1$,

$$\left\|\vec{\mathbf{w}}_{\mathbf{S},r}(t') - \vec{\mathbf{w}}_r(0)\right\|_2 \leq R, \tag{96}$$

$$|a_r(t')| \leq \frac{2\eta c_{\mathbf{u}} \sqrt{2d + 3\log(2mn)} T}{\sqrt{m}} = 2R. \tag{97}$$

*Proof.* We prove (96) and (97) by induction. First, (96) and (97) hold trivially when $t' = 0$. Suppose that (96) and (97) hold for all $t' \in [0, t'']$ with $t'' \in [0, t]$, we now prove that they also hold for $t' = t'' + 1$.

With $m \geq 4(2d + 3\log(2mn))\left(\eta c_{\mathbf{u}} T\right)^2$, we have $|a_r(t')| \leq 1$ for all $t' \in [0, t'']$. It follows from Lemma C.14 that (96) holds with $t' = t'' + 1$. Also, (97) holds with $t' = t'' + 1$ since $R \leq 1$ with such $m$. As a result, (96) and (97) hold for all $t' \in [0, t+1]$. $\qquad\square$

**Lemma C.13.** Suppose that $t \in [0, T - 1]$ for $T \geq 1$, $\|\widehat{\mathbf{y}}(t') - \mathbf{y}\|_2 \leq \sqrt{n} c_{\mathbf{u}}$ and $\left\|\vec{\mathbf{w}}_{\mathbf{S},r}(t') - \vec{\mathbf{w}}_r(0)\right\|_2 \leq 1$ hold for all $0 \leq t' \leq t$, $N \gtrsim (d \log N)^3$, $m \gtrsim (d \log m)^3$, and the random initialization $\mathbf{W}(0) \in \mathcal{W}_0$. Then with probability at least $1 - 1/n$ over $\mathbf{Q}$ and $\mathbf{W}(0)$,

$$|a_r(t')| \leq \frac{\eta c_{\mathbf{u}} \sqrt{2d + 3\log(2mn)} T}{\sqrt{m}}, \quad \forall 0 \leq t' \leq t + 1, \forall r \in [m]. \tag{98}$$

*Proof.* First, for every $t' \in [0, t]$ and every $r \in [m]$, we have

$$|a_r(t' + 1) - a_r(t')| \leq \frac{\eta}{n\sqrt{m}} \|\mathbf{A}_r\|_2 \|\boldsymbol{\sigma}(\mathbf{W}(t'), \mathbf{S})\|_2 \|\widehat{\mathbf{y}}(t') - \mathbf{y}\|_2, \tag{99}$$

where $\mathbf{A}_r$ is the $r$-th row of $\mathbf{A}$. It follows from Theorem C.8 that with probability at least $1 - 1/n$ over $\mathbf{Q}$ and $\mathbf{W}(0)$,

$$\|\mathbf{A}_r\|_2 \leq \frac{\sqrt{2d + 3\log(2mn)}}{\sqrt{m}} \left(\frac{1 + \pi}{2\pi} + C_1(N, d, 1/(2n))\right) \leq \frac{\sqrt{2d + 3\log(2mn)}}{\sqrt{m}}, \forall r \in [m], \tag{100}$$

with $N \gtrsim (d \log N)^3$.

Define $\mathbf{H}(0) \in \mathbb{R}^{n \times n} = \boldsymbol{\sigma}(\mathbf{W}(0), \mathbf{S})^\top \boldsymbol{\sigma}(\mathbf{W}(0), \mathbf{S})/(nm)$, then it follows from Theorem C.1 that $\|\mathbf{H}(0) - \mathbf{K}_n\|_2 \leq C_1(m, d, 1/n)$ since $\mathbf{W}(0) \in \mathcal{W}_0$. It follows that

$$\|\mathbf{H}(0)\|_2 \leq \|\mathbf{K}_n\|_2 + C_1(m, d, 1/n) \leq \frac{1 + \pi}{2\pi} + C_1(m, d, 1/n) \leq 1,$$

$$\|\boldsymbol{\sigma}(\mathbf{W}(0), \mathbf{S})\|_2 \leq \sqrt{mn}\sqrt{\|\mathbf{H}(0)\|_2} \leq \sqrt{mn}, \tag{101}$$

since $m \gtrsim (d \log m)^3$. We then have

$$\|\boldsymbol{\sigma}(\mathbf{W}(t'), \mathbf{S})\|_2 \leq \|\boldsymbol{\sigma}(\mathbf{W}(0), \mathbf{S})\|_2 + \|\boldsymbol{\sigma}(\mathbf{W}(t'), \mathbf{S}) - \boldsymbol{\sigma}(\mathbf{W}(0), \mathbf{S})\|_2$$
$$\leq \sqrt{mn} + \sqrt{mn} = 2\sqrt{mn}. \tag{102}$$

It then follows from (99)-(102) that for every $t' \in [0, t]$ and every $r \in [m]$,

$$\|a_r(t' + 1) - a_r(t')\|_2 \leq \frac{2\eta c_{\mathbf{u}} \sqrt{2d + 3\log(2mn)}}{\sqrt{m}},$$

which proves (98). $\qquad\square$

**Lemma C.14.** Suppose that $t \in [0, T-1]$ for $T \geq 1$, and $\|\widehat{\mathbf{y}}(t') - \mathbf{y}\|_2 \leq \sqrt{n} c_{\mathbf{u}}$, $|a_r(t')| \leq 1$ hold for all $0 \leq t' \leq t$ and every $r \in [m]$, $N \gtrsim (d \log N)^3$, and the random initialization $\mathbf{W}(0) \in \mathcal{W}_0$. Then with probability at least $1 - 1/n$ over $\mathbf{Q}$ and $\mathbf{W}(0)$,

$$\left\|\vec{\mathbf{w}}_{\mathbf{S},r}(t') - \vec{\mathbf{w}}_r(0)\right\|_2 \leq R, \quad \forall 0 \leq t' \leq t + 1. \tag{103}$$

*Proof.* First, it follows from (100) in the proof of Lemma C.13 that with probability at least $1 - 1/n$ over $\mathbf{Q}$ and $\mathbf{W}(0)$, for every $t' \in [0, t]$ and every $r \in [m]$, we have

$$\|\mathbf{A}\boldsymbol{a}(t')|_r| \leq \|\mathbf{A}_r\|_2 \|\boldsymbol{a}(t')\|_2 \leq \sqrt{2d + 3\log(2mn)}. \tag{104}$$

Let $[\mathbf{Z_S}(t)]_{[(r-1)d+1:rd]}$ denote the submatrix of $\mathbf{Z_S}(t)$ formed by the the rows of $\mathbf{Z_Q}(t)$ with row indices in $[(r-1)d+1:rd]$. By the GD update rule we have for $t \in [0, T-1]$ that

$$\overrightarrow{\mathbf{w}}_{\mathbf{S},r}(t'+1) - \overrightarrow{\mathbf{w}}_{\mathbf{S},r}(t') = -\frac{\eta}{n} [\mathbf{Z_S}(t)]_{[(r-1)d+1:rd]} (\widehat{\mathbf{y}}(t') - \mathbf{y}). \tag{105}$$

It follows from (104) that $\left\| [\mathbf{Z_S}(t')]_{[(r-1)d+1:rd]} \right\|_2 \leq \sqrt{2d + 3\log(2mn)} \cdot \sqrt{n/m}$ for all $t' \in [0, t]$. It then follows from (105) that

$$\left\| \overrightarrow{\mathbf{w}}_{\mathbf{S},r}(t'+1) - \overrightarrow{\mathbf{w}}_{\mathbf{S},r}(t') \right\|_2 \leq \frac{\eta}{n} \left\| [\mathbf{Z_S}(t')]_{[(r-1)d+1:rd]} \right\|_2 \|\widehat{\mathbf{y}}(t) - \mathbf{y}\|_2 \leq \frac{\eta c_{\mathbf{u}} \sqrt{2d + 3\log(2mn)}}{\sqrt{m}}. \tag{106}$$

Note that (103) trivially holds for $t' = 0$. For $t' \in [1, t+1]$, it follows from (106) that

$$\left\| \overrightarrow{\mathbf{w}}_{\mathbf{S},r}(t') - \overrightarrow{\mathbf{w}}_r(0) \right\|_2 \leq \sum_{t''=0}^{t'-1} \left\| \overrightarrow{\mathbf{w}}_{\mathbf{S},r}(t''+1) - \overrightarrow{\mathbf{w}}_{\mathbf{S},r}(t'') \right\|_2 \leq \frac{\eta c_{\mathbf{u}} \sqrt{2d + 3\log(2mn)} T}{\sqrt{m}} = R,$$

which completes the proof. $\qquad\square$

**Lemma C.15.** Let $h(\cdot) = \sum_{t'=0}^{t-1} h(\cdot, t')$ for $t \in [T], T \leq \widehat{T}$ where

$$h(\cdot, t') = v(\cdot, t') + \widehat{e}(\cdot, t'),$$

$$v(\cdot, t') = \frac{\eta}{n} \sum_{j=1}^{n} K^{(\mathrm{attn})}(\overrightarrow{\mathbf{x}}_j, \mathbf{x}) \mathbf{v}_j(t'),$$

$$\widehat{e}(\cdot, t') = \frac{\eta}{n} \sum_{j=1}^{n} K^{(\mathrm{attn})}(\overrightarrow{\mathbf{x}}_j, \mathbf{x}) \overrightarrow{\mathbf{e}}_j(t'),$$

and $\mathbf{v}(t') \in \mathcal{V}_{t'}$, $\mathbf{e}(t') \in \mathcal{E}_{t',\tau}$ for all $0 \leq t' \leq t - 1$. Suppose that

$$\tau \lesssim 1/(\eta T). \tag{107}$$

Then with probability at least $1 - \exp\left(-\Theta(n\widehat{\varepsilon}_n^2)\right)$ over $\mathbf{w}$,

$$\|h\|_{\mathcal{H}_{K^{(\mathrm{attn})}}} \leq B_h = \mu_0 + 1 + \sqrt{2}, \tag{108}$$

where $B_h$ is also defined in (29).

*Proof.* The proof is similar to the proof of (Yang & Li, 2025, Lemma B.5). $\qquad\square$

**Lemma C.16.** Suppose $\mathbf{W}(0) \in \mathcal{W}_0$, $m \gtrsim (d\log m)^3$, and $\|\mathbf{u}(t')\|_2 \leq c_{\mathbf{u}}\sqrt{n}$ with $c_{\mathbf{u}} = \Theta(1)$. Then with probability at least $1 - \delta$ over $\mathbf{Q}$,

$$\left\| \frac{1}{nm} \sum_{r'=1}^{m} \sum_{r=1}^{m} \sigma\left( \overrightarrow{\mathbf{w}}_{r'}(0)^{\top} \mathbf{x} \right) \mathbf{A}_{r'r} \mathbf{A}_r \boldsymbol{\sigma}(\mathbf{W}(0), \mathbf{S}) \mathbf{u}(t') - \frac{1}{n} \sum_{j=1}^{n} K^{(\mathrm{attn})}(\mathbf{x}, \overrightarrow{\mathbf{x}}_j) \mathbf{u}_j(t') \right\|_{\infty}$$

$$\lesssim C_1(m, d, 1/n) + \sqrt{\frac{\log(n/\delta)}{N}}. \tag{109}$$

*Proof.* For every $j \in [n]$, we have

$$\frac{1}{nm} \sum_{r'=1}^{m} \sum_{r=1}^{m} \sigma\left( \overrightarrow{\mathbf{w}}_{r'}(0)^{\top} \mathbf{x} \right) \mathbf{A}_{r'r} \mathbf{A}_r \boldsymbol{\sigma}(\mathbf{W}(0), \overrightarrow{\mathbf{x}}_j) = \frac{1}{nN^2} \widehat{K}_{\mathbf{x},\mathbf{Q}} \widehat{K}_{\mathbf{Q},\mathbf{Q}} \widehat{K}_{\mathbf{Q},\overrightarrow{\mathbf{x}}_j}, \tag{110}$$

where $\widehat{K}_{\mathbf{x},\mathbf{Q}} := \boldsymbol{\sigma}^\top(\mathbf{W}(0),\mathbf{x})\boldsymbol{\sigma}(\mathbf{W}(0),\mathbf{Q})/m$ and $\widehat{K}_{\mathbf{Q},\mathbf{x}} = \widehat{K}_{\mathbf{x},\mathbf{Q}}^\top$. The following notations defined in the proof of Lemma C.11 are also used in this proof. In particular, $\widehat{K}_{\mathbf{S},\mathbf{Q}} = \boldsymbol{\sigma}^\top(\mathbf{W}(0),\mathbf{S})\boldsymbol{\sigma}(\mathbf{W}(0),\mathbf{Q})/m$, $\widehat{K}_{\mathbf{Q},\mathbf{Q}} = \boldsymbol{\sigma}(\mathbf{W}(0),\mathbf{Q})^\top\boldsymbol{\sigma}(\mathbf{W}(0),\mathbf{Q})/m$, $\mathbf{K}_{\mathbf{S},\mathbf{Q}} \in \mathbb{R}^{n \times N}$ with $[\mathbf{K}_{\mathbf{S},\mathbf{Q}}]_{ij} = K(\vec{\mathbf{x}}_i, \vec{\mathbf{q}}_j)$ for all $i \in [n]$, $j \in [N]$, and $\mathbf{K}_{\mathbf{Q},\mathbf{Q}} \in \mathbb{R}^{N \times N}$ with $[\mathbf{K}_{\mathbf{Q},\mathbf{Q}}]_{ij} = K(\vec{\mathbf{q}}_i, \vec{\mathbf{q}}_j)$ for all $i, j \in [N]$. We further define $K(\mathbf{Q},\mathbf{x}) \in \mathbb{R}^N$ with $[K(\mathbf{Q},\mathbf{x})]_i = K(\mathbf{x}, \vec{\mathbf{q}}_i)$ for all $i \in [N]$, and $K(\mathbf{x},\mathbf{Q}) = K^\top(\mathbf{Q},\mathbf{x})$. $K^{(\mathrm{attn})}(\mathbf{Q},\mathbf{x}) \in \mathbb{R}^N$ is defined similarly for the kernel $\widehat{K}^{(\mathrm{attn})}$.

Since $\mathbf{W}(0) \in \mathcal{W}_0$, we have $\left\|\widehat{K}_{\mathbf{x},\mathbf{Q}} - K_{\mathbf{x},\mathbf{Q}}\right\|_2 \leq \sqrt{N}C_1(m,d,1/n)$ and $\left\|\widehat{K}_{\mathbf{Q},\mathbf{Q}} - \mathbf{K}_{\mathbf{Q},\mathbf{Q}}\right\|_2 \leq NC_1(m,d,1/n)$. As a result,

$$\widehat{K}_{\mathbf{x},\mathbf{Q}}\widehat{K}_{\mathbf{Q},\mathbf{Q}}\widehat{K}_{\mathbf{Q},\vec{\mathbf{x}}_j} = \underbrace{(\widehat{K}_{\mathbf{x},\mathbf{Q}} - K_{\mathbf{x},\mathbf{Q}})\widehat{K}_{\mathbf{Q},\mathbf{Q}}\widehat{K}_{\mathbf{Q},\vec{\mathbf{x}}_j}}_{E_1} + \underbrace{K_{\mathbf{x},\mathbf{Q}}(\widehat{K}_{\mathbf{Q},\mathbf{Q}} - \mathbf{K}_{\mathbf{Q},\mathbf{Q}})\widehat{K}_{\mathbf{Q},\vec{\mathbf{x}}_j}}_{E_2}$$
$$+ \underbrace{K_{\mathbf{x},\mathbf{Q}}\mathbf{K}_{\mathbf{Q},\mathbf{Q}}(\widehat{K}_{\mathbf{Q},\vec{\mathbf{x}}_j} - \mathbf{K}_{\mathbf{Q},\vec{\mathbf{x}}_j})}_{E_3} + K_{\mathbf{x},\mathbf{Q}}\mathbf{K}_{\mathbf{Q},\mathbf{Q}}\mathbf{K}_{\mathbf{Q},\vec{\mathbf{x}}_j}, \qquad (111)$$

where

$$\max\left\{\|E_1\|_2, \|E_2\|_2, \|E_3\|_2\right\} \leq N^2 C_1(m,d,1/n), \qquad (112)$$

due to the fact that $\max\left\{\left\|\widehat{K}_{\mathbf{Q},\mathbf{Q}}\right\|_\infty, \left\|\widehat{K}_{\mathbf{S},\mathbf{Q}}\right\|_\infty, \|\mathbf{K}_{\mathbf{S},\mathbf{Q}}\|_\infty, \|\mathbf{K}_{\mathbf{Q},\mathbf{Q}}\|_\infty\right\} \leq 1$ with $m \gtrsim (d\log m)^3$.

We also have $\mathbf{K}_{\mathbf{x},\mathbf{Q}}\mathbf{K}_{\mathbf{Q},\mathbf{Q}}\mathbf{K}_{\mathbf{Q},\vec{\mathbf{x}}_j}/(nN^2) = 1/n \cdot \widehat{K}^{(\mathrm{attn})}(\mathbf{x}, \vec{\mathbf{x}}_j)$. It follows from Theorem C.22 that with probability at least $1 - \delta$ over $\mathbf{Q}$, $\sup_{\mathbf{x}\in\mathcal{X}, j\in[n]}\left|K^{(\mathrm{attn})}(\mathbf{x},\vec{\mathbf{x}}_j) - \widehat{K}^{(\mathrm{attn})}(\mathbf{x},\vec{\mathbf{x}}_j)\right| \lesssim \sqrt{\frac{\log(n/\delta)}{N}}$. Combining such result with (110)-(112), we have

$$\left\|\frac{1}{nm}\sum_{r'=1}^{m}\sum_{r=1}^{m}\sigma\left(\vec{\mathbf{w}}_{r'}(0)^\top\mathbf{x}\right)\mathbf{A}_{r'r}\mathbf{A}_r\boldsymbol{\sigma}(\mathbf{W}(0),\mathbf{S})\mathbf{u}(t') - \frac{1}{n}\sum_{j=1}^{n}K^{(\mathrm{attn})}(\mathbf{x},\vec{\mathbf{x}}_j)\mathbf{u}_j(t')\right\|_\infty$$

$$\leq \left\|\frac{1}{nN^2}\sum_{j=1}^{n}\widehat{K}_{\mathbf{x},\mathbf{Q}}\widehat{K}_{\mathbf{Q},\mathbf{Q}}\widehat{K}_{\mathbf{Q},\vec{\mathbf{x}}_j}\mathbf{u}_j(t') - \frac{1}{n}\sum_{j=1}^{n}\widehat{K}^{(\mathrm{attn})}(\mathbf{x},\vec{\mathbf{x}}_j)\mathbf{u}_j(t')\right\|_\infty$$

$$+ \left\|\frac{1}{n}\sum_{j=1}^{n}\widehat{K}^{(\mathrm{attn})}(\mathbf{x},\vec{\mathbf{x}}_j)\mathbf{u}_j(t') - \frac{1}{n}\sum_{j=1}^{n}K^{(\mathrm{attn})}(\mathbf{x},\vec{\mathbf{x}}_j)\mathbf{u}_j(t')\right\|_\infty \lesssim C_1(m,d,1/n) + \sqrt{\frac{\log(n/\delta)}{N}},$$

which completes the proof. $\qquad\square$

**Lemma C.17** (In the proof of (Raskutti et al., 2014, Lemma 8)). For any $f \in \mathcal{H}_K(\mu_0)$, we have

$$\frac{1}{n}\sum_{i=1}^{n}\frac{\left[\mathbf{U}^\top f(\mathbf{S}')\right]_i^2}{\widehat{\lambda}_i} \leq \mu_0^2. \qquad (113)$$

**Lemma C.18.** For any positive real number $a \in (0,1)$ and natural number $t$, we have

$$(1-a)^t \leq e^{-ta} \leq \frac{1}{eta}. \qquad (114)$$

*Proof.* The result follows from the facts that $\log(1-a) \leq a$ for $a \in (0,1)$ and $\sup_{u\in\mathbb{R}}ue^{-u} \leq 1/e$. $\qquad\square$

**Lemma C.19** ((Yang & Li, 2024, Lemma B.7)). With probability at least $1 - 4\exp(-\Theta(n\varepsilon_n^2))$,

$$\varepsilon_n^2 \lesssim \widehat{\varepsilon}_n^2, \quad \widehat{\varepsilon}_n^2 \lesssim \varepsilon_n^2. \qquad (115)$$

Similarly, with probability at least $1 - 4\exp(-\Theta(n\varepsilon_{K,n}^2))$,

$$\varepsilon_{K,n}^2 \lesssim \widehat{\varepsilon}_{K,n}^2, \quad \widehat{\varepsilon}_{K,n}^2 \lesssim \varepsilon_{K,n}^2. \qquad (116)$$

**Proof of Lemma C.5.** We first decompose the Rademacher complexity of the function class $\{f \in \mathcal{F}(B, w)\colon \mathbb{E}_P\left[f^2\right] \leq r\}$ into two terms as follows:

$$\mathfrak{R}\left(\left\{f\colon f \in \mathcal{F}(B, w), \mathbb{E}_P\left[f^2\right] \leq r\right\}\right)$$

$$\leq \underbrace{\frac{1}{n}\mathbb{E}\left[\sup_{f \in \mathcal{F}(B, w)\colon \mathbb{E}_P[f^2] \leq r}\sum_{i=1}^{n}\sigma_i h(\vec{\mathbf{x}}_i)\right]}_{:=\mathcal{R}_1} + \underbrace{\frac{1}{n}\mathbb{E}\left[\sup_{f \in \mathcal{F}(B, w)\colon \mathbb{E}_P[f^2] \leq r}\sum_{i=1}^{n}\sigma_i e(\vec{\mathbf{x}}_i)\right]}_{:=\mathcal{R}_2}. \quad (117)$$

We now analyze the upper bounds for $\mathcal{R}_1, \mathcal{R}_2$ on the RHS of (117).

**Derivation for the upper bound for $\mathcal{R}_1$.**

According to definition of $\mathcal{F}(B, w)$ in (27), for any $f \in \mathcal{F}(B, w)$, we have $f = h + e$ with $h \in \mathcal{H}_{K^{(\mathrm{attn})}}(B), e \in L^\infty, \|e\|_\infty \leq w$.

When $\mathbb{E}_P\left[f^2\right] \leq r$, it follows from the triangle inequality that $\|h\|_{L^2} \leq \|f\|_{L^2} + \|e\|_{L^2} \leq \sqrt{r} + w := r_h$. We now consider $h \in \mathcal{H}_{K^{(\mathrm{attn})}}(B)$ with $\|h\|_{L^2} \leq r_h$ in the remaining of this proof. We have

$$\sum_{i=1}^{n}\sigma_i f(\vec{\mathbf{x}}_i) = \sum_{i=1}^{n}\sigma_i\left(h(\vec{\mathbf{x}}_i) + e(\vec{\mathbf{x}}_i)\right)$$

$$= \left\langle h, \sum_{i=1}^{n}\sigma_i K^{(\mathrm{attn})}(\cdot, \vec{\mathbf{x}}_i)\right\rangle_{\mathcal{H}_{K^{(\mathrm{attn})}}} + \sum_{i=1}^{n}\sigma_i e(\vec{\mathbf{x}}_i). \quad (118)$$

Because $\left\{v_q^{(\mathrm{int})} = \sqrt{\lambda_q^{(\mathrm{attn})}}e_q\right\}_{q \geq 1}$ is an orthonormal basis of $\mathcal{H}_{K^{(\mathrm{attn})}}$, for any $0 \leq Q \leq n$, we further express the first term on the RHS of (118) as

$$\left\langle h, \sum_{i=1}^{n}\sigma_i K^{(\mathrm{attn})}(\cdot, \vec{\mathbf{x}}_i)\right\rangle_{\mathcal{H}_{K^{(\mathrm{attn})}}} =$$

$$\left\langle \sum_{q=1}^{Q}\sqrt{\lambda_q^{(\mathrm{attn})}}\left\langle h, v_q^{(\mathrm{int})}\right\rangle_{\mathcal{H}_{K^{(\mathrm{attn})}}}v_q^{(\mathrm{int})}, \sum_{q=1}^{Q}\left\langle \sum_{i=1}^{n}\sigma_i K^{(\mathrm{attn})}(\cdot, \vec{\mathbf{x}}_i), v_q^{(\mathrm{int})}\right\rangle_{\mathcal{H}_{K^{(\mathrm{attn})}}}\frac{v_q^{(\mathrm{int})}}{\sqrt{\lambda_q^{(\mathrm{attn})}}}\right\rangle_{\mathcal{H}_{K^{(\mathrm{attn})}}}$$

$$+ \left\langle h, \sum_{q > Q}\left\langle \sum_{i=1}^{n}\sigma_i K^{(\mathrm{attn})}(\cdot, \vec{\mathbf{x}}_i), v_q^{(\mathrm{int})}\right\rangle_{\mathcal{H}_{K^{(\mathrm{attn})}}}v_q^{(\mathrm{int})}\right\rangle_{\mathcal{H}_{K^{(\mathrm{attn})}}}. \quad (119)$$

Due to the fact that $h \in \mathcal{H}_{K^{(\mathrm{attn})}}$, $h = \sum_{q=1}^{\infty}\beta_q^{(h)}v_q^{(\mathrm{int})} = \sum_{q=1}^{\infty}\sqrt{\lambda_q^{(\mathrm{attn})}}\beta_q^{(h)}e_q$ with $v_q^{(\mathrm{int})} = \sqrt{\lambda_q^{(\mathrm{attn})}}e_q$. Therefore, $\|h\|_{L^2}^2 = \sum_{q=1}^{\infty}\lambda_q^{(\mathrm{attn})}\beta_q^{(h)^2}$, and

$$\left\|\sum_{q=1}^{Q}\sqrt{\lambda_q^{(\mathrm{attn})}}\left\langle h, v_q^{(\mathrm{int})}\right\rangle_{\mathcal{H}_{K^{(\mathrm{attn})}}}v_q^{(\mathrm{int})}\right\|_{\mathcal{H}_{K^{(\mathrm{attn})}}} = \left\|\sum_{q=1}^{Q}\sqrt{\lambda_q^{(\mathrm{attn})}}\beta_q^{(h)}v_q^{(\mathrm{int})}\right\|_{\mathcal{H}_{K^{(\mathrm{attn})}}}$$

$$= \sqrt{\sum_{q=1}^{Q}\lambda_q^{(\mathrm{attn})}\beta_q^{(h)^2}} \leq \|h\|_{L^2} \leq r_h. \quad (120)$$

According to Mercer's Theorem, because the kernel $K$ is continuous symmetric positive definite, it has the decomposition

$$K^{(\mathrm{attn})}(\cdot, \vec{\mathbf{x}}_i) = \sum_{j=1}^{\infty}\lambda_j^{(\mathrm{attn})}e_j(\cdot)e_j(\vec{\mathbf{x}}_i),$$

so that we have

$$
\begin{aligned}
\left\langle \sum_{i=1}^{n} \sigma_i K^{(\mathrm{attn})}(\cdot, \vec{\mathbf{x}}_i), v_q^{(\mathrm{int})} \right\rangle_{\mathcal{H}_{K^{(\mathrm{attn})}}} &= \left\langle \sum_{i=1}^{n} \sigma_i \sum_{j=1}^{\infty} \lambda_j^{(\mathrm{attn})} e_j e_j(\vec{\mathbf{x}}_i), v_q^{(\mathrm{int})} \right\rangle_{\mathcal{H}_{K^{(\mathrm{attn})}}} \\
&= \left\langle \sum_{i=1}^{n} \sigma_i \sum_{j=1}^{\infty} \sqrt{\lambda_j^{(\mathrm{attn})}} e_j(\vec{\mathbf{x}}_i) \cdot v_j^{(\mathrm{int})}, v_q^{(\mathrm{int})} \right\rangle_{\mathcal{H}_{K^{(\mathrm{attn})}}} \\
&= \sum_{i=1}^{n} \sigma_i \sqrt{\lambda_q^{(\mathrm{attn})}} e_q(\vec{\mathbf{x}}_i). \tag{121}
\end{aligned}
$$

Combining (119), (120), and (121), we have

$$
\left\langle h, \sum_{i=1}^{n} \sigma_i K^{(\mathrm{attn})}(\cdot, \vec{\mathbf{x}}_i) \right\rangle \overset{\text{①}}{\leq} \left\| \sum_{q=1}^{Q} \sqrt{\lambda_q^{(\mathrm{attn})}} \left\langle h, v_q^{(\mathrm{int})} \right\rangle_{\mathcal{H}_{K^{(\mathrm{attn})}}} v_q^{(\mathrm{int})} \right\|_{\mathcal{H}_{K^{(\mathrm{attn})}}} \cdot
$$

$$
\left\| \sum_{q=1}^{Q} \frac{1}{\sqrt{\lambda_q^{(\mathrm{attn})}}} \left\langle \sum_{i=1}^{n} \sigma_i K^{(\mathrm{attn})}(\cdot, \vec{\mathbf{x}}_i), v_q^{(\mathrm{int})} \right\rangle_{\mathcal{H}_{K^{(\mathrm{attn})}}} v_q^{(\mathrm{int})} \right\|_{\mathcal{H}_{K^{(\mathrm{attn})}}}
$$

$$
+ \|h\|_{\mathcal{H}_{K^{(\mathrm{attn})}}} \cdot \left\| \sum_{q=Q+1}^{\infty} \left\langle \sum_{i=1}^{n} \sigma_i K^{(\mathrm{attn})}(\cdot, \vec{\mathbf{x}}_i), v_q^{(\mathrm{int})} \right\rangle_{\mathcal{H}_{K^{(\mathrm{attn})}}} v_q^{(\mathrm{int})} \right\|_{\mathcal{H}_{K^{(\mathrm{attn})}}}
$$

$$
\leq \|h\|_{L^2} \left\| \sum_{q=1}^{Q} \sum_{i=1}^{n} \sigma_i e_q(\vec{\mathbf{x}}_i) v_q^{(\mathrm{int})} \right\|_{\mathcal{H}_{K^{(\mathrm{attn})}}} + B \left\| \sum_{q=Q+1}^{\infty} \sum_{i=1}^{n} \sigma_i \sqrt{\lambda_q^{(\mathrm{attn})}} e_q(\vec{\mathbf{x}}_i) v_q^{(\mathrm{int})} \right\|_{\mathcal{H}_{K^{(\mathrm{attn})}}}
$$

$$
\leq r_h \sqrt{ \sum_{q=1}^{Q} \left( \sum_{i=1}^{n} \sigma_i e_q(\vec{\mathbf{x}}_i) \right)^2 } + B \sqrt{ \sum_{q=Q+1}^{\infty} \left( \sum_{i=1}^{n} \sigma_i \sqrt{\lambda_q^{(\mathrm{attn})}} e_q(\vec{\mathbf{x}}_i) \right)^2 }, \tag{122}
$$

where ① is due to Cauchy-Schwarz inequality. Moreover, by Jensen's inequality we have

$$
\begin{aligned}
\mathbb{E} \left[ \sqrt{ \sum_{q=1}^{Q} \left( \sum_{i=1}^{n} \sigma_i e_q(\vec{\mathbf{x}}_i) \right)^2 } \right] &\leq \sqrt{ \mathbb{E} \left[ \sum_{q=1}^{Q} \left( \sum_{i=1}^{n} \sigma_i e_q(\vec{\mathbf{x}}_i) \right)^2 \right] } \\
&\leq \sqrt{ \mathbb{E} \left[ \sum_{q=1}^{Q} \sum_{i=1}^{n} e_q^2(\vec{\mathbf{x}}_i) \right] } = \sqrt{nQ}. \tag{123}
\end{aligned}
$$

and similarly,

$$
\mathbb{E} \left[ \sqrt{ \sum_{q=Q+1}^{\infty} \left( \sum_{i=1}^{n} \sigma_i \sqrt{\lambda_q^{(\mathrm{attn})}} e_q(\vec{\mathbf{x}}_i) \right)^2 } \right] \leq \sqrt{ \mathbb{E} \left[ \sum_{q=Q+1}^{\infty} \lambda_q^{(\mathrm{attn})} \sum_{i=1}^{n} e_q^2(\vec{\mathbf{x}}_i) \right] } = \sqrt{ n \sum_{q=Q+1}^{\infty} \lambda_q^{(\mathrm{attn})} }. \tag{124}
$$

Since (122)-(124) hold for all $Q \geq 0$, it follows that

$$
\mathbb{E} \left[ \sup_{h \in \mathcal{H}_{K^{(\mathrm{attn})}}(B), \|h\|_{L^2} \leq r_h} \frac{1}{n} \sum_{i=1}^{n} \sigma_i h(\vec{\mathbf{x}}_i) \right] \leq \min_{Q: Q \geq 0} \left( r_h \sqrt{nQ} + B \sqrt{ n \sum_{q=Q+1}^{\infty} \lambda_q^{(\mathrm{attn})} } \right). \tag{125}
$$

It follows from (117), (118), and (125) that

$$
\mathcal{R}_1 \leq \frac{1}{n} \mathbb{E} \left[ \sup_{h \in \mathcal{H}_{K^{(\mathrm{attn})}}(B), \|h\|_{L^2} \leq r_h} \sum_{i=1}^{n} \sigma_i h(\vec{\mathbf{x}}_i) \right]
$$

$$\leq \min_{Q \colon Q \geq 0} \left( r_h \sqrt{\frac{Q}{n}} + B \left( \frac{\sum_{q=Q+1}^{\infty} \lambda_q^{(\text{attn})}}{n} \right)^{1/2} \right). \tag{126}$$

**Derivation for the upper bound for $\mathcal{R}_2$.**

Because $\left| 1/n \sum_{i=1}^{n} \sigma_i e(\vec{\mathbf{x}}_i) \right| \leq w$ when $\|e\|_\infty \leq w$, we have

$$\mathcal{R}_2 \leq \frac{1}{n} \mathbb{E} \left[ \sup_{e \in L^\infty \colon \|e\|_\infty \leq w} \sum_{i=1}^{n} \sigma_i e(\vec{\mathbf{x}}_i) \right] \leq w. \tag{127}$$

It follows from (126) and (127) that

$$\Re \left( \left\{ f \colon f \in \mathcal{F}(B, w), \mathbb{E}_P \left[ f^2 \right] \leq r \right\} \right) \leq \min_{Q \colon Q \geq 0} \left( r_h \sqrt{\frac{Q}{n}} + B \left( \frac{\sum_{q=Q+1}^{\infty} \lambda_q^{(\text{attn})}}{n} \right)^{1/2} \right) + w.$$

Plugging $r_h$ in the RHS of the above inequality completes the proof. $\qquad\square$

**Lemma C.20** ((Yang & Li, 2024, Lemma B.9)). Suppose $\psi \colon [0, \infty) \to [0, \infty)$ is a sub-root function with the unique fixed point $r^*$. Then the following properties hold.

(1) Let $a \geq 0$, then $\psi(r) + a$ as a function of $r$ is also a sub-root function with fixed point $r_a^*$, and $r^* \leq r_a^* \leq r^* + 2a$.

(2) Let $b \geq 1$, $c \geq 0$ then $\psi(br + c)$ as a function of $r$ is also a sub-root function with fixed point $r_b^*$, and $r_b^* \leq br^* + 2c/b$.

(3) Let $b \geq 1$, then $\psi_b(r) = b\psi(r)$ is also a sub-root function with fixed point $r_b^*$, and $r_b^* \leq b^2 r^*$.

## C.7 PROOFS FOR THE APPROXIMATE UNIFORM CONVERGENCE FOR THE KERNEL $K^{(\text{attn})}$

In this subsection, we present the main theorem, Theorem C.22, regarding the approximate uniform convergence of $\widehat{K}^{(\text{attn})}(\cdot, \mathbf{x}')$ to $K^{(\text{attn})}(\cdot, \mathbf{x}')$ for every fixed $\mathbf{x}' \in \mathcal{X}$. Theorem C.22 is the formal version of Theorem 5.1 in Section 5.2. We first present below the concentration inequality for independent random variables taking values in a Hilbert space $\mathcal{B}$ of functions defined on a measurable space $(S, \Sigma_S, \mu_S)$. Let $\{f_k\}_{k=0}^{\infty}$ be a martingale a separable Banach space $(\mathcal{B}, \|\cdot\|)$ with respect to an increasing sequence of $\sigma$-algebras $\{\mathcal{F}_k\}_{n=0}^{\infty}$ and $f_0 = 0$. Define $d_k := f_k - f_{k-1}$ for $k \geq 1$, $d_0 = 0$, and $f^* := \sup_{k \geq 0} \|f_k\|$.

For a function $g \colon \mathcal{B} \to \mathbb{R}$, The first Gâteaux derivative of $g$ at a point $x \in \mathcal{B}$ along a direction $h \in \mathcal{B}$ is defined as

$$g'(x)(h) := \lim_{t \to 0} \frac{\|g(x + th)\| - \|g(x)\|}{t}.$$

The second Gâteaux derivative of $g$ at a point $x \in \mathcal{B}$ along two directions $h_1, h_2 \in \mathcal{B}$ is defined as

$$g''(x)(h_1, h_2) := \lim_{t \to 0} \frac{g'(x + th_2)(h_1) - g'(x)(h_1)}{t}.$$

The class $D(A_1, A_2)$ consists of Banach spaces $\mathcal{B}$ such that $\left| \|x\|'(\Delta) \right| \leq A_1 \|\Delta\|$ and $\left| \|x\|''(\Delta, \Delta) \right| \leq A_2 \|\Delta\|^2 / \|x\|$ hold for all $x, \Delta \in \mathcal{B}$ and $x \neq 0$.

**Lemma C.21** (Martingale based concentration inequality for Banach space-valued process (Pinelis, 1992, Theorem 2)). Suppose that $\sum_{k=1}^{\infty} \text{esssup} \|d_k\|^2 \leq 1$ where $\text{esssup}(f) = \inf_{a \in \mathbb{R}} \{ \mu(f^{-1}(a, +\infty)) = 0 \}$ for a function denotes the essential supremum of a function, and $\mathcal{B} \in D(A_1, A_2)$ or $\mathcal{B} \subseteq L^p(S, \Sigma, \mu)$ with $p \geq 2$. Then for every $r > 0$,

$$\Pr[f^* > r] \leq 2 \exp \left( -\frac{r^2}{2B} \right) \tag{128}$$

with $B = A_1^2 + A_2$ for $\mathcal{B} \in D(A_1, A_2)$, and $B = p - 1$ for $\mathcal{B} \subseteq L^p(S, \Sigma_S, \mu_S)$.

**Remark.** It is pointed out in (Pinelis, 1992) that when $\mathcal{B} \subseteq L^p(S, \Sigma, \mu)$, $\mathcal{B} \in D(1, p-1)$, so that $B = A_1^2 + A_2 = p$ with $A_1 = 1, A_2 = p - 1$. However, for such specific case that $\mathcal{B} \subseteq L^p(S, \Sigma, \mu)$, a sharp bound with $B = p - 1$ can be achieved (Pinelis, 1992).

**Theorem C.22.** For every fixed $\mathbf{x}' \in \mathcal{X}$ and every $\delta \in (0, 1)$, with probability at least $1 - \delta$ over $\mathbf{Q} = \left\{ \vec{\mathbf{q}}_i \right\}_{i=1}^{N}$, we have

$$\left\| \widehat{K}^{(\text{attn})}(\mathbf{x}, \mathbf{x}') - K^{(\text{attn})}(\mathbf{x}, \mathbf{x}') \right\|_{\infty} \lesssim \sqrt{\frac{\log 1/\delta}{N}}. \tag{129}$$

As a result, with probability at least $1 - \delta$ over $\mathbf{Q}$,

$$\sup_{\mathbf{x} \in \mathcal{X}, i \in [n]} \left| \widehat{K}^{(\text{attn})}(\mathbf{x}, \vec{\mathbf{x}}_i) - K^{(\text{attn})}(\mathbf{x}, \vec{\mathbf{x}}_i) \right| \lesssim \sqrt{\frac{\log(n/\delta)}{N}}, \tag{130}$$

$$\left\| \widehat{\mathbf{K}}^{(\text{attn})} - \mathbf{K}^{(\text{attn})} \right\|_2 \lesssim n \sqrt{\frac{\log(n/\delta)}{N}}. \tag{131}$$

*Proof.* We define

$$p(\mathbf{q}, \mathbf{x}') := \frac{1}{N} \sum_{j=1}^{N} K(\mathbf{q}, \vec{\mathbf{q}}_j) K(\vec{\mathbf{q}}_j, \mathbf{x}'), \quad \forall \mathbf{q}, \mathbf{x}' \in \mathcal{X}. \tag{132}$$

Since $\sup_{\mathbf{x}, \mathbf{x}' \in \mathcal{X}} |K(\mathbf{x}, \mathbf{x}')| \leq \frac{1+\pi}{2\pi} < 1$, we have $\sup_{\mathbf{q}, \mathbf{x}' \in \mathcal{X}} p(\mathbf{q}, \mathbf{x}') = \Theta(1)$. We now fix $\mathbf{x}' \in \mathcal{X}$ in the following arguments. It follows from (137) of Lemma C.23 that for every $t > 0$ and every $i \in [N]$,

$$\Pr \left[ \left\| \frac{1}{N} \sum_{j=1}^{N} K(\cdot, \vec{\mathbf{q}}_j) K(\vec{\mathbf{q}}_j, \mathbf{x}') - \bar{K}(\cdot, \mathbf{x}') \right\|_{\mathcal{H}_K} < t \right] \geq 1 - 2 \exp \left( -\Theta(Nt^2) \right), \tag{133}$$

where $\bar{K}(\cdot, \mathbf{x}') := \mathbb{E}_{\mathbf{q}} \left[ K(\cdot, \mathbf{q}) K(\mathbf{q}, \mathbf{x}') \right]$. The following arguments are conditioned on the event that (133) holds.

For each $i \in [N]$, we have $K(\cdot, \vec{\mathbf{q}}_i) \in \mathcal{H}_K$, and $\bar{K}(\cdot, \mathbf{x}') \in \mathcal{H}_K$. It follows from (133) that, for all $\mathbf{q} \in \mathcal{X}$,

$$\left| p(\mathbf{q}, \mathbf{x}') - \bar{K}(\mathbf{q}, \mathbf{x}') \right| = \left\langle p(\cdot, \mathbf{x}') - \bar{K}(\cdot, \mathbf{x}'), K(\cdot, \mathbf{q}) \right\rangle$$

$$\leq \left\| \frac{1}{N} \sum_{j=1}^{N} K(\cdot, \vec{\mathbf{q}}_j) K(\vec{\mathbf{q}}_j, \mathbf{x}') - \bar{K}(\cdot, \mathbf{x}') \right\|_{\mathcal{H}_K} \cdot \| K(\cdot, \mathbf{q}) \|_{\mathcal{H}_K} \leq t. \tag{134}$$

Define

$$\bar{K}^{(\text{attn})}(\mathbf{x}, \mathbf{x}') := \frac{1}{N} \sum_{j=1}^{N} K(\mathbf{x}, \vec{\mathbf{q}}_i) \bar{K}(\vec{\mathbf{q}}_i, \mathbf{x}'), \quad \forall \mathbf{x}, \mathbf{x}' \in \mathcal{X}. \tag{135}$$

Then it follows from the definition of $\widehat{K}^{(\text{attn})}$ in (5) that for all $\mathbf{x} \in \mathcal{X}$,

$$\left| \widehat{K}^{(\text{attn})}(\mathbf{x}, \mathbf{x}') - \bar{K}^{(\text{attn})}(\mathbf{x}, \mathbf{x}') \right| = \left| \frac{1}{N} \sum_{j=1}^{N} K(\mathbf{x}, \vec{\mathbf{q}}_i) p(\vec{\mathbf{q}}_i, \mathbf{x}') - \frac{1}{N} \sum_{j=1}^{N} K(\mathbf{x}, \vec{\mathbf{q}}_i) \bar{K}(\vec{\mathbf{q}}_i, \mathbf{x}') \right|$$

$$\leq \frac{1}{N} \sum_{j=1}^{N} \left| K(\mathbf{x}, \vec{\mathbf{q}}_i) \right| \left| p(\vec{\mathbf{q}}_i, \mathbf{x}') - \bar{K}(\vec{\mathbf{q}}_i, \mathbf{x}') \right| \leq t, \tag{135}$$

where the last inequality follows from (134).

Given the fixed $\mathbf{x}' \in \mathcal{X}$, we now approximate $K^{(\text{attn})}(\cdot, \mathbf{x}')$ by $\bar{K}^{(\text{attn})}(\cdot, \mathbf{x}')$. First, it can be verified from the definition of $\bar{K}$ that $\sup_{\mathbf{q}, \mathbf{x}' \in \mathcal{X}} \left| \bar{K}(\mathbf{q}, \mathbf{x}') \right| = \Theta(1)$. It then follows from (137) of Lemma C.23 that

$$\Pr \left[ \left\| \bar{K}^{(\text{attn})}(\cdot, \mathbf{x}') - \mathbb{E}_{\mathbf{q}} \left[ K(\cdot, \mathbf{q}) \bar{K}(\mathbf{q}, \mathbf{x}') \right] \right\|_{\mathcal{H}_K} > t \right] \leq 2 \exp \left( -\Theta(Nt^2) \right), \tag{136}$$

and we have $\mathbb{E}_{\mathbf{q}} \left[ K(\cdot, \mathbf{q}) \bar{K}(\mathbf{q}, \mathbf{x}') \right] = K^{(\text{attn})}(\cdot, \mathbf{x}')$. It follows from (135) and (136) that for with probability at least $1 - 4 \exp \left( -\Theta(Nt^2) \right)$, for all $\mathbf{x} \in \mathcal{X}$,

$$\left| \widehat{K}^{(\text{attn})}(\mathbf{x}, \mathbf{x}') - K^{(\text{attn})}(\mathbf{x}, \mathbf{x}') \right|$$

$$\leq \left| \widehat{K}^{(\text{attn})}(\mathbf{x}, \mathbf{x}') - \bar{K}^{(\text{attn})}(\mathbf{x}, \mathbf{x}') \right| + \left| \bar{K}^{(\text{attn})}(\mathbf{x}, \mathbf{x}') - K^{(\text{attn})}(\mathbf{x}, \mathbf{x}') \right|$$

$$\leq t + \left\| \bar{K}^{(\text{attn})}(\cdot, \mathbf{x}') - \mathbb{E}_{\mathbf{q}} \left[ K(\cdot, \mathbf{q}) \bar{K}(\mathbf{q}, \mathbf{x}') \right] \right\|_{\mathcal{H}_K} \cdot \| K(\cdot, \mathbf{x})) \|_{\mathcal{H}_K} \leq 2t,$$

which proves (129). (130) and (131) follow from (129) by the union bound. $\qquad \square$

**Lemma C.23.** Suppose that $p$ is a function defined on $\mathcal{X}$ and $\| p(\mathbf{x}) \|_\infty = \Theta(1)$. Then for every $r > 0$,

$$\Pr \left[ \left\| \frac{1}{N} \sum_{i=1}^{N} K(\cdot, \vec{\mathbf{q}}_i) p(\vec{\mathbf{q}}_i) - \mathbb{E}_{\mathbf{q}} \left[ K(\cdot, \mathbf{q}) p(\mathbf{q}) \right] \right\|_{\mathcal{H}_K} > r \right] \leq 2 \exp \left( -\Theta(Nr^2) \right). \tag{137}$$

*Proof.* Let $\mathcal{B} = \mathcal{H}_K \subseteq L^2(\mathbb{S}^{d-1}, \mu)$, then $\mathcal{B} \in D(1,1)$ (Pinelis, 1992). Let $p_0 = \| p(\mathbf{x}) \|_\infty = \Theta(1)$. We then construct the martingale $\{ f_k \}_{k \in [N]}$. For each $k \in [N]$, we define

$$f_k := \mathbb{E} \left[ \frac{1}{2p_0 \sqrt{N}} \sum_{i=1}^{N} \left( K(\cdot, \vec{\mathbf{q}}_i) p(\vec{\mathbf{q}}_i) - \mathbb{E}_{\mathbf{q}} \left[ K(\cdot, \mathbf{q}) p(\mathbf{q}) \right] \right) \middle| \mathcal{F}_k \right], \forall k \in [N],$$

where $\{ \mathcal{F}_k \}_{k=0}^{N}$ is an increasing sequence of $\sigma$-algebras, $\mathcal{F}_k$ is the $\sigma$-algebra generated by $\left\{ \vec{\mathbf{q}}_t \right\}_{t=1}^{k}$. $\mathcal{F}_0$ is the trivial $\sigma$-algebra so that $f_0 = 0$. We note that

$$f_N = \frac{1}{2p_0 \sqrt{N}} \sum_{i=1}^{N} \left( K(\cdot, \vec{\mathbf{q}}_i) p(\vec{\mathbf{q}}_i) - \mathbb{E}_{\mathbf{q}} \left[ K(\cdot, \mathbf{q}) p(\mathbf{q}) \right] \right),$$

$$d_k = f_k - f_{k-1} = \frac{1}{2p_0 \sqrt{N}} \left( K(\cdot, \vec{\mathbf{q}}_k) p(\vec{\mathbf{q}}_k) - \mathbb{E}_{\mathbf{q}} \left[ K(\cdot, \mathbf{q}) p(\mathbf{q}) \right] \right), \forall k \in [N],$$

and $f^* = \max_{k \in [N]} \| f_k \|$. For every $k \in [N]$, we have

$$\| d_k \|_{\mathcal{H}_K} = \left\| \frac{1}{2p_0 \sqrt{N}} \left( K(\cdot, \vec{\mathbf{q}}_k) p(\vec{\mathbf{q}}_k) - \mathbb{E}_{\mathbf{q}} \left[ K(\cdot, \mathbf{q}) p(\mathbf{q}) \right] \right) \right\|_{\mathcal{H}_K}$$

$$\overset{①}{\leq} \frac{1}{2p_0 \sqrt{N}} \left( p_0 \left\| K(\cdot, \vec{\mathbf{q}}_k) \right\|_{\mathcal{H}_K} + p_0 \mathbb{E}_{\mathbf{q}} \left[ \| K(\cdot, \mathbf{q}) \|_{\mathcal{H}_K} \right] \right) \overset{②}{\leq} \frac{1}{\sqrt{N}}, \tag{138}$$

where ① follows from the triangle inequality and the Jensen's inequality, and ② follows from the fact that $\left\| K(\cdot, \vec{\mathbf{q}}_k) \right\|_{\mathcal{H}_K} \leq \frac{1+\pi}{2\pi} < 1$.

It follows from (138) that $\sum_{k=1}^{\infty} \| d_k \|^2 \leq 1$. Applying Lemma C.21 with the martingale $\{ f_k \}_{k=0}^{N}$ and $\mathcal{B} = \mathcal{H}_K \subseteq L^2(\mathbb{S}^{d-1}, \mu)$, $B = 1$, we have $\Pr \left[ f^* = \max_{k \in [N]} \| f_k \| > r \right] \leq 2 \exp \left( -\frac{r^2}{2} \right)$, and

it follows that for every $r > 0$,

$$\Pr\left[\left\|\frac{1}{2p_0\sqrt{N}}\sum_{i=1}^{N}\left(K(\cdot,\vec{\mathbf{q}}_i)p(\vec{\mathbf{q}}_i) - \mathbb{E}_{\mathbf{q}}\left[K(\cdot,\mathbf{q})p(\mathbf{q})\right]\right)\right\|_{\mathcal{H}_K} > r\right] \leq 2\exp\left(-\frac{r^2}{2}\right),$$

and it follows that

$$\Pr\left[\left\|\frac{1}{N}\sum_{i=1}^{N}K(\cdot,\vec{\mathbf{q}}_i)p(\vec{\mathbf{q}}_i) - \mathbb{E}_{\mathbf{q}}\left[K(\cdot,\mathbf{q})p(\mathbf{q})\right]\right\|_{\mathcal{H}_K} > r\right] \leq 2\exp\left(-\Theta(Nr^2)\right),$$

which completes the proof of (137) and the constant in $\Theta(Nr^2)$ depends on $p_0 = \Theta(1)$.

$\square$

## D MORE RESULTS ABOUT UNIVERSITY CONVERGENCE AND INTEGRAL OPERATORS

### D.1 RESULTS ABOUT EIGENVALUES OF THE INTEGRAL OPERATORS

**Theorem D.1.** Let $\{e_j\}_{j\geq 1} \subseteq L^2(\mathcal{X},\mu)$ be a countable orthonormal basis of $L^2(\mathcal{X},\mu)$ which comprise the eigenfunctions of the integral operator $T_K\colon L^2(\mathcal{X},\mu) \to L^2(\mathcal{X},\mu), (T_Kf)(\mathbf{x}) \coloneqq \int_{\mathcal{X}} K(\mathbf{x},\mathbf{x}')f(\mathbf{x}')\mathrm{d}\mu(\mathbf{x}')$, a positive, self-adjoint, and compact operator on $L^2(\mathcal{X},\mu)$. Let $\{\lambda_j\}_{j\geq 1}$ with $\frac{1+\pi}{2\pi} \geq \lambda_1 \geq \lambda_2 \geq \ldots > 0$ such that $e_j$ is the eigenfunction of $T_K$ with $\lambda_j$ being the corresponding eigenvalue. Then $e_j$ is the eigenfunction of $T_{K^{(\mathrm{attn})}}$ with $\lambda_j^2$ being the corresponding eigenvalue. That is, $T_{K^{(\mathrm{attn})}}e_j = \lambda_j^3 e_j$, so that $\lambda_j^{(\mathrm{attn})} = \lambda_j^3$ for all $j \geq 1$.

*Proof.* First, since $T_K e_1 = \lambda_1 e_1$ it follows from the Cauchy-Schwarz inequality that

$$\lambda_1^2 = \|\lambda_1 e_1\|_{L^2(\mathcal{X},\mu)}^2 \leq \int_{\mathcal{X}\times\mathcal{X}} K(\mathbf{x},\mathbf{x}')^2 e_1^2(\mathbf{x}')\mathrm{d}\mu(\mathbf{x}')\mathrm{d}\mu(\mathbf{x}) \leq \left(\frac{1+\pi}{2\pi}\right)^2,$$

which proves that $\lambda_j \in (0, (1+\pi)/(2\pi)]$ for all $j \geq 1$. It follows from Mercer's theorem that

$$K(\mathbf{v},\mathbf{v}') = \sum_{j\geq 1}\lambda_j e_j(\mathbf{v})e_j(\mathbf{v}'), \quad \forall \mathbf{v},\mathbf{v}' \in \mathcal{X},$$

and the convergence on the RHS of the above equality is uniform and absolute. Then it follows from the definition of $K^{(\mathrm{attn})}$ in (4) that

$$K^{(\mathrm{attn})}(\mathbf{x},\mathbf{x}') = \int_{\mathcal{X}\times\mathcal{X}} K(\mathbf{x},\mathbf{v})K(\mathbf{v},\mathbf{v}')K(\mathbf{v}',\mathbf{x}')\mathrm{d}\mu(\mathbf{v})\otimes\mu(\mathbf{v}')$$

$$\stackrel{①}{=} \int_{\mathcal{X}}\left(\int_{\mathcal{X}}\sum_{j\geq 1}\lambda_j e_j(\mathbf{x})e_j(\mathbf{v})\cdot\sum_{j\geq 1}\lambda_j e_j(\mathbf{v})e_j(\mathbf{v}')\mathrm{d}\mu(\mathbf{v})\right)\cdot\sum_{j\geq 1}\lambda_j e_j(\mathbf{v}')e_j(\mathbf{x}')\mu(\mathbf{v}')$$

$$\stackrel{②}{=} \int_{\mathcal{X}}\left(\sum_{j\geq 1}\lambda_j^2 e_j(\mathbf{x})e_j(\mathbf{v}')\right)\cdot\sum_{j\geq 1}\lambda_j e_j(\mathbf{v}')e_j(\mathbf{x}')\mu(\mathbf{v}') \stackrel{③}{=} \sum_{j\geq 1}\lambda_j^3 e_j(\mathbf{x})e_j(\mathbf{x}'), \qquad (139)$$

where ① follows from the Fubini's Theorem, and ②,③ follow by the orthogonality of the orthogonal basis $\{e_j\}_{j\geq 1}$.

It follows from (139) that for all $j \geq 1$,

$$\left(T_{K^{(\mathrm{attn})}}e_j\right)(\mathbf{x}) = \int_{\mathcal{X}}\left(\sum_{j'\geq 1}\lambda_j^3 e_{j'}(\mathbf{x})e_{j'}(\mathbf{x}')\right)e_j(\mathbf{x}')\mathrm{d}\mu(\mathbf{x}') = \lambda_j^3 e_j(\mathbf{x}),$$

which proves that $\lambda_j^{(\mathrm{attn})} = \lambda_j^3$ for all $j \geq 1$. $\square$

It is known, such as (Du et al., 2019b, Theorem 3.1), that $\mathbf{K}_n$ is non-singular. Based on this fact, we have the following propositions showing that $\mathbf{K}_n^{(\text{attn})}$ is also non-singular.

**Proposition D.2.** If $\vec{\mathbf{x}}_i \neq \vec{\mathbf{x}}_j$ for all $i, j \in [n]$ and $i \neq j$, then $\mathbf{K}_n^{(\text{attn})}$ is also non-singular.

*Proof.* (Du et al., 2019b, Theorem 3.1) shows that $\mathbf{K}_n$ is non-singular. Define the feature mapping $\Phi(\mathbf{x}) := \left[ \sqrt{\lambda}_1 e_1(\mathbf{x}), \sqrt{\lambda}_2 e_2(\mathbf{x}), \ldots, \right]$. Since $[\mathbf{K}_n]_{ij} = 1/n \cdot \Phi(\vec{\mathbf{x}}_i)^\top \Phi(\vec{\mathbf{x}}_j)$, the non-singularity of $\mathbf{K}$ indicates that the feature maps on the data $\mathbf{S}$, $\left\{ \Phi(\vec{\mathbf{x}}_i) \right\}_{i=1}^n$, are linearly independent.

On the other hand, Theorem D.1 shows that the $\left\{ \lambda_j^3, e_j \right\}_{j \geq 1}$ are the eigenvalues and the corresponding eigenfunctions of the integral operator $T_{K^{(\text{attn})}}$. Let $\tilde{\Phi} := \left[ \lambda_1^{\frac{3}{2}} e_1(\mathbf{x}), \lambda_2^{\frac{3}{2}} e_2(\mathbf{x}), \ldots, \right]$. Then $\left[ \mathbf{K}_n^{(\text{attn})} \right]_{ij} = 1/n \cdot \tilde{\Phi}(\vec{\mathbf{x}}_i)^\top \tilde{\Phi}(\vec{\mathbf{x}}_j)$. Because $\left\{ \Phi(\vec{\mathbf{x}}_i) \right\}_{i=1}^n$ are linearly independent, it can be verified by definition that $\left\{ \tilde{\Phi}(\vec{\mathbf{x}}_i) \right\}_{i=1}^n$ are also linearly independent, so that $\mathbf{K}_n^{(\text{attn})}$ is not singular. $\square$

### D.2 PROOFS OF THEOREM C.8

We need the definition of $\varepsilon$-net in Definition D.1 for the proof of Theorem C.8.

*Definition* D.1 ($\varepsilon$-net). Let $(X, d)$ be a metric space and let $\varepsilon > 0$. A subset $N_\varepsilon(X, d)$ is called an $\varepsilon$-net of $X$ if for every point $x \in X$, there exists some point $y \in N_\varepsilon(X, d)$ such that $d(x, y) \leq \varepsilon$. The minimal cardinality of an $\varepsilon$-net of $X$, if finite, is denoted by $N(X, d, \varepsilon)$ and is called the covering number of $X$ at scale $\varepsilon$.

**Proof of Theorem C.8**. First, we have $\mathbb{E}_{\mathbf{w} \sim \mathcal{N}(\mathbf{0}, \kappa^2 \mathbf{I}_d)} [h(\mathbf{w}, \mathbf{u}, \mathbf{v})] = K(\mathbf{u}, \mathbf{v})$. For any $\mathbf{u} \in \mathcal{X}$, $\mathbf{v} \in \mathcal{X}$, and $s > 0$, define function class

$$\mathcal{H}_{\mathbf{u}, \mathbf{v}, s} := \left\{ h_\delta(\cdot, \mathbf{u}', \mathbf{v}') \colon \mathbb{R}^d \to \mathbb{R} \colon \mathbf{u}' \in \mathbf{B}(\mathbf{u}; s) \cap \mathcal{X}, \mathbf{v}' \in \mathbf{B}(\mathbf{v}; s) \cap \mathcal{X} \right\}, \qquad (140)$$

where $h_\delta(\mathbf{w}, \mathbf{u}', \mathbf{v}') = \sigma \left( \left[ \mathbf{w}^\top \mathbf{u}' \right]_{M_\delta} \right) \sigma \left( \left[ \mathbf{w}^\top \mathbf{v}' \right]_{M_\delta} \right)$ for all $\mathbf{w} \in \mathbb{R}^d$ and $\mathbf{u}', \mathbf{v}' \in \mathcal{X}$ with

$$M_\delta := \tilde{M}_\delta + \frac{\sqrt{2d + 3\log(3m/\delta)}}{m}, \quad \tilde{M}_\delta := \kappa\sqrt{2\log(2m)} + \kappa\sqrt{2\log(3/\delta)},$$

for $\delta \in (0, 1)$. Also, for all $a > 0$ $[\cdot]_a$ is defined as $[\cdot]_a := \text{sgn}(\cdot) \min\{|\cdot|, a\}$. We first build an $s$-net for the unit sphere $\mathcal{X}$. By (Vershynin, 2012, Lemma 5.2), there exists an $s$-net $N_s(\mathcal{X}, \|\cdot\|_2)$ of $\mathcal{X}$ such that $N(\mathcal{X}, \|\cdot\|_2, s) \leq \left( 1 + \frac{2}{s} \right)^d$.

In the sequel, a function in the class $\mathcal{H}_{\mathbf{u}, \mathbf{v}, s}$ is also denoted as $h_\delta(\mathbf{w})$, omitting the presence of variables $\mathbf{u}'$ and $\mathbf{v}'$ when no confusion arises. Let $P_m$ be the empirical distribution over $\left\{ \vec{\mathbf{w}}_r(0) \right\}$ so that $\mathbb{E}_{\mathbf{w} \sim P_m} [h_\delta(\mathbf{w})] = 1/m \cdot \sum_{r=1}^m h_\delta(\vec{\mathbf{w}}_r(0))$. Given $\mathbf{u} \in N(\mathcal{X}, s)$, we aim to estimate the upper bound for the supremum of empirical process $\mathbb{E}_{\mathbf{w} \sim \mathcal{N}(\mathbf{0}, \kappa^2 \mathbf{I}_d)} [h(\mathbf{w})] - \mathbb{E}_{\mathbf{w} \sim P_m} [h(\mathbf{w})]$ when function $h$ ranges over the function class $\mathcal{H}_{\mathbf{u}, \mathbf{v}, s}$. To this end, we apply Theorem A.1 to the function class $\mathcal{H}_{\mathbf{u}, \mathbf{v}, s}$ with $\mathbf{W}(0) = \left\{ \vec{\mathbf{w}}_r(0) \right\}_{r=1}^m$. Since $h_\delta(\cdot, \mathbf{u}', \mathbf{v}') \in [0, M_\delta^2]$ for any $h_\delta \in \mathcal{H}_{\mathbf{u}, \mathbf{v}, s}$, we set $a = 0, b = M_\delta^2, \alpha = 1/2$ in Theorem A.1, and $\text{Var}[h_\delta] \leq M_\delta^4$. As a result, with probability at least $1 - \delta$ over the random initialization $\mathbf{W}(0)$,

$$\sup_{\substack{\mathbf{u}' \in \mathbf{B}(\mathbf{u}; s) \cap \mathcal{X}, \\ \mathbf{v}' \in \mathbf{B}(\mathbf{v}; s) \cap \mathcal{X}}} |K_\delta(\mathbf{u}', \mathbf{v}') - \mathbb{E}_{\mathbf{w} \sim P_m} [h_\delta(\mathbf{w}, \mathbf{u}', \mathbf{v}')]| \leq 6\widehat{\mathcal{R}}(\mathcal{H}_{\mathbf{u}, \mathbf{v}, s}) + M_\delta^2 \sqrt{\frac{2\log \frac{2}{\delta}}{m}} + \frac{16 M_\delta^2 \log \frac{2}{\delta}}{3m},$$

$$(141)$$

where $K_\delta(\mathbf{u}', \mathbf{v}') := \mathbb{E}_{\mathbf{w}} [h_\delta(\mathbf{w}, \mathbf{u}', \mathbf{v}')], \widehat{\mathcal{R}}(\mathcal{H}_{\mathbf{u}, \mathbf{v}, s}) = \mathbb{E}_{\{\sigma_r\}_{r=1}^m} \left[ \sup_{h_\delta \in \mathcal{H}_{\mathbf{u}, \mathbf{v}, s}} \frac{1}{m} \sum_{r=1}^m \sigma_r h_\delta(\vec{\mathbf{w}}_r(0)) \right]$ is the empirical Rademacher complexity of the function class $\mathcal{H}_{\mathbf{u}, \mathbf{v}, s}$, $\{\sigma_r\}_{r=1}^m$ are i.i.d.

Rademacher random variables taking values of $\pm 1$ with equal probability. By Lemma D.3, $\widehat{\mathcal{R}}(\mathcal{H}_{\mathbf{u},\mathbf{v},s}) \le 2s\tilde{M}_\delta \max_{r \in [m]} \left\| \vec{\mathbf{w}}_r(0) \right\|_2$. Plugging such upper bound for $\widehat{\mathcal{R}}(\mathcal{H}_{\mathbf{u},\mathbf{v},s})$ in (141) and setting $s = \frac{1}{m}$, we have

$$
\sup_{\substack{\mathbf{u}' \in \mathbf{B}(\mathbf{u};s) \cap \mathcal{X}, \\ \mathbf{v}' \in \mathbf{B}(\mathbf{v};s) \cap \mathcal{X}}} |K_\delta(\mathbf{u}', \mathbf{v}') - \mathbb{E}_{\mathbf{w} \sim P_m}[h_\delta(\mathbf{w}, \mathbf{u}', \mathbf{v}')]| \le \frac{12 M_\delta \max_{r \in [M]} \left\| \vec{\mathbf{w}}_r(0) \right\|_2}{m} + M_\delta^2 \sqrt{\frac{2\log \frac{2}{\delta}}{m}} + \frac{16 M_\delta^2 \log \frac{2}{\delta}}{3m}.
$$
(142)

It follows from Lemma D.5 that with probability at least $1 - \delta$ over $\mathbf{W}(0)$,

$$
\max_{r \in [m]} \max \left\{ \left| \vec{\mathbf{w}}_r(0)^\top \mathbf{v} \right|, \left| \vec{\mathbf{w}}_r(0)^\top \mathbf{u} \right| \right\} \le \tilde{M}_\delta,
$$

$$
\max_{r \in [m]} \left\| \vec{\mathbf{w}}_r(0) \right\|_2^2 \le d + 2\sqrt{d \log(3m/\delta)} + 2\log(3m/\delta) \le 2d + 3\log(3m/\delta).
$$

As a result, with probability at least $1 - \delta$ over $\mathbf{W}(0)$,

$$
\max_{r \in [m]} \sup_{\substack{\mathbf{u}' \in \mathbf{B}(\mathbf{u};s) \cap \mathcal{X}, \\ \mathbf{v}' \in \mathbf{B}(\mathbf{v};s) \cap \mathcal{X}}} \left\{ \left| \vec{\mathbf{w}}_r(0)^\top \mathbf{v}' \right|, \left| \vec{\mathbf{w}}_r(0)^\top \mathbf{u}' \right| \right\} \le \tilde{M}_\delta + \frac{\sqrt{2d + 3\log(3m/\delta)}}{m} = M_\delta. \quad (143)
$$

When (143) holds, $h_\delta = h$ and $\mathbb{E}_{\mathbf{w} \sim P_m}[h_\delta(\mathbf{w}, \mathbf{u}', \mathbf{v}')] = \widehat{h}(\mathbf{W}(0), \mathbf{u}', \mathbf{v}')$, $K_\delta(\mathbf{u}', \mathbf{v}') = K(\mathbf{u}', \mathbf{v}')$. It then follows from (142), (143), and the union bound that with probability at least $1 - 2\delta$ over $\mathbf{W}(0)$,

$$
\sup_{\substack{\mathbf{u}' \in \mathbf{B}(\mathbf{u};s) \cap \mathcal{X}, \\ \mathbf{v}' \in \mathbf{B}(\mathbf{v};s) \cap \mathcal{X}}} \left| K(\mathbf{u}', \mathbf{v}') - \widehat{h}(\mathbf{W}(0), \mathbf{u}', \mathbf{v}') \right| \le \frac{12 M_\delta (2d + 3\log(3m/\delta))}{m} + M_\delta^2 \sqrt{\frac{2\log \frac{2}{\delta}}{m}}
$$

$$
+ \frac{16 M_\delta^2 \log \frac{2}{\delta}}{3m}. \quad (144)
$$

By the union bound, with probability at least $1 - 2(1 + 2m)^{2d}\delta$ over $\mathbf{W}(0)$, (144) holds for arbitrary $\mathbf{u}, \mathbf{v} \in N(\mathcal{X}, s)$. In this case, for any $\mathbf{u}' \in \mathcal{X}, \mathbf{v}' \in \mathcal{X}$, there exists $\mathbf{u}, \mathbf{v} \in N_s(\mathcal{X}, \|\cdot\|_2)$ such that $\|\mathbf{u}' - \mathbf{u}\|_2 \le s, \|\mathbf{v}' - \mathbf{v}\|_2 \le s$, so that $\mathbf{u}' \in \mathbf{B}(\mathbf{u};s) \cap \mathcal{X}, \mathbf{v}' \in \mathbf{B}(\mathbf{v};s) \cap \mathcal{X}$, and (144) holds.

Changing the notations $\mathbf{u}', \mathbf{v}'$ to $\mathbf{u}, \mathbf{v}$, (47) is proved by the union bound.

$\square$

**Lemma D.3.** Let $\widehat{\mathcal{R}}(\mathcal{H}_{\mathbf{u},\mathbf{v},s}) := \mathbb{E}_{\{\sigma_r\}_{r=1}^m} \left[ \sup_{h \in \mathcal{H}_{\mathbf{u},\mathbf{v},s}} \frac{1}{m} \sum_{r=1}^m \sigma_r h(\vec{\mathbf{w}}_r(0)) \right]$ be the Rademacher complexity of the function class $\mathcal{H}_{\mathbf{u},\mathbf{v},s}$, and $B$ is a positive constant. Then

$$
\widehat{\mathcal{R}}(\mathcal{H}_{\mathbf{u},\mathbf{v},s}) \le 2s M_\delta \max_{r \in [m]} \left\| \vec{\mathbf{w}}_r(0) \right\|_2. \quad (145)
$$

*Proof.* We have

$$
\widehat{\mathcal{R}}(\mathcal{H}_{\mathbf{u},\mathbf{v},s}) = \mathbb{E}_{\{\sigma_r\}_{r=1}^m} \left[ \sup_{\mathbf{u}' \in \mathbf{B}(\mathbf{u};s) \cap \mathcal{X}, \mathbf{v}' \in \mathbf{B}(\mathbf{v};s) \cap \mathcal{X}} \frac{1}{m} \sum_{r=1}^m \sigma_r h_\delta(\vec{\mathbf{w}}_r(0), \mathbf{u}', \mathbf{v}') \right] \le \mathcal{R}_1 + \mathcal{R}_2 + \mathcal{R}_3,
$$
(146)

where

$$
\mathcal{R}_1 = \mathbb{E}_{\{\sigma_r\}_{r=1}^m} \left[ \sup_{\mathbf{u}' \in \mathbf{B}(\mathbf{u};s) \cap \mathcal{X}, \mathbf{v}' \in \mathbf{B}(\mathbf{v};s) \cap \mathcal{X}} \frac{1}{m} \sum_{r=1}^m \sigma_r \left( h_\delta(\vec{\mathbf{w}}_r(0), \mathbf{u}', \mathbf{v}') - h_\delta(\vec{\mathbf{w}}_r(0), \mathbf{u}, \mathbf{v}') \right) \right],
$$

$$
\mathcal{R}_2 = \mathbb{E}_{\{\sigma_r\}_{r=1}^m} \left[ \sup_{\mathbf{u}' \in \mathbf{B}(\mathbf{u};s) \cap \mathcal{X}, \mathbf{v}' \in \mathbf{B}(\mathbf{v};s) \cap \mathcal{X}} \frac{1}{m} \sum_{r=1}^m \sigma_r \left( h_\delta(\vec{\mathbf{w}}_r(0), \mathbf{u}, \mathbf{v}') - h_\delta(\vec{\mathbf{w}}_r(0), \mathbf{u}, \mathbf{v}) \right) \right],
$$

$$\mathcal{R}_3 = \mathbb{E}_{\{\sigma_r\}_{r=1}^m} \left[ \sup_{\mathbf{u}' \in \mathbf{B}(\mathbf{u};s) \cap \mathcal{X}, \mathbf{v}' \in \mathbf{B}(\mathbf{v};s) \cap \mathcal{X}} \frac{1}{m} \sum_{r=1}^m \sigma_r h_\delta(\vec{\mathbf{w}}_r(0), \mathbf{u}, \mathbf{v}) \right]. \tag{147}$$

Here (146) follows from the subadditivity of supremum. Now we bound $\mathcal{R}_1$, $\mathcal{R}_2$, and $\mathcal{R}_3$ separately. First, $\mathcal{R}_3 = 0$ by the definition of the Rademacher variables. For $\mathcal{R}_1$, we have

$$\left| h_\delta(\vec{\mathbf{w}}_r(0), \mathbf{u}', \mathbf{v}') - h_\delta(\vec{\mathbf{w}}_r(0), \mathbf{u}, \mathbf{v}') \right|$$

$$\leq \left| \sigma\left( \left[ \vec{\mathbf{w}}_r(0)^\top \mathbf{u}' \right]_{M_\delta} \right) \sigma\left( \left[ \vec{\mathbf{w}}_r(0)^\top \mathbf{v}' \right]_{M_\delta} \right) - \sigma\left( \left[ \vec{\mathbf{w}}_r(0)^\top \mathbf{u} \right]_{M_\delta} \right) \sigma\left( \left[ \vec{\mathbf{w}}_r(0)^\top \mathbf{v}' \right]_{M_\delta} \right) \right|$$

$$\leq s \left\| \vec{\mathbf{w}}_r(0) \right\|_2 \left| \sigma\left( \left[ \vec{\mathbf{w}}_r(0)^\top \mathbf{v}' \right]_{M_\delta} \right) \right| \leq s M_\delta \max_{r \in [m]} \left\| \vec{\mathbf{w}}_r(0) \right\|_2. \tag{148}$$

It follows from (148) that

$$\mathcal{R}_1 = \mathbb{E}_{\{\sigma_r\}_{r=1}^m} \left[ \sup_{\mathbf{u}' \in \mathbf{B}(\mathbf{u};s) \cap \mathcal{X}, \mathbf{v}' \in \mathbf{B}(\mathbf{v};s) \cap \mathcal{X}} \frac{1}{m} \sum_{r=1}^m \sigma_r \left( h_\delta(\vec{\mathbf{w}}_r(0), \mathbf{u}', \mathbf{v}') - h_\delta(\vec{\mathbf{w}}_r(0), \mathbf{u}, \mathbf{v}') \right) \right]$$

$$\leq \mathbb{E}_{\{\sigma_r\}_{r=1}^m} \left[ \frac{1}{m} \sum_{r=1}^m \left| h_\delta(\vec{\mathbf{w}}_r(0), \mathbf{u}', \mathbf{v}') - h_\delta(\vec{\mathbf{w}}_r(0), \mathbf{u}, \mathbf{v}') \right| \right] \leq s M_\delta \max_{r \in [m]} \left\| \vec{\mathbf{w}}_r(0) \right\|_2. \tag{149}$$

Applying the argument for $\mathcal{R}_1$ to $\mathcal{R}_2$, we have $\mathcal{R}_2 \leq s M_\delta \max_{r \in [m]} \left\| \vec{\mathbf{w}}_r(0) \right\|_2$. Plugging such upper bound for $\mathcal{R}_2$, (149), and $\mathcal{R}_3 = 0$ in (146), (145) is proved.

$$\square$$

**Lemma D.4.** Let $\mathbf{w} \sim \mathcal{N}(\mathbf{0}, \kappa^2 \mathbf{I}_d)$ with $\kappa > 0$. Then for any $\varepsilon \in (0, 1)$ and fixed $\mathbf{u} \in \mathcal{X}$, $\Pr\left[ \frac{|\mathbf{u}^\top \mathbf{w}|}{\|\mathbf{w}\|_2} \leq \varepsilon \right] \leq B\sqrt{d}\varepsilon$ where $B$ is an absolute positive constant, and $B$ can be set to $\pi^{-1/2}$.

*Proof.* Let $z = \frac{\mathbf{u}^\top \mathbf{w}}{\|\mathbf{w}\|_2}$. It can be verified that $z^2 \sim z_1$ where $z_1$ is a random variable following the Beta distribution $\mathrm{Beta}(\frac{1}{2}, \frac{d-1}{2})$. Therefore, the distribution of $z$ has the following continuous probability density function $p_z$ with respect to the Lebesgue measure,

$$p_z(x) = (1 - x^2)^{\frac{d-3}{2}} \mathbb{I}_{\{|x| \leq 1\}} / B', \tag{150}$$

where $B' = \int_{-1}^1 (1 - x^2)^{\frac{d-3}{2}} \mathrm{d}x$ is the normalization factor. It can be verified by standard calculation that $1/B' \leq B\sqrt{d}/2$ for an absolute positive constant $B$. Since $1 - x^2 \leq 1$ over $x \in [-1, 1]$, we have

$$\Pr\left[ \frac{|\mathbf{u}^\top \mathbf{w}|}{\|\mathbf{w}\|_2} \leq \varepsilon \right] = \Pr\left[ -\varepsilon \leq z \leq \varepsilon \right] = \frac{1}{B'} \int_{-\varepsilon}^\varepsilon (1 - x^2)^{\frac{d-3}{2}} \mathrm{d}x \leq B\sqrt{d}\varepsilon, \tag{151}$$

where the last inequality is due to the fact that $1 - x^2 \leq 1$ for $x \in [-\varepsilon, \varepsilon]$ with $\varepsilon \in (0, 1)$. $\square$

For every $\mathbf{v} \in \mathbb{S}^{d-1}$, noting that $\vec{\mathbf{w}}_r(0)^\top \mathbf{v} \sim \mathcal{N}(0, \kappa^2)$, we have the following standard upper bound for $\max_{r \in [m]} \left| \vec{\mathbf{w}}_r(0)^\top \mathbf{v} \right|$.

**Lemma D.5.** For every fixed $\mathbf{u} \in \mathbb{R}^d$ and $\delta \in (0, 1)$, with probability at least $1 - \delta$ over $\mathbf{W}(0)$,

$$\max_{r \in [m]} \left| \vec{\mathbf{w}}_r(0)^\top \mathbf{v} \right| \leq \kappa \sqrt{2 \log(2m)} + \kappa \sqrt{2 \log \frac{1}{\delta}}. \tag{152}$$

Moreover, it follows from Lemma D.6 that with probability at least $1 - \delta$ over $\mathbf{W}(0)$, $\max_{r \in [m]} \left\| \vec{\mathbf{w}}_r(0) \right\|_2^2 \leq d + 2\sqrt{d \log(m/\delta)} + 2 \log(m/\delta)$.

**Lemma D.6.** ((Laurent & Massart, 2000, Lemma 1)) Let $\{X_i\}_{i=1}^k$ be i.i.d. standard Gaussian random variables and $X = \sum_{i=1}^k X_i^2$, then

$$\Pr\left[X - k \geq 2\sqrt{kx} + 2x\right] \leq \exp(-x)$$

$$\Pr\left[k - X \geq 2\sqrt{kx}\right] \leq \exp(-x) \tag{153}$$

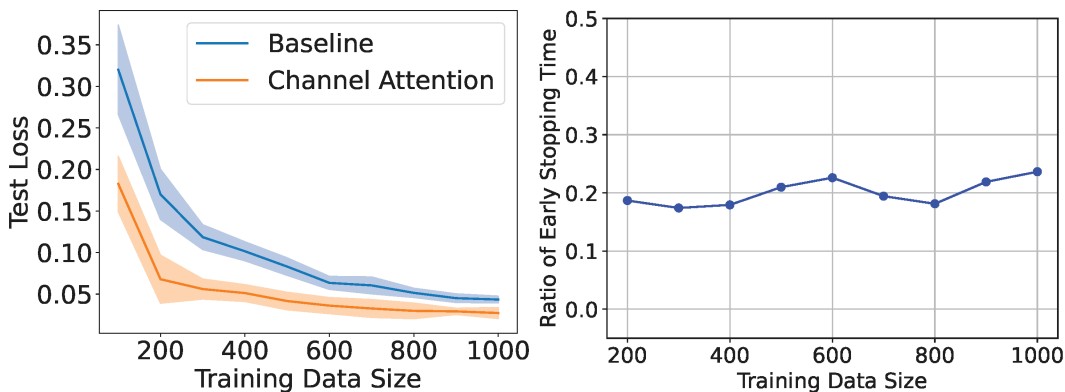

Figure 1: Left: illustration of the test loss by the vanilla network and the network with the proposed channel attention for varying $n$ in $[100, 1000]$ with a step size of $100$. The shaded area in each plot indicates the standard deviation across $10$ random initializations of the neural network. Right: illustration of the ratio of early stopping time.

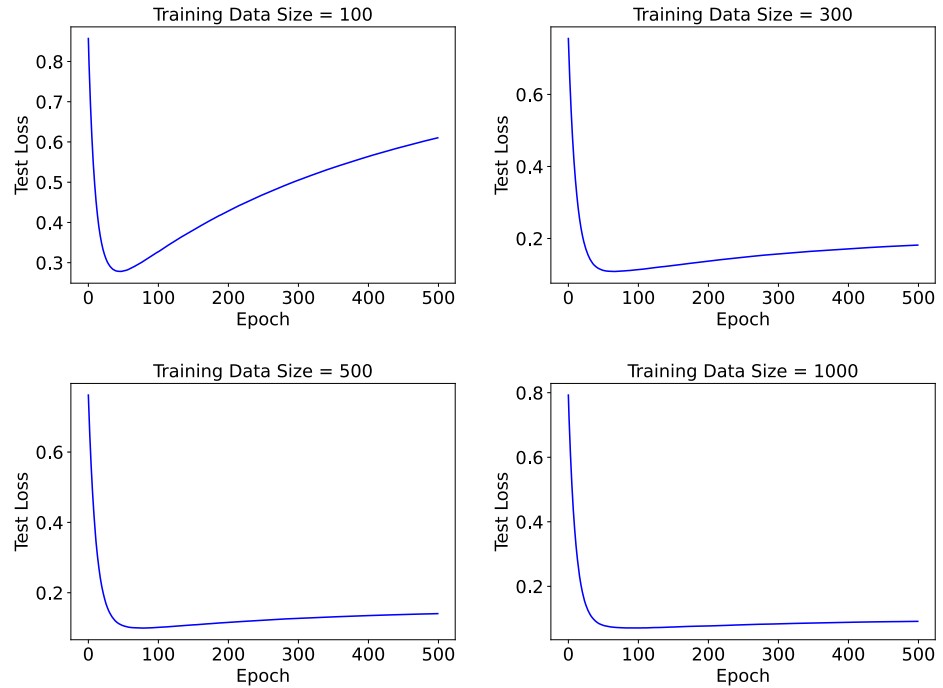

Figure 2: Illustration of the test loss by GD, averaged over 10 random initializations of the neural network.

# E    SIMULATION RESULTS

We provide simulation results on both synthetic data and real data in this section.

### E.1 RESULTS ON SYNTHETIC DATA

We present simulation results on synthetic data in this section. We randomly sample $n$ points $\left\{\vec{\mathbf{x}}_i\right\}_{i=1}^{n}$ distributed uniformly on the unit sphere $\mathbb{S}^{49}$ in $\mathbb{R}^{50}$. The sample size $n$ ranges from 100 to 1000 with a step size of 100. We set the target function as $f^*(\mathbf{x}) = \mathbf{s}^\top \mathbf{x}$, where $\mathbf{x} \in \mathbb{S}^{49}$ and $\mathbf{s} \sim \mathrm{Unif}\,(\mathcal{X})$ is randomly sampled. The noise variance is set to $\sigma_0^2 = 1$. We also uniformly and independently sample 1000 points on the unit sphere in $\mathbb{R}^{50}$ to form the test set. The two-layer NN with channel attention (1) is trained by Algorithm 1 with network width $m = 10000$, sample size $N = 10000$ for $\mathbf{Q}$, and learning rate $\eta = 1$. The training is executed on an NVIDIA A100 GPU, and the test loss is reported in Figure 1 and Figure 2. From Figure 1, it is evident that the network with channel attention (1) consistently exhibits better generalization than the vanilla two-layer neural network without such an attention mechanism, that is, $f^{(\mathrm{vanilla})}$ (3), by achieving lower test losses across different training data sizes. Figure 2 presents the test loss as a function of GD steps for $n = 100, 300, 500, 1000$. As shown in Figure 2, early stopping reliably improves generalization in neural network training, since the test loss initially decreases and later increases due to overfitting.

For each $n \in \{100, 200, \ldots, 1000\}$, we denote the GD step that attains the minimum test loss as $\widehat{t}_n$, which acts as the empirical early stopping time. We note that the early stopping time theoretically predicted by Theorem 3.2 scales as $\widehat{T} \asymp n^{\frac{6\alpha}{6\alpha+1}} \asymp n^{\frac{3d}{4d-1}}$ with $2\alpha = d/(d-1)$. We compute the ratio of early stopping time, defined as $\widehat{t}_n/n^{\frac{3d}{4d-1}}$ and averaged over 10 random neural network initializations for each $n$, and display it in the right plot of Figure 1. It can be observed that the ratio of early stopping time is relatively stable and lies within $[0.17, 0.23]$ with respect to different training data sizes, suggesting that the theoretically predicted early stopping time is indeed empirically proportional to the empirical early stopping time.

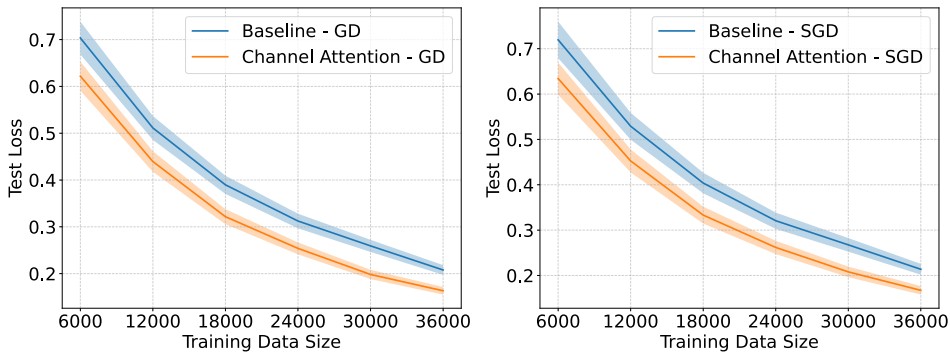

Figure 3: Illustration of the test loss by the vanilla network and the network with the proposed channel attention for varying $n$ in $[100, 1000]$ with a step size of 100. The shaded area in each plot indicates the standard deviation across 10 random initializations of the neural network.

### E.2 RESULTS ON REAL DATA (MINI-IMAGENET)

We herein provide additional empirical results for the two-layer NN with channel attention (1) trained on a real dataset, mini-ImageNet (Vinyals et al., 2016) with 60000 images from 100 classes, where the training features follow a complex distribution rather than the spherical uniform distribution and the target function may not lie in a RKHS ball of bounded radius associated with the neural network. The literature such as (Yu et al., 2025) shows that the class labels of such dataset cannot be explained by the NTK of the neural network, so that the target function is not in the RKHS associated with the NTK, and it is not in the interpolation space studied in this paper either. We use 60 classes comprising 36000 images as the training features and the remaining classes as the test data, and the size of the sample $\mathbf{Q}$ is set to be three times the size of the training data. We sample $\mathbf{Q}$ from a DiT (Peebles & Xie, 2023) trained on the training data. Figure 3 illustrates the test loss of the vanilla network, $f^{(\mathrm{vanilla})}$, and our two-layer NN with attention channel (1) with respect to different training data sizes where the one-hot class labels serve as the response vectors for regression. It can

be observed that network with channel attention always outperforms the vanilla network without channel attention with lower test losses, when both networks are trained by GD or standard SGD.

