# OpenReview forum: "Sharp Generalization for Nonparametric Regression in Interpolation Space by Shallow Neural Networks with Channel Attention"
_ICLR.cc/2026/Conference — ICLR 2026 Conference Withdrawn Submission_

### Official Review · Reviewer_SLNB · 2025-11-01

**Soundness:** 3
**Presentation:** 2
**Contribution:** 1
**Rating:** 2
**Confidence:** 3

**Summary:**

This paper studies the performance of a two-layered neural network with a channel attention architecture when the covariates are generated by an arbitrary distribution over the sphere.
The authors show that having the additional channel attention part allows the network to escape the NTK regime, and to be represented by an 'attention kernel'.

Given wide enough hidden layer and attention layer, under the eigenvalue decay rate of the covariates $j^{-2\alpha}$, the network trained by an early-stopped gradient descent achieves risk bound $O(n^{-\frac{6\alpha}{6\alpha + 1}})$, which improves upon prior work.

Among the techniques used in the proof are martingale based concentration inequalities to derive the uniform convergence of the empirical attention kernel to the attention kernel and local Rademacher complexity to control the function classes obtained during gradient descent.

**Strengths:**

Authors introduce a new model with channel attention that goes beyond linear NTK regime. Understanding the effect of various attention mechanisms is an important question, motivated by its prevalent use in practice.

Theoretical results are rigorously presented and improve upon prior work. Used techniques seem to be applicable to other network architectures for the future research.

Furthermore, experimental results are present that support theoretical findings, showing that the neural network with channel attention outperforms classical two-layered neural network in terms of sample complexity.

**Weaknesses:**

My main concern is the novelty of technical contribution in this paper compared to prior work. The authors claim in the abstract:

*Our analysis is based on two key technical contributions*. *First, we establish a principled decomposition of the network output <...> Second, building on this decomposition, we employ local Rademacher complexity to obtain sharp bound <...>*.

Later, in lines 424-426: *we introduce a new technique based on local Rademacher complexity to obtain a tight bound on the Rademacher complexity of the function class formed by all neural network functions generated through GD iterations*.

However, (Yang' 2025), which studies an overparameterized two-layered network without channel attention, also follows the same proof technique in terms of the output function decomposition and the use of local Rademacher complexity.

Current work would benefit from a clear comparison with techniques introduced in prior works, such as (Yang'25), and in highlighting what is exactly novel in the present analysis.

Also, the general clarity can be improved, the proof strategy in the Appendix is complicated and a clear roadmap with a figure of lemma dependencies would help the readers.

Finally, the particular proposed channel attention mechanism in Section 2.1 is not well-motivated.
It appears that the attention matrix A is independent of the input and only depends of the data distribution $P$.

*Yingzhen Yang. Sharp generalization for nonparametric regression by over-parameterized neural
networks: A distribution-free analysis in spherical covariate. In International Conference on
Machine Learning (ICML), 2025.*

**Questions:**

See 'Weaknesses' above: what is the novel technical contribution, compared to (Yang'25)?

Minor:

How is Lemma C.13 used? I only found some intermediate results, such as Equations (100, 101) used in other proofs, but not Lemma C.13 itself.

Lines 160-161: should be $\sigma(W(0), Q) \in R^{m \times N}$ (instead of $m \times n$), and furthermore, $\sigma(W(0), Q)^{(i)} = \sigma(W(0), q_i)$, instead of $\sigma(W(0), x_i)$?

In Lines 340-342, I believe it is a typo, as vector $a$ is optimized and should be initialized at 0?
Also, (10) misses vector $a$ in the loss function.

Lemma D.3: it should be empirical Rademacher complexity?

---

### Official Review · Reviewer_QSih · 2025-11-01

**Soundness:** 3
**Presentation:** 3
**Contribution:** 3
**Rating:** 8
**Confidence:** 2

**Summary:**

This paper analyzes two-layer neural networks with channel attention for nonparametric regression, proving a sharp rate that improves prior distribution-free bounds through novel attention kernel analysis. This work establishes the first theoretical benefit of channel attention for nonparametric regression in interpolation spaces.

**Strengths:**

This is a well-written, rigorous mathematical paper addressing an important theoretical problem.

1) The authors report a substantial rate improvement over Yang (2025) for the same distribution-free setting. They prove that for a two-layer networks with channel attention achieve improved nonparametric regression rates O$(n^{-6\alpha/6\alpha+1})$ via novel attention kernel analysis.
2) Authors also use innovative new proof techniques: principled network decomposition into RKHS component plus bounded residual (Theorem 5.2), and tight generalization bounds via local Rademacher complexity tailored to the attention kernel.

**Weaknesses:**

1) Contributions: they need to be stated a bit more clearly I think (can be structured better), right now they are mixed in with related work.
2) There should be some empirical validation of the claim in the main paper: despite it being a theoretical paper, I think there should some experimental validation supporting the authors' claim. Additionally, a schematic figure to explain their contribution might also be helpful.
3) Unclear Attention Kernel Choice: Why is the triple product $K(x,v)K(v,v')K(v',x')$ in Eq. (4) the right composition? Paper shows it works but provides limited intuition. Are other kernel compositions viable?
4) I think there is some substantial gap in sample complexity bounds (m,n) wrt to the experimental setup that is tested (n=10000). I am worried about the practical relevance of the theoretical guarantees?

**Questions:**

NA

---

### Official Review · Reviewer_6dmV · 2025-11-01

**Soundness:** 3
**Presentation:** 3
**Contribution:** 3
**Rating:** 6
**Confidence:** 3

**Summary:**

This paper study nonparametric regression using an overparameterized two-layer ReLU neural network equipped with XCA-style channel attention and trained via gradient descent. Unlike traditional self-attention mechanisms that operate over tokens, the XCA-style attention applies self-attention across channels. It is assumed that the data following any arbitrary continuous distribution on the unit sphere. Additionally, the target function, denoted as $f^\ast$ is assumed to lie within a subset of the RKHS induced by the attention kernel defined in equation (4), and it is shown that such a $f^\ast$ has strong spectral bias.

The authors establish a nonparametric regression excess risk bound (under early stopping) of order $\mathcal{O}(\varepsilon_n^2)$, where $\varepsilon_n$ denotes the critical population rate associated with the kernel induced by the network with channel attention. They further provide a comprehensive comparison with related theoretical results, demonstrating that the obtained convergence rate surpasses the current state-of-the-art in nonparametric regression analysis.

**Strengths:**

1. As a theory paper, this work showcases substantial technical depth, which I believe meet the standards expected of ICLR. In particular, establishing the uniform convergence of the empirical attention kernel to its population counterpart, when training a two-layer ReLU network with channel attention via gradient descent, is a nontrivial and technically demanding result.

2. To the best of my knowledge, this work is the first to provide a rigorous theoretical support for the benefits of incorporating XCA-style channel attention in ReLU neural networks. The results show that, without the channel attention, training a standard ReLU network fails to achieve comparably sharp excess risk bounds. In contrast, the network with XCA attention achieves rate even faster than the state-of-the-art results, clearly highlighting the theoretical advantage of channel-based self-attention in overparameterized neural networks.

3. I appreciate the authors for providing detailed comparisons with related theoretical results in nonparametric regression of neural networks. I also appreciate how the authors articulate how their framework extends beyond the typical NTK regime, emphasizing the novel aspects introduced by attention mechanisms

**Weaknesses:**

1. In Theorem 3.2, the authors prove that when the data distribution $P$ admits a polynomial eigenvalue decay rate (EDR), the excess risk bound achieves the rate $\mathcal{O}(n^{-\frac{6\alpha}{ 6\alpha +1}})$, which is indeed sharper than many existing results. However, the paper does not provide explicit examples of distributions that satisfy this EDR condition. For instance, would common distributions such as the Gaussian distribution on the unit sphere meet this requirement? Without a concrete example or discussion, it is difficult for readers to assess the practical relevance of such assumption.

2. The generalization guarantees critically depend on the early stopping rule and the overparameterized structure of the neural network. Can the authors comment on the sensitivity of the excess risk bound to these factors? This would help readers understand the robustness of the result and its relevance to practical neural network training.

**Questions:**

See weaknesses

---

### Official Review · Reviewer_m9fJ · 2025-11-01

**Soundness:** 3
**Presentation:** 3
**Contribution:** 3
**Rating:** 4
**Confidence:** 3

**Summary:**

The paper studies a two-layer ReLU network with **channel attention**. From the vanilla two-layer NTK kernel $K$, the authors define an **“attention kernel”** $K^{(\mathrm{attn})} = K * K * K$ (a triple convolution), implying eigenvalues $\lambda_j^{(\mathrm{attn})} = \lambda_j^3$ with the same eigenfunctions as $K$. They identify the target class as the RKHS ball $\mathcal H_{K^{(\mathrm{attn})}} = [\mathcal H_K]^3$ (an interpolation space biased more strongly to low frequency). The central claim is that **full-batch GD** on the channel-attention model, with a **data-dependent early stopping rule**, is *approximately equivalent* to kernel GD with kernel $K^{(\mathrm{attn})}$, yielding an excess risk of order $\tilde O(\varepsilon_n^2)$, where $\varepsilon_n$ is the critical radius of $K^{(\mathrm{attn})}$. The paper structures the proof via: (i) a uniform convergence result for the empirical attention kernel $\hat K^{(\mathrm{attn})}$, (ii) a trajectory decomposition $f_t = h_t + e_t$ with $h_t \in \mathcal H_{K^{(\mathrm{attn})}}$ and small $|e_t\|_\infty$, (iii) local Rademacher bounds + an early-stopping fixed point.

**Strengths:**

**Conceptual clarity on spectral bias:** The channel-attention construction leads to a **spectral sharpening** $\lambda_j \mapsto \lambda_j^3$, making the learning problem statistically “easier” within the interpolation space $[\mathcal H_K]^3$. That is simple, transparent, and potentially reusable.
- **Rates in a standard spectral regime:** Under polynomial eigenvalue decay, the obtained risk rate matches the classical optimal shapes (removing some log factors).
- **Proof organization:** The intended high-level flow (kernel consistency → local complexity → early stopping) is standard and clear enough for verification, with explicit references to the relevant lemmas and theorems.

**Weaknesses:**

### (A) The “kernel-equivalence” of the GD trajectory is **not established convincingly**.

The key step is the **one-step recursion** on the prediction vector:
$$
\widehat y(t+1) - \widehat y(t) = -\eta K^{(\mathrm{attn})}_n\big(\widehat y(t) - y\big) + E(t),
$$
where $E(t)$ aggregates three residuals (from feature drift, replacing $\sigma(W(t),S)$ by $\sigma(W(0),S)$, and from $\hat K^{(\mathrm{attn})}$ to $K^{(\mathrm{attn})}$).

- In the draft, the **norm bound** of $E(t)$ is **loose at the point of use** (the block of equations around (85)–(91)). Without *subsequent* substitutions, the bound essentially reads like
  $$
  \|E(t)\|_2 \lesssim R^2\sqrt{mn}+\eta c_u \sqrt n \big(R + C_1(m,d,1/n) + 1/\sqrt N\big),
  $$
  i.e., **potentially $O(\sqrt n)$**—the same scale as the main term. This does **not** by itself show that the residual is controlled.

- The paper attempts to “fix” this later by introducing a **drift radius** $R$ and showing
  $$
   R =\max_{t' \le T} (\max_r \|w_r(t') - w_r(0)\|_2 \max_r |a_r(t')|) \approx \frac{\eta T}{\sqrt m}\times \mathrm{polylog}(n,m,d),
  $$
  so that after substituting **$R = \eta T/\sqrt m$** and imposing **large-width** and **large-$N$** conditions, one can conclude $\|E(t)\|_2 = O(\varepsilon_n^4 \sqrt n)$, and the cumulative deviation over $T$ steps is $O(\varepsilon_n^2 \sqrt n)$.

- **Issue:** This control fundamentally **hinges on a very specific normalization** of the model that **forces** small steps in function space:
  - The network output is scaled by $1/\sqrt m$.
  - The attention matrix is defined with an additional $1/(Nm)$ factor, i.e., $A = \frac{1}{Nm}\sigma(W(0),Q)\sigma(W(0),Q)^\top$.

  Together these inject **extra $1/\sqrt m$** and **$1/m$** factors into the one-step prediction change. As a result, the “large width $\Rightarrow$ small drift $\Rightarrow$ small residual” story appears **largely an artifact of these ad hoc scalings**, not a robust phenomenon of channel attention per se. Under more standard channel-mixing/attention parameterizations (where the mixing map does **not** vanish as $m$ grows), the argument $R \sim \eta T/\sqrt m$ may **fail**, and thus the “kernel equivalence” may **not** hold.

- **Takeaway:** The draft **does not** demonstrate that GD *actually* stays in a “kernel regime” **unless** one accepts these strong, crafted scalings and the corresponding width/sample scaling laws (**e.g.,** $m \gtrsim (\eta T)^2 \cdot \mathrm{polylog}$, $T \asymp \varepsilon_n^{-2}$, $m \gtrsim \varepsilon_n^{-16}$, $N \gtrsim \varepsilon_n^{-8}$). This severely limits the external relevance of the result.

### (B) **Learning-rate stability** is not tied to spectral quantities.

The draft allows $\eta = \Theta(1)$ but **does not** provide a clear, explicit **stability range** (e.g., $\eta < 2/\lambda_{\max}(K^{(\mathrm{attn})}_n)$ or an equivalent statement). The entire “kernel-like” recursion must hold **for all $t \le T$**; without a stability condition linked to spectrum, the behavior under practical $\eta \in [0.1,1]$ remains unclear, especially when width is **not** inflated to the aggressive scales assumed.

### (C) **Parameter update inconsistency** (clarity/consistency).

The draft has mutually inconsistent statements about the second-layer weights $a$: one place says “optimize only $W$, keep $a$ fixed at $\pm 1$”, another gives a GD update for $a$, and elsewhere it says “initialize $a=0$”. This impacts the definition and control of $R$ and must be fixed.

### (D) **Dependence on $Q \sim P$** is under-quantified in the main theorem.

Both $A$ and $\hat K^{(\mathrm{attn})}$ require samples $Q$ from the feature distribution $P$. The draft mentions using approximate $Q \sim \hat P$ (e.g., via a generative model) when $P$ is unknown, but the **bias** induced by using $\hat P$ is **not** carried into the **final** bound; it appears only as a separate uniform-convergence lemma. This creates a gap between the assumptions and the main guarantee.

### (E) **“Beyond NTK”** phrasing is potentially misleading.

The construction still works wholly in a **kernel** framework—simply with a *different* kernel $K^{(\mathrm{attn})}$. It would be more accurate to say the method **changes the kernel** (and the RKHS) rather than “goes beyond” kernel views. The current phrasing may overstate the conceptual leap.

**Questions:**

Please see the weakness

---

### Official Review · Reviewer_CuhC · 2025-11-05

**Soundness:** 3
**Presentation:** 1
**Contribution:** 2
**Rating:** 2
**Confidence:** 2

**Summary:**

This paper studies nonparametric regression using an overparameterized two-layer neural network with XCA-style channel attention (used in prior empirical work as well). The authors show that gradient descent with early stopping achieves a sharp risk bound of $O(\varepsilon_n^2)$, where $\varepsilon_n$ is the critical population rate of a newly defined attention kernel $K_{\mathrm{attn}}$.
Under polynomial eigenvalue decay $\lambda_j \asymp j^{-2\alpha}$, the bound specializes to the minimax-optimal rate $O(n^{-6\alpha/(6\alpha+1)})$. The analysis relies on (i) a decomposition of the network output into an RKHS component plus a small $L_\infty$ residual and (ii) sharp generalization control via local Rademacher complexity.
Overall, the work provides one of the first theoretical justifications for the benefit of channel attention in nonparametric regression.

**Strengths:**

The paper presents a new and rigorous theoretical analysis of channel attention in overparameterized neural networks, introducing the ``attention kernel'' that yields provably sharper generalization bounds than the standard NTK.
The results are technically deep. Overall, the contribution is technical and solid.

**Weaknesses:**

The paper is technically sound, though I am not familiar with either of the relevant prior work on the channel attention itself, kernel analysis with eigendecay, or NTK analysis without channel attention.

**Presentation** Having acknowledged my own limitations, my main problem with this paper is the presentation. The paper has serious presentation issues. While it is a technically dense paper, I understand that it is difficult to write, but authors should seriously invest efforts and time to present what is the best way to present their results so it is easy to gauge the contributions.

(1) The main table is referred to in the Intro but placed on page 6. That is probably fine, although I feel it is much more important keep this in the intro (or some easy to parse version of it), to help understand the contribution. Another big issue with the table is that the font therein is excessively small, so one cannot even read.

(2) Also, I would feel experiment plots should have been part of the main paper, at least one of the Figures 1,2 or 3.

(3) Authors should also identify clearly what are the key contributions are, and emphasize them in equal proportion in the abstract and intro. Currently, I don't see it that way. Is one of the main contributions, an improvement over the $\log(1/\delta)$ factor from prior work. If so, then it should be highlighted clearly in the intro as well, with better details than in the abstract. If it is not the main and only minor discussion point, then it should be removed from the abstract to allow the reader to focus on what are more significant contributions. (See the paragraph about contribution).

(4) Authors should revisit what the minimal notation needed to introduce their key ideas and theorems, and remove some trivial notations with a meaningless notation paragraph, which occupies space. This can be easily deferred to appendices, or even mostly removed, I would say.

(5) What is the point of defining $\mathcal{X}=\mathbb{S}^{d-1}$ if you always consider just one domain? Can you directly not use $\mathbb{S}^{d-1}$. Even if you want to use the additional notation for the domain, why define $\mathbb{S}^{d-1}$ first and then define $\mathcal{X}$ after 8 lines of other unrelated notation. Can they not be defined in the same line. And even if you were to define it, I might still use $\mathbb{S}^{d-1}$ at important places. It was unclear to me how inner product can be taken for abstract $\mathcal{X}$ and had hard time finding where do you say even that it is real valued or on the sphere.

(6) Also, in the intro, the way I see is there is one paragraph on ``your contribution" (line 72-91) between the paragraphs on notation and existing work. If so, then it should also get a paragraph status. The way it is written seems like you are still discussing existing work.

**Concerns on Technical Contributions** While I like papers with technical contributions, here I am also not sure if technical contributions are entirely significant. Thus, I will have my concerns unless some other reviewer who is more familiar with the background can vouch for the novelty.

(1) The way I see it is that the main contribution is in providing an analysis with a channel attention kernel. And show that it can be better than a vanilla kernel. Authors should highlight what are the exact challenges in achieving this over prior results. Some of it is in page 9, but still difficult to say what exactly is the technique that was difficult to develop for vanilla kernel.

(2) Also, if I understood, the difference between $\epsilon_{K,n}$ vs $\epsilon_n$ is only about vanilla kernel vs attention kernel, right?

(3) If so, then only imrpovement you are showing over other kernel is for the special case of power decay type input distribution, and the imrovement is in getting rid of $\log^2(1/\delta)$ factor, right? Any other distribution where there is significant gain?

(4) Also, would you not say that the rate with the above factors is minimax optimal as well? If not, how are you precisely defining minimax optimality?

If this is all what you are doing, then there is much better and efficient way to rewrite the abstract, intro and paper, and these points can be conveyed in few line with even different type of table. Overall,  I feel like this comes out as a confused paper that is not clear of what are its strength, and what it thinks is its core contribution.

**Questions:**

See weaknesses.

I don't think I fully understand what is going on in Table 1, and how I should be comparing results between different assumptions, kernels, and rates from prior work. I would like to go over this again as to what exactly is done in prior work, how it compares with the results here, and where exactly are the improvements. What new techniques are neeeded to achieve this improvements.

---

### Note · Authors · 2025-11-12

I have read and agree with the venue's withdrawal policy on behalf of myself and my co-authors.